# On the Shift Invariance of Max Pooling Feature Maps in Convolutional Neural Networks

## Abstract

This paper focuses on improving the mathematical interpretability of convolutional neural networks (CNNs) in the context of image classification. Specifically, we tackle the instability issue arising in their first layer, which tends to learn parameters that closely resemble oriented band-pass filters when trained on datasets like ImageNet. Subsampled convolutions with such Gabor-like filters are prone to aliasing, causing sensitivity to small input shifts. In this context, we establish conditions under which the max pooling operator approximates a complex modulus, which is nearly shift invariant. We then derive a measure of shift invariance for subsampled convolutions followed by max pooling. In particular, we highlight the crucial role played by the filter's frequency and orientation in achieving stability. We experimentally validate our theory by considering a deterministic feature extractor based on the dual-tree complex wavelet packet transform, a particular case of discrete Gabor-like decomposition.

## 1 Introduction

Understanding the mathematical properties of deep convolutional neural networks (CNNs) (LeCun et al., 2015) remains a challenging issue today. On the other hand, wavelet and multi-resolution analysis are built upon a well-established mathematical framework. They have proven to be efficient for tasks such as signal compression and denoising (Vetterli, 2001), and have been widely used as feature extractors for signal, image and texture classification (Laine & Fan, 1993; Pittner & Kamarthi, 1999; Yen, 2000; Huang & Aviyente, 2008). There is a broad literature revealing strong connections between these two paradigms, as discussed in Sections 1.1 and 1.2. Inspired by this line of research, the present paper extends existing knowledge about CNN properties. Specifically, we assess the shift invariance of max pooling feature maps through both theoretical and empirical approaches in the context of image classification, by leveraging the properties of oriented band-pass filters.

### 1.1 Motivations and Main Contributions

CNNs process input images through convolutions and nonlinear pooling operations, transforming them into high-level feature vectors that are subsequently used for the task at hand. In image classification, the feature vectors are fed into a linear classifier. To achieve high classification accuracy, a convolutional network must preserve discriminative image features while reducing intra-class variability (LeCun et al., 1998; Bruna & Mallat, 2013). An important and often desired property of CNNs is their ability to remain invariant to small input transformations, such as translations, rotations, distortions, or scaling (Liao & Peng, 2010; Bruna & Mallat, 2013; Sifre & Mallat, 2013; Bietti & Mairal, 2017; Wiatowski & Bölcskei, 2018; Cahill et al., 2024).

In particular, an image in which the main subject is slightly shifted from its original position should retain its initial classification. This property, known as *translation invariance* or *shift invariance*, is crucial for model robustness, as its absence can negatively impact the model's predictive performance. Desirable properties such as shift invariance in deep learning models can be achieved either through extensive training and data augmentation, or through careful architectural design. Relying on the former may result in increased computational cost, reduced generalizability, and limited interpretability. In contrast, understanding when and

why a model exhibits such properties by design can provide valuable insights that can guide the development of more efficient and robust architectures. The main focus of this paper is to assess whether shift invariance is inherently guaranteed by the model's architecture. Since perfect invariance is rarely achieved, we also use the term *stability* to refer to this behavior.

More specifically, we focus on a commonly observed phenomenon in CNNs when trained on image datasets: many convolution kernels in the first layer resemble band-pass, oriented waveforms (Yosinski et al., 2014; Rai & Rivas, 2020), referred to as *Gabor-like filters*. This class of filters is central to our analysis; for mathematical tractability, a formal definition is provided in (11). Whether the features extracted by Gabor-like filters remain stable under translations has been partly addressed by Azulay & Weiss (2019); Zhang (2019), who highlight that strided convolution and pooling operations can significantly diverge from shift invariance, due to aliasing effects when subsampling high-frequency signals. In response, Zhang (2019); Karras et al. (2021); Vasconcelos et al. (2021); Zou et al. (2023) introduced antialiasing methods based on low-pass filtering, improving both stability and predictive performance—albeit at the cost of some loss of information.

In the current paper, we investigate shift invariance properties that are already present in standard CNN architectures, even before the application of antialiasing techniques such as those mentioned above. Specifically, we show that, under certain conditions that we establish, the max pooling operator can partially restore shift invariance that is otherwise degraded by subsampled convolutions. We unveil a connection between the output of the first max pooling layer and the pointwise magnitude of the convolution with complex Gabor-like filters, a quantity known to be nearly shift invariant. This work offers a promising direction for improving shift invariance in CNNs while preserving high-frequency information—unlike the previously-mentioned approaches.

We note that this study primarily focuses on the first layer of CNNs, where a significant proportion of learned convolution kernels typically resemble Gabor-like filters. Extending the analysis to deeper layers would require a different theoretical framework which falls beyond the scope of this paper. Nevertheless, early CNN layers play a critical role in model performance, as they extract low-level geometric features essential for building more complex representations in subsequent stages (Oyallon et al., 2017). Furthermore, as discussed in Section 7, instabilities introduced in the first layer can propagate through the network—a phenomenon we empirically observed in a recent study published as a conference paper (Authors, 2024).

Before proceeding further, we emphasize that our study is not limited to purely convolutional architectures. In recent years, self-attention mechanisms have gained significant interest in computer vision due to their ability to model complex, long-range dependencies in image representations. Notably, the *Vision Transformer* (ViT) (Dosovitskiy et al., 2021) and the *Swin Transformer* (Liu et al., 2021) adapt the Transformer (Vaswani et al., 2017)—originally developed for natural language processing (NLP)—to computer vision tasks. Unlike CNNs, these models operate without convolutional layers. Instead, input images are partitioned into fixed-size patches that serve as tokens for the self-attention modules. However, more recent research has explored hybrid architectures that integrate self-attention with convolutional components (Wu et al., 2021; Yuan et al., 2021; Hassani et al., 2022; Li et al., 2023; Yin et al., 2024). This approach allows for reducing the amount of labeled data required, while achieving faster training and improving generalizability. In particular, the first layers of a CNN can be used as a "convolutional token embedding," replacing the naive patch extraction used in original models. The theoretical framework presented in this paper can also apply to such hybrid architectures, providing a better understanding of the invariance properties of the inputs to self-attention modules. Note however that we do not claim direct applicability to all hybrid CNN-Transformer models. Any adjustments that may be needed in this context are left for future work.

## 1.2   Related Work

Analyzing the invariance properties of CNNs is critical as it enables to identify their shortcomings and provides an opportunity to enhance their performance. In recent years, several works focused on this topic.

### 1.2.1 Wavelet Scattering Networks

Most notably, Bruna & Mallat (2013) developed a family CNN-like architectures, named *wavelet scattering networks* (ScatterNets), based on a succession of complex convolutions with wavelet filters followed by nonlinear modulus pooling. They produce translation-invariant image representations which are stable to deformation and preserve high-frequency information (Mallat, 2012; 2016; Czaja et al., 2024). A variation has been proposed by Sifre & Mallat (2013) to include rotational invariance. ScatterNets achieve strong performance on handwritten digits and texture datasets, but do not scale well to more complex ones. To overcome this, Oyallon et al. (2017; 2018) introduced hybrid ScatterNets, where the scattering coefficients are fed into a standard CNN architecture, showing that the network complexity can be reduced while keeping competitive performance. Derived models include ScatterNets built upon the dual-tree complex wavelet transform (Singh & Kingsbury, 2017), learnable and parametric ScatterNets (Cotter & Kingsbury, 2019; Gauthier et al., 2022), geometric ScatterNets operating on Riemanian manifolds (Perlmutter et al., 2020), and graph ScatterNets (Gama et al., 2019; Zou & Lerman, 2020). Also worth mentioning, Czaja & Li (2019; 2020) studied ScatterNets based on uniform covering frames, i.e., frames splitting the frequency domain into windows of roughly equal size, much like the dual-tree complex wavelet packet transform (DT-$\mathbb{C}$WPT), as used in the present paper. Other works by Zarka et al. (2020; 2021) proposed to sparsify wavelet scattering coefficients by learning a dictionary matrix, to learn $1 \times 1$ convolutions between feature maps of scattering coefficients and to apply soft thresholding to reduce within-class variability.

ScatterNets are specifically designed to meet some desired properties. As deep learning architectures with well-established mathematical properties, they are sometimes used as explanatory models for standard, freely-trained networks. However, whether their properties are transferable to a broader class of models is unclear, because the former rely on complex-valued convolutions whereas more conventional architectures exclusively employ real-valued kernels. Moreover, the modulus operator is used as an activation and pooling layer in ScatterNets, whereas standard CNNs implement pointwise nonlinear operators such as ReLU and spatial pooling layers such as max pooling. This limitation has been pointed out by Tygert et al. (2016) as an argument in favor of complex-valued CNNs. In this context, our work seeks evidence that properties established for complex-valued networks are—to some extent—embedded in standard architectures.

### 1.2.2 Invariance Studies in CNNs

Wiatowski & Bölcskei (2018) considered a wide variety of feature extractors involving convolutions, Lipschitz-continuous non-linearities and pooling operators. The paper shows that outputs become more translation invariant with increasing network depth. Additionally, Cahill et al. (2024) designed a family of operators called *max filters*, which encompass a wide variety of operators including, in specific cases, the max pooling operator. Stability with respect to diffeomorphisms were established, following the ideas developed for scattering networks. However, these results do not fully extend to the discrete framework, because subsampled convolutions with band-pass real-valued filters can introduce aliasing artifacts, resulting in instability to translations (Azulay & Weiss, 2019; Zhang, 2019). The current paper specifically addresses this issue.

Another line of work is focused on modeling and studying CNNs from the point of view of convolutional kernel networks (Bietti & Mairal, 2019a;b; Scetbon & Harchaoui, 2020; Bietti, 2022; Chen, 2023). These authors showed that certain classes of CNNs are contained in the reproducing kernel Hilbert space (RKHS) of a multilayer convolutional kernel representation. As such, stability metrics are estimated, based on the RKHS norm which is difficult to control in practice. Kernel representations do not seem to suffer from aliasing effects; this can be explained by the Gaussian pooling layers that have been employed instead of max pooling: by discarding high-frequency information, shift invariance is preserved.

Finally, some papers studied stability of CNNs in a broader sense, measured in terms of Lipschitz continuity (Szegedy et al., 2014; Balan et al., 2018; Virmaux & Scaman, 2018; Pérez et al., 2020; Zou et al., 2020; Gupta et al., 2022; Zühlke & Kudenko, 2024). However, the Lipschitz bounds, which have been obtained theoretically, are generally several orders of magnitude higher than empirical results. This discrepancy may be due to the fact that these bounds were obtained for generic situations and represent overly conservative worst-case scenarios, rather than typical real-world situations. Furthermore, the specific case of convolutions with band-pass Gabor-like filters have been overlooked, except for Pérez et al. (2020).

In summary, we have identified the following blind spots in the literature, regarding the topic of studying shift invariance in CNNs.

- The effect of the max pooling operator on network stability under small input shifts has not been investigated, particularly when used in combination with Gabor-like convolutions.

- While the shift invariance of CNNs tends to increase with network depth in the continuous framework (as formally introduced in Section 2.1), in the discrete case (as implemented in practice), the presence of subsampled convolutions with oriented band-pass filters can lead to aliasing artifacts. To our knowledge, the literature lacks theoretical studies that take these aliasing effects into account.

- Although extensive studies have been conducted on complex-valued convolutions followed by modulus, a link is missing to extend these results to standard CNNs, which implement real-valued convolutions and spatial pooling operators.

All these points have been tackled in the present paper, from both theoretical and empirical perspectives.

### 1.3 Paper Outline

In what follows, $l_{\mathbb{R}}^2(\mathbb{Z}^2)$ and $l_{\mathbb{C}}^2(\mathbb{Z}^2)$ represent the discrete spaces of square-summable two-dimensional sequences with values in $\mathbb{R}$ and $\mathbb{C}$, respectively. Let $W \in l_{\mathbb{C}}^2(\mathbb{Z}^2)$ denote a two-dimensional band-pass, oriented and analytic *Gabor-like* filter, for which a formal definition will be provided in (11). We first consider an operator, referred to as *real-max-pooling* ($\mathbb{R}$Max), which computes the subsampled cross-correlation between an input image $X \in l_{\mathbb{R}}^2(\mathbb{Z}^2)$ and the real part of $W$; then calculates the maximum value over a sliding discrete grid:

$$U_{m,\,q}^{\max}[W] : X \mapsto \mathrm{MaxPool}_q\left(\left(X * \overline{\mathrm{Re}\,W}\right) \downarrow m\right), \tag{1}$$

where $m \in \mathbb{N} \setminus \{0\}$ denotes a subsampling factor (corresponding to the stride of the convolution), $\overline{V}$ denotes the "flipped" sequence for any given $V \in l_{\mathbb{R}}^2(\mathbb{Z}^2)$ or $l_{\mathbb{C}}^2(\mathbb{Z}^2)$, satisfying, for any $\boldsymbol{n} \in \mathbb{Z}^2$,

$$\overline{V}[\boldsymbol{n}] := V[-\boldsymbol{n}], \tag{2}$$

and $*$, $\downarrow$ respectively refer to the convolution and subsampling operations, defined by

$$(X * \overline{V})[\boldsymbol{n}] := \sum_{\boldsymbol{p} \in \mathbb{Z}^2} X[\boldsymbol{p}]\,\overline{V}[\boldsymbol{n} - \boldsymbol{p}] \qquad \text{and} \qquad (Y \downarrow m)[\boldsymbol{n}] := Y[m\boldsymbol{n}]. \tag{3}$$

In the above expression, $\mathrm{MaxPool}_q$ selects the maximum value over a sliding grid of size $(2q + 1) \times (2q + 1)$, with a subsampling factor of 2. More formally, for any $Y \in l_{\mathbb{R}}^2(\mathbb{Z}^2)$ and any $\boldsymbol{n} \in \mathbb{Z}^2$,

$$\mathrm{MaxPool}_q(Y)[\boldsymbol{n}] := \max_{\|\boldsymbol{p}\|_\infty \leq q} Y[2\boldsymbol{n} + \boldsymbol{p}]. \tag{4}$$

On the other hand, we consider an operator, referred to as *complex-modulus* ($\mathbb{C}$Mod), computing the modulus of subsampled cross-correlation between $X$ and $W$:

$$U_m^{\mathrm{mod}}[W] : X \mapsto \left|(X * \overline{W}) \downarrow (2m)\right|. \tag{5}$$

To improve readability and ease computations, we use standard convolution ($*$) throughout the paper instead of cross-correlation ($\star$). This choice allows us to leverage well-known mathematical properties—most notably, that the Fourier transform of a convolution between two functions or vectors equals the product of their Fourier transforms.

First, we show that, under the hypotheses stated above, $\mathbb{C}$Mod is stable with respect to small input shifts (Section 2, with main result stated in Theorem 1). We then establish conditions on the filter's frequency and orientation under which $\mathbb{C}$Mod and $\mathbb{R}$Max produce comparable outputs (Section 3, with main result stated in Theorem 2):

$$U_m^{\mathrm{mod}}[W](X) \approx U_{m,\,q}^{\max}[W](X). \tag{6}$$

We deduce a measure of shift invariance for $\mathbb{R}\mathrm{Max}$ operators, which benefits from the stability of $\mathbb{C}\mathrm{Mod}$ (Section 4, with main result stated in Theorem 3). Next, we extend our results to multichannel operators (*i.e.*, applied on RGB input images), such as implemented in conventional CNN architectures (Section 5, Corollaries 2 to 4). Our framework therefore provides a theoretical grounding to study these networks.

**Remark 1.** In the above definitions, cross-correlations are computed with a subsampling factor which is twice larger for $\mathbb{C}\mathrm{Mod}$, compared to $\mathbb{R}\mathrm{Max}$. However, since max pooling is also computed with subsampling, both operators have the same subsampling factor of $2m$.

Finally, in Section 6, we assess our theoretical findings on a deterministic setting based on the dual-tree complex wavelet packet transform (DT-$\mathbb{C}$WPT), a particular case of discrete Gabor-like decomposition with perfect reconstruction properties (Bayram & Selesnick, 2008). DT-$\mathbb{C}$WPT spawns a set of convolution kernels which tile the Fourier domain into square regions of identical size. Such kernels possess characteristics that are comparable to those found in the first convolution layer of CNNs after training with image datasets such as ImageNet (Russakovsky et al., 2015). More specifically, given an input image, we compute the mean square error between the outputs of $\mathbb{C}\mathrm{Mod}$ and $\mathbb{R}\mathrm{Max}$, for each wavelet packet filter. We then observe that shift invariance, when measured on $\mathbb{R}\mathrm{Max}$ feature maps, is nearly achieved when they remain close to $\mathbb{C}\mathrm{Mod}$ outputs. We therefore establish a domain of validity for shift invariance of the $\mathbb{R}\mathrm{Max}$ operator.

This work builds upon an idea sketched by Waldspurger (2015, pp. 190–191), which suggests a potential connection between the combinations "real wavelet transform $\rightarrow$ max pooling" on the one hand and "complex wavelet transform $\rightarrow$ modulus" on the other hand. Consequently, the former operator could inherit the shift-invariance properties of the latter by leveraging the properties of the max pooling operator. The formulation, however, was limited to continuous images and filters, and the max pooling layer operating over continuous windows. Our study extends this idea to discrete convolutions and max pooling grids. As shown in this paper, the initial principles do not fully extend to this more realistic framework. To address this limitation, we adopted a probabilistic point of view. By doing so, we revealed that shift invariance is inherently dependent on the filter's frequency.

## 2   Shift Invariance of $\mathbb{C}\mathrm{Mod}$ Outputs

The primary goal of this paper is to theoretically establish conditions for near-shift invariance at the output of the first max pooling layer. In this section, we start by proving shift invariance of $\mathbb{C}\mathrm{Mod}$ operators. Then, in Section 3, we establish conditions under which $\mathbb{R}\mathrm{Max}$ and $\mathbb{C}\mathrm{Mod}$ produce closely related outputs. Finally, in Section 4, we derive a probabilistic measure of shift invariance for $\mathbb{R}\mathrm{Max}$.

### 2.1   Notations

The complex conjugate of any number $z \in \mathbb{C}$ is denoted by $z^*$. For any $p \in \mathbb{R}_{>0} \cup \{\infty\}$, $\boldsymbol{x} \in \mathbb{R}^2$ and $r \in \mathbb{R}_+$, we denote by $B_p(\boldsymbol{x},\, r) \subset \mathbb{R}^2$ the closed $l^p$-ball with center $\boldsymbol{x}$ and radius $r$. When $\boldsymbol{x} = \boldsymbol{0}$, we write $B_p(r)$.

**Continuous Framework**   Considering a measurable subset $E$ of $\mathbb{R}^2$, we denote by $L^2_{\mathbb{C}}(E)$ the Hilbert space of square-integrable functions $F : E \rightarrow \mathbb{C}$. Whenever we talk about equality in $L^p_{\mathbb{C}}(E)$ or inclusion in $E$, it shall be understood as "almost everywhere with respect to the Lebesgue measure." Additionally, we denote by $L^2_{\mathbb{R}}(E) \subset L^2_{\mathbb{C}}(E)$ the subspace of real-valued functions. For any $F \in L^2_{\mathbb{C}}(\mathbb{R}^2)$, $\overline{F}$ denotes its flipped version: $\overline{F}(\boldsymbol{x}) := F(-\boldsymbol{x})$.

The 2D Fourier transform of any $F \in L^2_{\mathbb{C}}(\mathbb{R}^2)$ is denoted by $\widehat{F} \in L^2_{\mathbb{C}}(\mathbb{R}^2)$, such that

$$\forall \boldsymbol{\nu} \in \mathbb{R}^2,\ \widehat{F}(\boldsymbol{\nu}) := \iint_{\mathbb{R}^2} F(\boldsymbol{x})\mathrm{e}^{-i\langle \boldsymbol{\nu},\, \boldsymbol{x}\rangle}\,\mathrm{d}^2\boldsymbol{x}. \tag{7}$$

For any $\varepsilon > 0$ and $\boldsymbol{\nu} \in \mathbb{R}^2$, we denote by $\mathcal{V}(\boldsymbol{\nu},\, \varepsilon) \subset L^2_{\mathbb{C}}(\mathbb{R}^2)$ the set of functions whose Fourier transform is supported in a square region of size $\varepsilon \times \varepsilon$ centered in $\boldsymbol{\nu}$:

$$\mathcal{V}(\boldsymbol{\nu},\, \varepsilon) := \left\{ \Psi \in L^2_{\mathbb{C}}(\mathbb{R}^2) \ \middle|\ \mathrm{supp}\,\widehat{\Psi} \subset B_{\infty}(\boldsymbol{\nu},\, \varepsilon/2) \right\}. \tag{8}$$

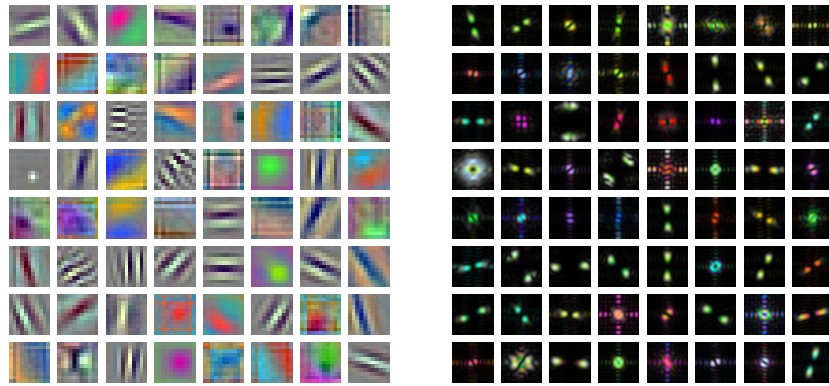

Figure 1. Spatial (left) and Fourier (right) representations of convolution kernels in the first layer of AlexNet, after training with ImageNet ILSVRC 2012-2017 (Russakovsky et al., 2015). Each kernel connects the 3 RGB input channels to one of the 64 output channels.

$\boldsymbol{\nu}$ and $\varepsilon$ are respectively referred to as *characteristic frequency* and *bandwidth*. Finally, for any $\boldsymbol{h} \in \mathbb{R}^2$, we consider the translation operator, denoted by $\mathcal{T}_{\boldsymbol{h}}$, defined by

$$\mathcal{T}_{\boldsymbol{h}}F : \boldsymbol{x} \mapsto F(\boldsymbol{x} - \boldsymbol{h}). \tag{9}$$

**Discrete Framework**   We denote by $l^2_{\mathbb{C}}(\mathbb{Z}^2)$ the space of 2D complex-valued square-summable sequences, represented by straight capital letters. Indexing is made between square brackets: $\forall \mathrm{X} \in l^2_{\mathbb{C}}(\mathbb{Z}^2), \forall \boldsymbol{n} \in \mathbb{Z}^2, \mathrm{X}[\boldsymbol{n}] \in \mathbb{C}$, and we denote by $l^2_{\mathbb{R}}(\mathbb{Z}^2) \subset l^2_{\mathbb{C}}(\mathbb{Z}^2)$ the subset of real-valued sequences. For any $\mathrm{V} \in l^2_{\mathbb{C}}(\mathbb{Z}^2)$, $\overline{\mathrm{V}}$ denotes its "flipped" version as defined in (2). The convolution and subsampling operators, respectively denoted by $*$ and $\downarrow$, are defined in (3). 2D images, feature maps and convolution kernels are considered as elements of $l^2_{\mathbb{C}}(\mathbb{Z}^2)$. Additionally, multichannel arrays of 2D sequences are denoted by bold straight capital letters, for instance: $\mathbf{X} := (\mathrm{X}_k)_{k \in \{0, \dots, K-1\}}$. Note that indexing starts at 0 to comply with practical implementations.

The 2D discrete-time Fourier transform of any $\mathrm{X} \in l^2_{\mathbb{C}}(\mathbb{Z}^2)$, denoted by $\widehat{\mathrm{X}} \in L^2_{\mathbb{C}}([-\pi, \pi]^2)$, is defined by

$$\forall \boldsymbol{\theta} \in [-\pi, \pi]^2, \, \widehat{\mathrm{X}}(\boldsymbol{\theta}) := \sum_{\boldsymbol{n} \in \mathbb{Z}^2} \mathrm{X}[\boldsymbol{n}] \mathrm{e}^{-i\langle \boldsymbol{\theta}, \boldsymbol{n} \rangle}. \tag{10}$$

For any $\kappa \in \,]0, \, 2\pi]$ and $\boldsymbol{\theta} \in B_{\infty}(\pi)$, we denote by $\mathcal{J}(\boldsymbol{\theta}, \kappa) \subset l^2_{\mathbb{C}}(\mathbb{Z}^2)$ the set of 2D sequences whose Fourier transform is supported in a square region of size $\kappa \times \kappa$ centered in $\boldsymbol{\theta}$:

$$\mathcal{J}(\boldsymbol{\theta}, \kappa) := \left\{ \mathrm{W} \in l^2_{\mathbb{C}}(\mathbb{Z}^2) \, \Big| \, \mathrm{supp}\,\widehat{\mathrm{W}} \subset B_{\infty}(\boldsymbol{\theta}, \kappa/2) \right\}. \tag{11}$$

As in the continuous framework, $\boldsymbol{\theta}$ and $\kappa$ are respectively referred to as *characteristic frequency* and *bandwidth*. The elements of $\mathcal{J}(\boldsymbol{\theta}, \kappa)$ are designated as *Gabor-like filters*.

**Remark 2.** The support $B_{\infty}(\boldsymbol{\theta}, \kappa/2)$ actually lives in the quotient space $[-\pi, \pi]^2 / (2\pi\mathbb{Z}^2)$. Consequently, when $\boldsymbol{\theta}$ is close to an edge, a fraction of this region is located at the far end of the frequency domain. From now on, the choice of $\boldsymbol{\theta}$ and $\kappa$ is implicitly assumed to avoid such a situation.

## 2.2   Intuition

In many CNNs for computer vision, input images are first transformed through subsampled (or strided) convolutions. For instance, in AlexNet, convolution kernels are of size $11 \times 11$ and the subsampling factor is equal to 4. Figure 1 displays the corresponding kernels after training with ImageNet. This linear transform is generally followed by rectified linear unit (ReLU) and max pooling.

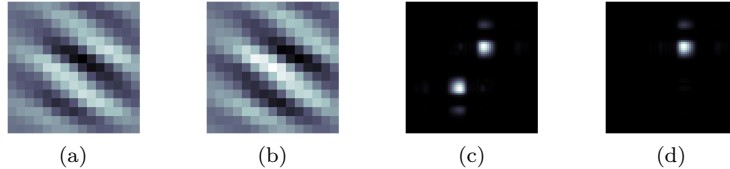

|        |        |        |        |
|--------|--------|--------|--------|
| (a)    | (b)    | (c)    | (d)    |

Figure 2. (a), (b): Real and imaginary parts of a Gabor-like filter W as defined in (12). (c), (d): Magnitude spectra (modulus of the Fourier transform) of V and W, respectively.

We can observe that many kernels display oscillating patterns with well-defined orientations (Gabor-like filters). We denote by $V \in l_{\mathbb{R}}^2(\mathbb{Z}^2)$ one of these "well-behaved" filters. Its Fourier spectrum roughly consists in two bright spots which are symmetric with respect to the origin.[1] Next, we consider a complex-valued companion $W \in l_{\mathbb{C}}^2(\mathbb{Z}^2)$ such that

$$\widehat{W}(\boldsymbol{\omega}) := \left(1 + \operatorname{sgn}\langle \boldsymbol{\omega}, \, \boldsymbol{u} \rangle\right) \cdot \widehat{V}(\boldsymbol{\omega}) \qquad \forall \boldsymbol{\omega} \in [-\pi, \, \pi]^2 \,, \tag{12}$$

where $\boldsymbol{u}$ denotes a unit vector orthogonal to the filter's orientation.

We can show that V is the real part of W, and that $W = V + i\mathcal{H}(V)$, where $\mathcal{H}$ denotes the two-dimensional Hilbert transform as introduced by Havlicek et al. (1997). It satisfies

$$\widehat{\mathcal{H}(V)}(\boldsymbol{\omega}) := -i \operatorname{sgn}\langle \boldsymbol{\omega}, \, \boldsymbol{u} \rangle \, \widehat{V}(\boldsymbol{\omega}). \tag{13}$$

As a consequence, $\widehat{W}$ is equal to $2\widehat{V}$ on one half of the Fourier domain, and 0 on the other half. Therefore, only one bright spot remains in the spectrum. We refer the reader to Figure 2 for visual example of complex-valued Gabor-like filter. It turns out that such complex filters with high frequency resolution produce stable signal representations, as we will see in Section 2. In the subsequent sections, we then wonder whether this property is kept when considering the max pooling of real-valued convolutions.

In what follows, W will be referred to as a discrete Gabor-like filter, and the coefficients resulting from the convolution with W will be referred to as discrete Gabor-like coefficients. The aim of this section is to show that, if the convolution kernels $W \in l_{\mathbb{C}}^2(\mathbb{Z}^2)$ belong to $\mathcal{J}(\boldsymbol{\theta}, \kappa)$ as introduced in (11), then $\mathbb{C}\mathrm{Mod}$ is nearly shift-invariant. To clarify, we establish that

$$U_m^{\mathrm{mod}}[W](X) \approx U_m^{\mathrm{mod}}[W](\mathcal{T}_{\boldsymbol{u}}X), \tag{14}$$

for "small" translation vectors $\boldsymbol{u} \in \mathbb{R}^2$, where a formal definition of the translation operator will be defined in (28). This result is hinted by Kingsbury & Magarey (1998) but not formally proven.

### 2.3 Continuous Framework

We introduce several results regarding functions defined on the continuous space $\mathbb{R}^2$. Near-shift invariance on discrete 2D sequences will then be derived from these results by taking advantage of sampling theorems. Lemma 1 below is adapted from Waldspurger (2015, pp. 190–191).

**Lemma 1.** *Given $\varepsilon > 0$ and $\boldsymbol{\nu} \in \mathbb{R}^2$, let $\Psi \in \mathcal{V}(\boldsymbol{\nu}, \varepsilon)$ denote a complex-valued filter such as defined in (8). Next, for any real-valued function $F \in L_{\mathbb{R}}^2(\mathbb{R}^2)$, we consider the complex-valued function $F_0 \in L_{\mathbb{C}}^2(\mathbb{R}^2)$ defined by*

$$F_0 : \boldsymbol{x} \mapsto (F * \overline{\Psi})(\boldsymbol{x}) \, \mathrm{e}^{i\langle \boldsymbol{\nu}, \, \boldsymbol{x} \rangle}. \tag{15}$$

*Then $F_0$ is low-frequency. Specifically,*

$$\operatorname{supp} \widehat{F_0} \subset B_\infty(\varepsilon/2). \tag{16}$$

*Proof.* See Appendix A.1. □

---

[1] Actually, the Fourier transform of any real-valued sequence is centrally symmetric: $\widehat{V}(-\boldsymbol{\omega}) = \widehat{V}(\boldsymbol{\omega})^*$. The specificity of well-oriented filters lies in the concentration of their power spectrum around two precise locations.

On the other hand, the following proposition provides a shift invariance bound for low-frequency functions such as introduced above.

**Proposition 1.** *For any $F_0 \in L^2_{\mathbb{R}}(\mathbb{R}^2)$ such that $\operatorname{supp} \widehat{F_0} \subset B_\infty(\varepsilon/2)$, and any $\boldsymbol{h} \in \mathbb{R}^2$,*

$$\|\mathcal{T}_{\boldsymbol{h}} F_0 - F_0\|_{L^2} \le \alpha(\varepsilon \boldsymbol{h}) \|F_0\|_{L^2}, \tag{17}$$

*where we have defined*

$$\alpha : \boldsymbol{\tau} \mapsto \frac{\|\boldsymbol{\tau}\|_1}{2}. \tag{18}$$

*Proof.* See Appendix A.2. $\qquad\square$

### 2.4 Adaptation to Discrete 2D Sequences

Given $\kappa \in ]0, 2\pi]$ and $\boldsymbol{\theta} \in B_\infty(\pi)$, let $\mathrm{W} \in \mathcal{J}(\boldsymbol{\theta}, \kappa)$ denote a discrete Gabor-like filter such as defined in (11). For any image $\mathrm{X} \in l^2_{\mathbb{C}}(\mathbb{Z}^2)$ with finite support and any subsampling factor $m \in \mathbb{N} \setminus \{0\}$, we express $(\mathrm{X} * \overline{\mathrm{W}}) \downarrow m$ using the continuous framework introduced above, and derive an invariance formula.

For any sampling interval $s \in \mathbb{R}_{>0}$, let $\varPhi^{(s)} \in L^2_{\mathbb{R}}(\mathbb{R}^2)$ denote the Shannon scaling function parameterized by $s$, such that

$$\widehat{\varPhi^{(s)}} := s \mathbb{1}_{B_\infty(\pi/s)}. \tag{19}$$

This 2D function is a tensor product of scaled and normalized sinc functions. For any $\boldsymbol{n} \in \mathbb{Z}^2$, we denote by $\varPhi^{(s)}_{\boldsymbol{n}}$ a shifted version of $\varPhi^{(s)}$, satisfying

$$\varPhi^{(s)}_{\boldsymbol{n}}(\boldsymbol{x}) := \varPhi^{(s)}(\boldsymbol{x} - s\boldsymbol{n}). \tag{20}$$

Then, $\{\varPhi^{(s)}_{\boldsymbol{n}}\}_{\boldsymbol{n} \in \mathbb{Z}^2}$ is an orthonormal basis of

$$\mathcal{V}^{(s)} := \{F \in L^2_{\mathbb{C}}(\mathbb{R}^2) \mid \operatorname{supp} \widehat{F} \subset B_\infty(\pi/s)\}. \tag{21}$$

Then, using the notation introduced in (8), we have $\mathcal{V}^{(s)} = \mathcal{V}(\boldsymbol{0}, 2\pi/s)$.

We now consider the following lemma.

**Lemma 2.** *Let $s > 0$. For any $F \in \mathcal{V}^{(s)}$ and any $\boldsymbol{\xi} \in B_\infty(\pi/s)$, we have*

$$\widehat{F}(\boldsymbol{\xi}) = s \widehat{\mathrm{X}}(s\boldsymbol{\xi}), \tag{22}$$

*where $\mathrm{X} \in l^2_{\mathbb{C}}(\mathbb{Z}^2)$ is a uniform sampling of $F$, defined such that $\mathrm{X}[\boldsymbol{n}] := s F(s\boldsymbol{n})$, for any $\boldsymbol{n} \in \mathbb{Z}^2$. Moreover, we have the following norm equality:*

$$\|F\|_{L^2} = \|\mathrm{X}\|_2. \tag{23}$$

*Proof.* See Appendix A.3. $\qquad\square$

We then get the following proposition, which draws a bond between the discrete and continuous frameworks.

**Proposition 2.** *Let $\mathrm{X} \in l^2_{\mathbb{R}}(\mathbb{Z}^2)$ denote an input image with finite support, and $\mathrm{W} \in \mathcal{J}(\boldsymbol{\theta}, \kappa)$. Considering a sampling interval $s \in \mathbb{R}_{>0}$, we define $F_{\mathrm{X}}$ and $\Psi_{\mathrm{W}} \in \mathcal{V}^{(s)}$ such that*

$$F_{\mathrm{X}} := \sum_{\boldsymbol{n} \in \mathbb{Z}^2} \mathrm{X}[\boldsymbol{n}] \, \varPhi^{(s)}_{\boldsymbol{n}} \qquad and \qquad \Psi_{\mathrm{W}} := \sum_{\boldsymbol{n} \in \mathbb{Z}^2} \mathrm{W}[\boldsymbol{n}] \, \varPhi^{(s)}_{\boldsymbol{n}}. \tag{24}$$

*Then,*

$$\Psi_{\mathrm{W}} \in \mathcal{V}(\boldsymbol{\theta}/s, \kappa/s). \tag{25}$$

*Moreover, for all $\boldsymbol{n} \in \mathbb{Z}$,*

$$\mathrm{X}[\boldsymbol{n}] = s F_{\mathrm{X}}(s\boldsymbol{n}); \qquad \mathrm{W}[\boldsymbol{n}] = s \Psi_{\mathrm{W}}(s\boldsymbol{n}), \tag{26}$$

*and, for a given subsampling factor $m \in \mathbb{N} \setminus \{0\}$,*

$$\left( (\mathrm{X} * \overline{\mathrm{W}}) \downarrow m \right) [\boldsymbol{n}] = \left( F_{\mathrm{X}} * \overline{\Psi}_{\mathrm{W}} \right) (ms\boldsymbol{n}) . \tag{27}$$

*Proof.* See Appendix A.4. □

Proposition 2 introduces a latent subspace of $L_{\mathbb{R}}^2(\mathbb{R}^2)$ from which input images are uniformly sampled. This allows us to define, for any $\boldsymbol{u} \in \mathbb{R}^2$, a translation operator $\mathcal{T}_{\boldsymbol{u}}$ on discrete sequences, even if $\boldsymbol{u}$ has non-integer values:

$$\mathcal{T}_{\boldsymbol{u}}\mathrm{X}[\boldsymbol{n}] := s\,\mathcal{T}_{s\boldsymbol{u}}F_{\mathrm{X}}(s\boldsymbol{n}), \tag{28}$$

where $F_{\mathrm{X}}$ is defined in (24). We can indeed show that this definition is independent from the choice of sampling interval $s > 0$. Moreover, given $\mathrm{X} \in l_{\mathbb{R}}^2(\mathbb{Z}^2)$, we have

$$\forall \boldsymbol{p} \in \mathbb{Z}^2, \; \mathcal{T}_{\boldsymbol{p}}\mathrm{X}[\boldsymbol{n}] = \mathrm{X}[\boldsymbol{n} - \boldsymbol{p}]; \tag{29}$$

$$\forall \boldsymbol{u}, \, \boldsymbol{v} \in \mathbb{R}^2, \; \mathcal{T}_{\boldsymbol{u}}(\mathcal{T}_{\boldsymbol{v}}\mathrm{X}) = \mathcal{T}_{\boldsymbol{u}+\boldsymbol{v}}\mathrm{X}, \tag{30}$$

which shows that $\mathcal{T}_{\boldsymbol{u}}$ corresponds to the intuitive idea of a translation operator. Expressions (29) and (30) are direct consequence of the following lemma, which bonds the shift operator in the discrete and continuous frameworks.

**Lemma 3.** *For any $\mathrm{X} \in l_{\mathbb{R}}^2(\mathbb{Z}^2)$ and any $\boldsymbol{u} \in \mathbb{R}^2$,*

$$F_{\mathcal{T}_{\boldsymbol{u}}\mathrm{X}} = \mathcal{T}_{s\boldsymbol{u}}F_{\mathrm{X}}. \tag{31}$$

*Proof.* See Appendix A.5. □

We now consider the following corollary to Proposition 2.

**Corollary 1.** *For any shift vector $\boldsymbol{u} \in \mathbb{R}^2$, we have*

$$\left( (\mathcal{T}_{\boldsymbol{u}}\mathrm{X} * \overline{\mathrm{W}}) \downarrow m \right) [\boldsymbol{n}] = \left( \mathcal{T}_{s\boldsymbol{u}}F_{\mathrm{X}} * \overline{\Psi}_{\mathrm{W}} \right) (ms\boldsymbol{n}) . \tag{32}$$

*Proof.* Applying (27) in Proposition 2 with $\mathrm{X} \leftarrow \mathcal{T}_{\boldsymbol{u}}\mathrm{X}$, we get

$$\left( (\mathcal{T}_{\boldsymbol{u}}\mathrm{X} * \overline{\mathrm{W}}) \downarrow m \right) [\boldsymbol{n}] = \left( F_{\mathcal{T}_{\boldsymbol{u}}\mathrm{X}} * \overline{\Psi}_{\mathrm{W}} \right) (ms\boldsymbol{n}) , \tag{33}$$

and Lemma 3 concludes the proof. □

## 2.5 Shift Invariance in the Discrete Framework

We consider the $\mathbb{C}$Mod operator defined in (5). For the sake of conciseness, in what follows we will write $U_m^{\mathrm{mod}}$ instead of $U_m^{\mathrm{mod}}[\mathrm{W}]$, when no ambiguity is possible. First, we state the following lemma.

**Lemma 4.** *For any input image $\mathrm{X} \in l_{\mathbb{R}}^2(\mathbb{Z}^2)$ with finite support, and any Gabor-like filter $\mathrm{W} \in \mathcal{J}(\boldsymbol{\theta}, \kappa)$, we consider the low-frequency function*

$$F_0 : \boldsymbol{x} \mapsto (F_{\mathrm{X}} * \overline{\Psi}_{\mathrm{W}})(\boldsymbol{x}) \, \mathrm{e}^{i\langle \boldsymbol{\theta}/s, \, \boldsymbol{x} \rangle}, \tag{34}$$

*with $F_{\mathrm{X}}$ and $\Psi_{\mathrm{W}}$ satisfying (24). If $\kappa \le \pi/m$, then*

$$F_0 \in \mathcal{V}^{(s')}. \tag{35}$$

*Moreover, for any $\boldsymbol{h} \in \mathbb{R}^2$,*

$$\sum_{\boldsymbol{n} \in \mathbb{Z}^2} \left| \mathcal{T}_{\boldsymbol{h}}F_0(s'\boldsymbol{n}) - F_0(s'\boldsymbol{n}) \right|^2 = \frac{1}{s'^2} \left\| \mathcal{T}_{\boldsymbol{h}}F_0 - F_0 \right\|_{L^2}^2, \tag{36}$$

*where we have denoted $s' := 2ms$. Finally,*

$$\left\| U_m^{\mathrm{mod}} \mathrm{X} \right\|_2 = \frac{1}{s'} \left\| F_0 \right\|_{L^2}. \tag{37}$$

*Proof.* See Appendix A.6. □

We are now ready to state the main result about shift invariance of $\mathbb{C}$Mod outputs.

**Theorem 1** (Shift invariance of $\mathbb{C}$Mod)**.** *Let* $\mathrm{W} \in \mathcal{J}(\boldsymbol{\theta}, \kappa)$ *denote a discrete Gabor-like filter and* $m \in \mathbb{N} \backslash \{0\}$ *denote a subsampling factor. Then, under the following condition:*

$$\kappa \leq \pi / m, \tag{38}$$

*we have, for any input image* $\mathrm{X} \in l_{\mathbb{R}}^2(\mathbb{Z}^2)$ *with finite support and any translation vector* $\boldsymbol{u} \in \mathbb{R}^2$,

$$\left\| U_m^{\mathrm{mod}}(\mathcal{T}_{\boldsymbol{u}} \mathrm{X}) - U_m^{\mathrm{mod}} \mathrm{X} \right\|_2 \leq \alpha(\kappa \boldsymbol{u}) \left\| U_m^{\mathrm{mod}} \mathrm{X} \right\|_2, \tag{39}$$

*where $\alpha$ has been defined in* (18)*.*

*Proof.* See Appendix A.7. □

Interestingly, the reference value used in Theorem 1, *i.e.*, $\left\| U_m^{\mathrm{mod}} \mathrm{X} \right\|_2$, is fully shift-invariant, as stated in the following proposition.

**Proposition 3.** *Let* $\mathrm{W} \in \mathcal{J}(\boldsymbol{\theta}, \kappa)$ *and* $m \in \mathbb{N} \setminus \{0\}$*. Under condition* (38)*, we have, for any* $\mathrm{X} \in l_{\mathbb{R}}^2(\mathbb{Z}^2)$ *and any* $\boldsymbol{u} \in \mathbb{R}^2$,

$$\left\| U_m^{\mathrm{mod}}(\mathcal{T}_{\boldsymbol{u}} \mathrm{X}) \right\|_2 = \left\| U_m^{\mathrm{mod}} \mathrm{X} \right\|_2. \tag{40}$$

*Proof.* See Appendix A.8. □

## 3 From $\mathbb{C}$Mod to $\mathbb{R}$Max

$\mathbb{C}$Mod operators are found in ScatterNets and complex-valued convolutional networks (Tygert et al., 2016). However, they are absent from conventional, freely-trained CNN architectures. Therefore, Theorem 1 cannot be applied as is. Instead, the first convolution layer contains real-valued kernels, and is generally followed by ReLU and max pooling. As shown in Section 5, this process can be described with $\mathbb{R}$Max operators, such as defined in (1).

As explained in Section 1.1, an important number of trained convolution kernels exhibit oscillating patterns with well-defined frequencies and orientations. To elaborate, let $\mathrm{V} \in l_{\mathbb{R}}^2(\mathbb{Z}^2)$ denote such a trained kernel, and consider $\mathrm{W} \in l_{\mathbb{C}}^2(\mathbb{Z}^2)$ as the complex-valued companion of $\mathrm{V}$ satisfying (12). Then, $\mathrm{W}$ has its energy concentrated in a small region of the Fourier domain. We thus state the hypotheses that $\mathrm{W} \in \mathcal{J}(\boldsymbol{\theta}, \kappa)$ (11) for a certain value of $\boldsymbol{\theta} \in [-\pi, \pi]^2$ and $\kappa \in {]}0, 2\pi]$. For the sake of conciseness, from now on we write $U_{m,q}^{\max}$ instead of $U_{m,q}^{\max}[\mathrm{W}]$, when no ambiguity is possible. In what follows, we establish conditions on $\mathrm{W}$ under which $\mathbb{C}$Mod (5) and $\mathbb{R}$Max (1) operators produce comparable outputs. The final goal, achieved in Section 4, is to provide a shift invariance bound for $\mathbb{R}$Max.

To give an intuition about why $\mathbb{R}$Max may act as a proxy for $\mathbb{C}$Mod, we place ourselves in the continuous framework. Consider the real-valued wavelet transform output $\operatorname{Re} F_1 := F * \operatorname{Re} \overline{\Psi}$, employed in $\mathbb{R}$Max, as the real part of the complex-valued wavelet transform output $F_1 := F * \overline{\Psi}$, used in $\mathbb{C}$Mod. At a given location $\boldsymbol{x} \in \mathbb{R}^2$, the corresponding imaginary part may carry a large amount of information, which somehow needs to be retrieved. The key idea is that, if $\Psi$ is sufficiently localized in the Fourier domain, then only the phase of $F_1$ significantly varies in the vicinity of $\boldsymbol{x}$, whereas its magnitude remains nearly constant. Therefore, finding the maximum value of $\operatorname{Re} F_1$ within a local neighborhood around $\boldsymbol{x}$ is nearly equivalent to shifting

the phase of $F_1(\boldsymbol{x})$ towards 0. The resulting value then approximates $|F_1(\boldsymbol{x})|$. To put it differently, max pooling pushes energy towards lower frequencies, in a similar way as the modulus does for complex-valued transforms (Bruna & Mallat, 2013). This result is hinted in Section 3.1.

Regretfully, things do not work so smoothly in the discrete case. At first glance, this is surprising because Shannon's sampling theorem allows to cast discrete problems into the continuous framework, as done in Section 2.4. However, as explained in Section 3.2, max pooling operates over a discrete grid instead of a continuous window. Consequently, in some situations, the maximum value may fall far away from any zero-phase coefficient. Taking into account this behavior, we adopt a probabilistic point of view, as detailed in Section 3.4. Then, we provide in Section 3.5 an upper bound for the expected gap between $\mathbb{C}$Mod and $\mathbb{R}$Max outputs.

### 3.1  Continuous Framework

This section, inspired from Waldspurger (2015, pp. 190–191), provides an intuition about resemblance between $\mathbb{R}$Max and $\mathbb{C}$Mod in the continuous framework. As will be highlighted in Section 3.2, adaptation to discrete 2D sequences is not straightforward and will require a probabilistic approach.

We consider an input function $F \in L^2_{\mathbb{R}}(\mathbb{R}^2)$ and a band-pass filter $\Psi \in \mathcal{V}(\boldsymbol{\nu}, \varepsilon)$. Let us also consider

$$G : (\boldsymbol{x}, \boldsymbol{h}) \mapsto \cos\big(\langle \boldsymbol{\nu}, \boldsymbol{h}\rangle - H(\boldsymbol{x})\big), \tag{41}$$

where $H : \mathbb{R}^2 \to [0, 2\pi[$ denotes the phase of $F * \overline{\Psi}$. Lemma 1 introduced low-frequency functions $F_0$, with slow variations. In a nutshell, since $\operatorname{supp} F_0 \subset B_\infty(\varepsilon/2)$, we can write

$$\|\boldsymbol{h}\|_2 \ll \lambda_{F_0} \implies F_0(\boldsymbol{x} + \boldsymbol{h}) \approx F_0(\boldsymbol{x}), \tag{42}$$

where we have defined $\lambda_{F_0} := 2\pi/\varepsilon$. Therefore, according to Proposition 4 below, we get the following approximation of $F * \operatorname{Re}\overline{\Psi}$ in a neighborhood around any point $\boldsymbol{x} \in \mathbb{R}^2$:

$$\|\boldsymbol{h}\|_2 \ll \lambda_{F_0} \implies (F * \operatorname{Re}\overline{\Psi})(\boldsymbol{x} + \boldsymbol{h}) \approx \big|(F * \overline{\Psi})(\boldsymbol{x})\big|\, G(\boldsymbol{x}, \boldsymbol{h}). \tag{43}$$

**Proposition 4.** *For any $\boldsymbol{h} \in \mathbb{R}^2$,*

$$\Big|(F * \operatorname{Re}\overline{\Psi})(\boldsymbol{x} + \boldsymbol{h}) - \big|(F * \overline{\Psi})(\boldsymbol{x})\big|\, G(\boldsymbol{x}, \boldsymbol{h})\Big| \le \big|F_0(\boldsymbol{x} + \boldsymbol{h}) - F_0(\boldsymbol{x})\big|. \tag{44}$$

*Proof.* See Appendix A.9. □

On the one hand, we consider a continuous equivalent of the $\mathbb{C}$Mod operator $U^{\mathrm{mod}}_m[\mathrm{W}]$ as introduced in (5). Such an operator, denoted by $U^{\mathrm{mod}}[\Psi]$, is defined, for any $F \in L^2_{\mathbb{R}}(\mathbb{R}^2)$, by

$$U^{\mathrm{mod}}[\Psi]\,(F) : \boldsymbol{x} \mapsto \big|(F * \overline{\Psi})(\boldsymbol{x})\big|. \tag{45}$$

On the other hand, we consider the continuous counterpart of $\mathbb{R}$Max as introduced in (1). It is defined as the maximum value of $F * \operatorname{Re}\overline{\Psi}$ over a sliding spatial window of size $r > 0$. This is possible because $F$ and $\operatorname{Re}\overline{\Psi}$ both belong to $L^2_{\mathbb{R}}(\mathbb{R}^2)$, and therefore $F * \operatorname{Re}\overline{\Psi}$ is continuous. Such an operator, denoted by $U^{\max}_r[\Psi]$, is defined, for any $F \in L^2_{\mathbb{R}}(\mathbb{R}^2)$, by

$$U^{\max}_r[\Psi]\,(F) : \boldsymbol{x} \mapsto \max_{\|\boldsymbol{h}\|_\infty \le r} (F * \operatorname{Re}\overline{\Psi})(\boldsymbol{x} + \boldsymbol{h}). \tag{46}$$

For the sake of conciseness, the parameter between square brackets is ignored from now on. If $r \ll \lambda_{F_0}$, then (43) is valid for any $\boldsymbol{h} \in B_\infty(r)$. Then, using (45) and (46), we get

$$r \ll \lambda_{F_0} \implies U^{\max}_r F(\boldsymbol{x}) \approx U^{\mathrm{mod}} F(\boldsymbol{x}) \max_{\|\boldsymbol{h}\|_\infty \le r} G(\boldsymbol{x}, \boldsymbol{h}). \tag{47}$$

Using the periodicity of $G$, we can show that, if $r \ge \frac{\pi}{\|\boldsymbol{\nu}\|_2}$, then $\boldsymbol{h} \mapsto G(\boldsymbol{x}, \boldsymbol{h})$ necessarily reaches its maximum value (*i.e.*, 1) on $B_\infty(r)$. We therefore get

$$\frac{\pi}{\|\boldsymbol{\nu}\|_2} \le r \ll \frac{2\pi}{\varepsilon} \implies U^{\max}_r F(\boldsymbol{x}) \approx U^{\mathrm{mod}} F(\boldsymbol{x}). \tag{48}$$

## 3.2 Adaptation to Discrete 2D Sequences

As in Section 2.4, we consider an input image $X \in l^2_{\mathbb{R}}(\mathbb{Z}^2)$, a complex, analytic convolution kernel $W \in \mathcal{J}(\boldsymbol{\theta}, \kappa)$, a subsampling factor $m \in \mathbb{N} \setminus \{0\}$ and an integer $q \in \mathbb{N} \setminus \{0\}$, referred to as a *half-size*, such that max pooling operates on a grid of size $(2q + 1) \times (2q + 1)$. We seek a relationship between

$$Y^{\mathrm{max}} := U^{\mathrm{max}}_{m, q}[W](X) \qquad \text{and} \qquad Y^{\mathrm{mod}} := U^{\mathrm{mod}}_m[W](X), \tag{49}$$

where $U^{\mathrm{max}}_{m, q}[W]$ (ℝMax) and $U^{\mathrm{mod}}_m[W]$ (ℂMod) have been respectively defined in (1) and (5). As before, in what follows we omit the parameter between square brackets.

We now use the sampling results from Proposition 2. Let $F_X$ and $\Psi_W \in \mathcal{V}^{(s)}$ denote the functions satisfying (24). Recall that the continuous versions of ℂMod and ℝMax operators have been defined in (45) and (46), respectively. On the one hand, we apply (27) with $m \leftarrow 2m$ to $Y^{\mathrm{mod}}$. For any $\boldsymbol{n} \in \mathbb{Z}^2$,

$$U^{\mathrm{mod}}_m X[\boldsymbol{n}] = (F_X * \overline{\Psi}_W)(\boldsymbol{x_n}) \tag{50}$$

$$= U^{\mathrm{mod}} F_X(\boldsymbol{x_n}), \tag{51}$$

with $\boldsymbol{x_n} := 2ms\boldsymbol{n}$. On the other hand, we postulate that

$$U^{\mathrm{max}}_{m, q} X[\boldsymbol{n}] = U^{\mathrm{max}}_r F_X(\boldsymbol{x_n}) \tag{52}$$

for a certain value of $r \in \mathbb{R}_{>0}$. Then, (48) implies $Y^{\mathrm{mod}} \approx Y^{\mathrm{max}}$. However, as explained hereafter, (52) is not satisfied, due to the discrete nature of the max pooling grid. According to (1) and (4), we have

$$U^{\mathrm{max}}_{m, q} X[\boldsymbol{n}] = \max_{\|\boldsymbol{p}\|_\infty \leq q} \mathrm{Re}\left((X * \overline{W}) \downarrow m\right)[2\boldsymbol{n} + \boldsymbol{p}]. \tag{53}$$

Therefore, according to (27) in Proposition 2, we get

$$U^{\mathrm{max}}_{m, q} X[\boldsymbol{n}] = \max_{\|\boldsymbol{p}\|_\infty \leq q} (F_X * \mathrm{Re}\,\overline{\Psi}_W)(\boldsymbol{x_n} + \boldsymbol{h_p}), \tag{54}$$

with

$$\boldsymbol{x_n} := 2ms\boldsymbol{n} \qquad \text{and} \qquad \boldsymbol{h_p} := ms\boldsymbol{p}. \tag{55}$$

By considering $r_q := ms\left(q + \frac{1}{2}\right)$, we get a variant of (52) in which the maximum is evaluated on a discrete grid of $(2q + 1)^2$ elements, instead of the continuous region $B_\infty(r_q)$, as defined in (46) with $r \leftarrow r_q$. As a consequence, (47) is replaced in the discrete framework by

$$q \ll 2\pi/(m\kappa) \quad \Longrightarrow \quad U^{\mathrm{max}}_{m, q} X[\boldsymbol{n}] \approx U^{\mathrm{mod}}_m X[\boldsymbol{n}] \max_{\|\boldsymbol{p}\|_\infty \leq q} G_X(\boldsymbol{x_n}, \boldsymbol{h_p}), \tag{56}$$

where we have introduced, similarly to (41),

$$G_X : (\boldsymbol{x}, \boldsymbol{h}) \mapsto \cos(\langle \boldsymbol{\nu}, \boldsymbol{h} \rangle - H_X(\boldsymbol{x})), \tag{57}$$

with

$$\boldsymbol{\nu} := \boldsymbol{\theta}/s \qquad \text{and} \qquad H_X := \angle\left(F_X * \overline{\Psi}_W\right), \tag{58}$$

where $\angle : \mathbb{C} \to [0, 2\pi[$ denotes the phase operator. Unlike the continuous case, even if the window size $r_q$ is large enough, the existence of $\boldsymbol{p} \in \{-q, \dots, q\}^2$ such that $G_X(\boldsymbol{x_n}, \boldsymbol{h_p}) = 1$ is not guaranteed, as illustrated in Figure 3 with $q = 1$. Instead, we can only seek a probabilistic estimation of the normalized mean squared error between $Y^{\mathrm{max}}$ and $Y^{\mathrm{mod}}$.

Approximation (56) implies

$$q \ll 2\pi/(m\kappa) \implies \left\|U^{\mathrm{mod}}_m X - U^{\mathrm{max}}_{m, q} X\right\|_2 \approx \|\delta_{m, q} X\|_2, \tag{59}$$

where $\delta_{m, q} X \in l^2_{\mathbb{R}}(\mathbb{Z}^2)$ is defined such that, for any $\boldsymbol{n} \in \mathbb{Z}^2$,

$$\delta_{m, q} X[\boldsymbol{n}] := U^{\mathrm{mod}}_m X[\boldsymbol{n}]\left(1 - \max_{\|\boldsymbol{p}\|_\infty \leq q} G_X(\boldsymbol{x_n}, \boldsymbol{h_p})\right). \tag{60}$$

Expression (59) suggests that the difference between the left and right terms can be bounded by a quantity which only depends on the product $m\kappa$ (subsampling factor × frequency localization) and the grid half-size $q$. In what follows, we establish a bound characterizing this approximation, which will be provided in Proposition 5.

For the sake of conciseness, we introduce the following notations:

$$A_X : (\boldsymbol{x}, \boldsymbol{h}) \mapsto (F_X * \operatorname{Re} \overline{\varPsi}_W)(\boldsymbol{x} + \boldsymbol{h}); \tag{61}$$

$$\widetilde{A}_X : (\boldsymbol{x}, \boldsymbol{h}) \mapsto \left|(F_X * \overline{\varPsi}_W)(\boldsymbol{x})\right| G_X(\boldsymbol{x}, \boldsymbol{h}). \tag{62}$$

We now consider, for any $\boldsymbol{n} \in \mathbb{Z}^2$, the vectors $\boldsymbol{h}_{\boldsymbol{n}}^{\max}$ and $\boldsymbol{h'}_{\boldsymbol{n}}^{\max} \in ms\{-q, \ldots, q\}^2$ achieving the maximum value of $A_X(\boldsymbol{x_n}, \boldsymbol{h_p})$ and $\widetilde{A}_X(\boldsymbol{x_n}, \boldsymbol{h_p})$ over the max pooling grid, respectively. They satisfy

$$A_X^{\max}(\boldsymbol{x_n}) := A_X\big(\boldsymbol{x_n}, \boldsymbol{h}_{\boldsymbol{n}}^{\max}\big) = \max_{\|\boldsymbol{p}\|_\infty \leq q} A_X(\boldsymbol{x_n}, \boldsymbol{h_p}); \tag{63}$$

$$\widetilde{A}_X^{\max}(\boldsymbol{x_n}) := \widetilde{A}_X\big(\boldsymbol{x_n}, \boldsymbol{h'}_{\boldsymbol{n}}^{\max}\big) = \max_{\|\boldsymbol{p}\|_\infty \leq q} \widetilde{A}_X(\boldsymbol{x_n}, \boldsymbol{h_p}). \tag{64}$$

Then, according to (50) and (54), we get, for any $\boldsymbol{n} \in \mathbb{Z}^2$,

$$A_X^{\max}(\boldsymbol{x_n}) = U_{m,q}^{\max} X[\boldsymbol{n}]; \tag{65}$$

$$\widetilde{A}_X^{\max}(\boldsymbol{x_n}) = U_m^{\mathrm{mod}} X[\boldsymbol{n}] \max_{\|\boldsymbol{p}\|_\infty \leq q} G_X\big(\boldsymbol{x_n}, \boldsymbol{h_p}\big), \tag{66}$$

and (56) becomes

$$q \ll 2\pi/(m\kappa) \quad \Longrightarrow \quad A_X^{\max}(\boldsymbol{x_n}) \approx \widetilde{A}_X^{\max}(\boldsymbol{x_n}). \tag{67}$$

**Remark 3.** Expression (43) implies that, if $q \ll 2\pi/(m\kappa)$, then $A_X(\boldsymbol{x_n}, \boldsymbol{h_p}) \approx \widetilde{A}_X(\boldsymbol{x_n}, \boldsymbol{h_p})$ for all $\boldsymbol{p} \in \{-q, \ldots, q\}^2$. However, this property does not guarantee that $A_X$ and $\widetilde{A}_X$ reach their maximum in the same exact location; i.e., that $\boldsymbol{h}_{\boldsymbol{n}}^{\max} = \boldsymbol{h'}_{\boldsymbol{n}}^{\max}$.

The following lemma provides a bound for approximation (67).

**Lemma 5.** *For any $\boldsymbol{x} \in \mathbb{R}^2$,*

$$\left|A_X^{\max}(\boldsymbol{x_n}) - \widetilde{A}_X^{\max}(\boldsymbol{x_n})\right| \leq \max_{\boldsymbol{h} \in \{\boldsymbol{h}_{\boldsymbol{n}}^{\max}, \boldsymbol{h'}_{\boldsymbol{n}}^{\max}\}} \left|F_0(\boldsymbol{x_n} + \boldsymbol{h}) - F_0(\boldsymbol{x_n})\right|. \tag{68}$$

*Proof.* See Appendix A.10. □

Before stating Proposition 5, we consider the following hypothesis:

**Hypothesis 1.** There exists $\boldsymbol{h}_0 \in \mathbb{R}^2$ with $\|\boldsymbol{h}_0\|_2 = \sqrt{2}qms$, such that

$$\sum_{\boldsymbol{n} \in \mathbb{Z}^2} \max_{\boldsymbol{h} \in \{\boldsymbol{h}_{\boldsymbol{n}}^{\max}, \boldsymbol{h'}_{\boldsymbol{n}}^{\max}\}} \left|F_0(\boldsymbol{x_n} + \boldsymbol{h}) - F_0(\boldsymbol{x_n})\right|^2 \leq \sum_{\boldsymbol{n} \in \mathbb{Z}^2} \left|F_0(\boldsymbol{x_n} + \boldsymbol{h}_0) - F_0(\boldsymbol{x_n})\right|^2. \tag{69}$$

The underlying idea is explained as follows. The absolute difference between $F_0(\boldsymbol{x_n} + \boldsymbol{h})$ and $F_0(\boldsymbol{x_n})$ is more likely to increase with the norm of $\boldsymbol{h}$. For any given $\boldsymbol{n} \in \mathbb{Z}^2$, we have, by construction, $\|\boldsymbol{h}_{\boldsymbol{n}}^{\max}\|_2 \leq \sqrt{2}qms$ and $\|\boldsymbol{h'}_{\boldsymbol{n}}^{\max}\|_2 \leq \sqrt{2}qms$. Therefore, we can expect to observe

$$\max_{\boldsymbol{h} \in \{\boldsymbol{h}_{\boldsymbol{n}}^{\max}, \boldsymbol{h'}_{\boldsymbol{n}}^{\max}\}} \left|F_0(\boldsymbol{x_n} + \boldsymbol{h}) - F_0(\boldsymbol{x_n})\right|^2 \leq \left|F_0(\boldsymbol{x_n} + \boldsymbol{h}_0) - F_0(\boldsymbol{x_n})\right|^2. \tag{70}$$

While this might occasionally not be true, Hypothesis 1 postulates that, when summing over all the data-points, the inequality holds.

We now formally state the result characterizing approximation (59).

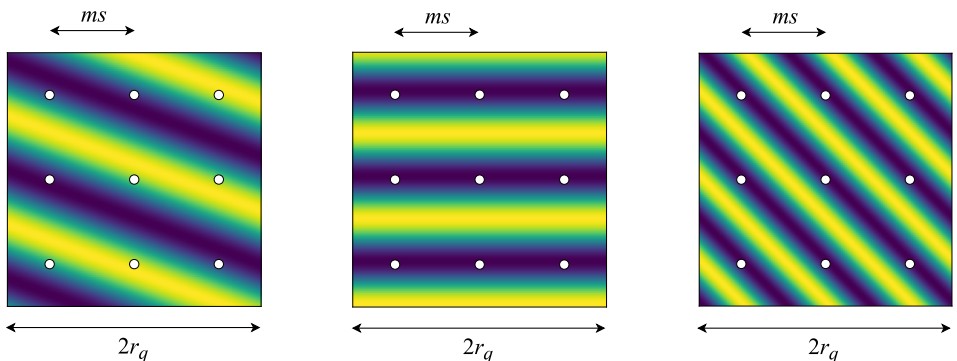

Figure 3. Search for the maximum value of $\boldsymbol{h} \mapsto G_{\mathrm{X}}(\boldsymbol{x}, \boldsymbol{h})$ over a discrete grid of size $3 \times 3$, *i.e.*, $q = 1$. This figure displays 3 examples with different frequencies $\boldsymbol{\nu} := \boldsymbol{\theta}/s$ and phases $H_{\mathrm{X}}(\boldsymbol{x})$. Hopefully the result will be close to the true maximum (left), but there are some pathological cases in which all points in the grid fall into pits (middle and right).

**Proposition 5.** *We assume that condition* (38) *is satisfied:* $\kappa \leq \pi/m$. *Then, under Hypothesis 1,*

$$\left\| U_m^{\mathrm{mod}} \mathrm{X} - U_{m,q}^{\max} \mathrm{X} \right\|_2 \leq \left\| \delta_{m,q} \mathrm{X} \right\|_2 + \beta_q(m\kappa) \left\| U_m^{\mathrm{mod}} \mathrm{X} \right\|_2, \tag{71}$$

*where* $\beta_q : \mathbb{R}_+ \to \mathbb{R}_+$ *is defined by*

$$\beta_q : \kappa' \mapsto q\kappa'. \tag{72}$$

*Proof.* See Appendix A.11. □

We now seek a probabilistic estimation of $\left\| \delta_{m,q} \mathrm{X} \right\|_2$. For this purpose, we first reformulate the problem using the unit circle $\mathbb{S}^1 \subset \mathbb{C}$, before introducing a probabilistic framework in Section 3.4.

### 3.3 Notations on the Unit Circle

In what follows, for any $z \in \mathbb{C} \setminus \{0\}$, we denote by $\angle z \in [0, 2\pi[$ the argument of $z$. For any $z$, $z' \in \mathbb{S}^1$, the angle between $z$ and $z'$ is given by $\angle(z^* z')$. We then denote by $[z, z']_{\mathbb{S}^1} \subset \mathbb{S}^1$ the arc on the unit circle going from $z$ to $z'$ counterclockwise:

$$[z, z']_{\mathbb{S}^1} := \left\{ z'' \in \mathbb{S}^1 \mid \angle(z^* z'') \leq \angle(z^* z') \right\}. \tag{73}$$

We remind readers that $\boldsymbol{x_n}$ and $\boldsymbol{h_p} \in \mathbb{R}^2$ have been defined in (55). By using the relation $\cos \alpha = \mathrm{Re}(e^{i\alpha})$, (57) becomes, for any $\boldsymbol{n} \in \mathbb{Z}^2$ and any $\boldsymbol{p} \in \{-q, \ldots, q\}^2$,

$$G_{\mathrm{X}}(\boldsymbol{x_n}, \boldsymbol{h_p}) = \mathrm{Re}\left( Z_{\mathrm{X}}^*(\boldsymbol{x_n}) Z_{\boldsymbol{p}}(m\boldsymbol{\theta}) \right), \tag{74}$$

where we have defined the following functions with outputs on the unit circle:

$$Z_{\mathrm{X}} : \boldsymbol{x} \mapsto e^{i H_{\mathrm{X}}(\boldsymbol{x})} \qquad \text{and} \qquad Z_{\boldsymbol{p}} : \boldsymbol{\omega} \mapsto= e^{i \langle \boldsymbol{\omega}, \boldsymbol{p} \rangle}, \tag{75}$$

where $H_{\mathrm{X}}$ denotes the phase of $F_{\mathrm{X}} * \overline{\Psi}_{\mathrm{W}}$ as introduced in (58). On the one hand, $Z_{\mathrm{X}}(\boldsymbol{x_n})$ is the phase (represented on the unit circle $\mathbb{S}^1$) of the complex wavelet transform $F_{\mathrm{X}} * \overline{\Psi}_{\mathrm{W}}$ at location $\boldsymbol{x_n}$. On the other hand, $Z_{\boldsymbol{p}}(m\boldsymbol{\theta})$ approximates the phase shift between any two evaluations of $F_{\mathrm{X}} * \overline{\Psi}_{\mathrm{W}}$ at locations $\boldsymbol{x}$, $\boldsymbol{x}'$ such that $\boldsymbol{x}' - \boldsymbol{x} = \boldsymbol{h_p}$. This however is only true if we assume that $\Psi_{\mathrm{W}}$ exhibits slow amplitude variations. Then, $G_{\mathrm{X}}(\boldsymbol{x_n}, \boldsymbol{h_p})$ approximates the cosine of the phase of $F_{\mathrm{X}} * \overline{\Psi}_{\mathrm{W}}$ at location $\boldsymbol{x_n} + \boldsymbol{h_p}$.

According to (56), $\max_{\|\boldsymbol{p}\|_\infty \leq q} G_{\mathrm{X}}(\boldsymbol{x_n}, \boldsymbol{h_p})$ approximates the ratio between $\mathbb{R}\mathrm{Max}$ and $\mathbb{C}\mathrm{Mod}$ outputs at discrete location $\boldsymbol{n} \in \mathbb{Z}^2$. The intuition behind this is that max pooling seeks a point in a discrete grid

around $\boldsymbol{x_n}$ where the phase of $F_{\mathrm{X}} * \overline{\Psi}_{\mathrm{W}}$ is the closest to 1, thereby maximizing the amount of energy on the real part of the signal. Assuming slow amplitude variations of $\Psi_{\mathrm{W}}$, the result therefore approximates the modulus of the complex coefficients.

To get an estimation of $\delta_{m,q}\mathrm{X}[\boldsymbol{n}]$ (60), we will exploit the following property. If the phases $Z_{\boldsymbol{p}}(m\boldsymbol{\theta})$ for $\boldsymbol{p} \in \{-q, \ldots, q\}^2$ are well distributed on the unit circle, then the values of $G_{\mathrm{X}}(\boldsymbol{x_n}, \boldsymbol{h_p})$ are evenly spread out on $[-1, 1]$. Therefore, its maximum value is more likely to be close to 1, and (60) becomes

$$\delta_{m,q}\mathrm{X}[\boldsymbol{n}] \ll U_m^{\mathrm{mod}}\mathrm{X}[\boldsymbol{n}] \qquad \forall \boldsymbol{n} \in \mathbb{Z}^2. \tag{76}$$

Let $n_q := (2q+1)^2$ denote the number of evaluation points for the max pooling operator. For any $\boldsymbol{\omega} \in \mathbb{R}^2$, we consider a sequence of values on $\mathbb{S}^1$, denoted by $\left(Z_i^{(q)}(\boldsymbol{\omega})\right)_{i \in \{0, \ldots, n_q-1\}}$, obtained by sorting $\{Z_{\boldsymbol{p}}(\boldsymbol{\omega})\}_{\boldsymbol{p} \in \{-q, \ldots, q\}^2}$ (75) in ascending order of their arguments:

$$0 = H_0^{(q)}(\boldsymbol{\omega}) \leq \cdots \leq H_{n_q-1}^{(q)}(\boldsymbol{\omega}) < 2\pi, \tag{77}$$

where $H_i^{(q)}(\boldsymbol{\omega})$ denotes the phase of $Z_i^{(q)}(\boldsymbol{\omega})$. In addition, we close the loop with $H_{n_q}^{(q)}(\boldsymbol{\omega}) := 2\pi$ and $Z_{n_q}^{(q)}(\boldsymbol{\omega}) := 1$. Then, we split $\mathbb{S}^1$ into $n_q$ arcs delimited by $Z_i^{(q)}(\boldsymbol{\omega})$:

$$\mathfrak{A}_i^{(q)}(\boldsymbol{\omega}) := \begin{cases} \left[Z_i^{(q)}(\boldsymbol{\omega}), \, Z_{i+1}^{(q)}(\boldsymbol{\omega})\right]_{\mathbb{S}^1} & \text{if } H_{i+1}^{(q)}(\boldsymbol{\omega}) - H_i^{(q)}(\boldsymbol{\omega}) < 2\pi; \\ \mathbb{S}^1 & \text{otherwise.} \end{cases} \tag{78}$$

Finally, for any $i \in \{0, \ldots, n_q - 1\}$, we denote by

$$\delta H_i^{(q)} : \boldsymbol{\omega} \mapsto H_{i+1}^{(q)}(\boldsymbol{\omega}) - H_i^{(q)}(\boldsymbol{\omega}) \tag{79}$$

the function computing the angular measure of arc $\mathfrak{A}_i^{(q)}(\boldsymbol{\omega})$, for any $\boldsymbol{\omega} \in \mathbb{R}^2$.

### 3.4 Probabilistic Framework

From now on, input X is considered as a discrete 2D stochastic process. In order to "randomize" $F_{\mathrm{X}}$ introduced in (24), we define a continuous stochastic process from X, denoted by $\mathsf{F}_{\mathrm{X}}$, such that

$$\forall \boldsymbol{x} \in \mathbb{R}^2, \; \mathsf{F}_{\mathrm{X}}(\boldsymbol{x}) := \sum_{\boldsymbol{n} \in \mathbb{Z}^2} \mathrm{X}[\boldsymbol{n}] \, \Phi_{\boldsymbol{n}}^{(s)}(\boldsymbol{x}). \tag{80}$$

Next, we consider the following stochastic processes, which are parameterized by X:

$$\mathsf{M}_{\mathrm{X}} := |\mathsf{F}_{\mathrm{X}} * \overline{\Psi}_{\mathrm{W}}|; \qquad \mathsf{H}_{\mathrm{X}} := \angle(\mathsf{F}_{\mathrm{X}} * \overline{\Psi}_{\mathrm{W}}); \qquad \mathsf{Z}_{\mathrm{X}} := \mathrm{e}^{i\mathsf{H}_{\mathrm{X}}}, \tag{81}$$

and, for any $\boldsymbol{p} \in \{-q, \ldots, q\}^2$,

$$\mathsf{G}_{\mathrm{X}, \boldsymbol{p}} := \mathrm{Re}\left(\mathsf{Z}_{\mathrm{X}}^* \, Z_{\boldsymbol{p}}(m\boldsymbol{\theta})\right); \qquad \mathsf{G}_{\mathrm{X}}^{\mathrm{max}} := \max_{\|\boldsymbol{p}\|_\infty \leq q} \mathsf{G}_{\mathrm{X}, \boldsymbol{p}}, \tag{82}$$

where the deterministic function $Z_{\boldsymbol{p}}$ has been defined in (75).

**Remark 4.** By continuous extension, $\mathsf{H}_{\mathrm{X}}(\boldsymbol{x})$ and $\mathsf{Z}_{\mathrm{X}}(\boldsymbol{x})$ are uniquely defined at $\boldsymbol{x}$ such that $\mathsf{M}_{\mathrm{X}}(\boldsymbol{x}) = 0$.

For any $\boldsymbol{x} \in \mathbb{R}^2$, $F_{\mathrm{X}}(\boldsymbol{x})$ (24) and $H_{\mathrm{X}}(\boldsymbol{x})$ (58) are respectively drawn from $\mathsf{F}_{\mathrm{X}}(\boldsymbol{x})$ and $\mathsf{H}_{\mathrm{X}}(\boldsymbol{x})$. Then, $Z_{\mathrm{X}}(\boldsymbol{x})$ (75) is a realization of $\mathsf{Z}_{\mathrm{X}}(\boldsymbol{x})$. Consequently, according to (74), $G_{\mathrm{X}}(\boldsymbol{x}, \boldsymbol{h_p})$ is a realization of $\mathsf{G}_{\mathrm{X}, \boldsymbol{p}}(\boldsymbol{x})$. Furthermore, according to the definition of $\mathbb{C}\mathrm{Mod}$ in (5) and $\boldsymbol{x_n}$ in (55), Proposition 2 with $m \leftarrow 2m$ implies that

$$\mathsf{M}_{\mathrm{X}}(\boldsymbol{x_n}) = U_m^{\mathrm{mod}}\mathrm{X}[\boldsymbol{n}]. \tag{83}$$

We remind that $\boldsymbol{\theta} \in [-\pi, \pi]^2$ and $\kappa \in ]0, 2\pi]$ respectively denote the center and size of the Fourier support of the complex kernel $W \in \mathcal{J}(\boldsymbol{\theta}, \kappa)$. To compute the expected discrepancy between $Y^{\max}$ and $Y^{\mod}$ such as introduced in (49), we assume that

$$\|\boldsymbol{\theta}\|_2 \gg 2\pi/N; \tag{84}$$

$$\|\boldsymbol{\theta}\|_2 \gg \kappa, \tag{85}$$

where $N \in \mathbb{N} \setminus \{0\}$ denotes the support size of input images. These assumptions exclude low-frequency filters from the scope of our study. We then state the following hypotheses, for which a justification is provided in Appendix B.

**Hypothesis 2.** For any $\boldsymbol{x} \in \mathbb{R}^2$, $Z_X(\boldsymbol{x})$ is uniformly distributed on $\mathbb{S}^1$.

**Hypothesis 3.** For any $n \in \mathbb{N} \setminus \{0\}$ and $\boldsymbol{x}, \boldsymbol{y}_0, \ldots, \boldsymbol{y}_{n-1} \in \mathbb{R}^2$, the random variables $M_X(\boldsymbol{y}_i)$ for $i \in \{0, \ldots, n-1\}$ are jointly independent of $Z_X(\boldsymbol{x})$.

### 3.5 Expected Quadratic Error between $\mathbb{R}\text{Max}$ and $\mathbb{C}\text{Mod}$

In this section, we propose to estimate the expected value of the stochastic quadratic error $\widetilde{P}_X^2$, defined such that

$$\widetilde{P}_X := \left\| U_m^{\mod} X - U_{m,q}^{\max} X \right\|_2 / \left\| U_m^{\mod} X \right\|_2. \tag{86}$$

According to (49), this is an estimation of the relative error between $Y^{\mod}$ and $Y^{\max}$.

First, let us reformulate $\delta_{m,q} X$, introduced in (60), using the probabilistic framework. According to (74) and (82), we have, for any $\boldsymbol{n} \in \mathbb{Z}^2$,

$$\delta_{m,q} X[\boldsymbol{n}] := U_m^{\mod} X[\boldsymbol{n}] \left( 1 - G_X^{\max}(\boldsymbol{x_n}) \right). \tag{87}$$

We now consider the stochastic process

$$Q_X := 1 - G_X^{\max}, \tag{88}$$

and the random variable

$$\widetilde{Q}_X := \left\| \delta_{m,q} X \right\|_2 / \left\| U_m^{\mod} X \right\|_2. \tag{89}$$

The next steps are as follows: (1) at the pixel level, show that $\mathbb{E}[Q_X(\boldsymbol{x})^2]$ depends on the subsampling factor $m$ and the filter frequency $\boldsymbol{\theta}$, and remains close to zero with some exceptions; (2) at the image level, show that the expected value of $\widetilde{Q}_X^2$ is equal to the latter quantity; (3) use Proposition 5, which implies that $\widetilde{P}_X \approx \widetilde{Q}_X$, to deduce an upper bound on the expected value of $\widetilde{P}_X^2$.

The first point, which is established in Proposition 6 below, is a key result of this paper. It will be used to prove Theorem 2, which corresponds to the two remaining points.

**Proposition 6.** *Assuming Hypothesis 2, the expected value of $Q_X(\boldsymbol{x})^2$ is independent from the choice of $\boldsymbol{x} \in \mathbb{R}^2$, and*

$$\mathbb{E}\left[ Q_X(\boldsymbol{x})^2 \right] = \gamma_q(m\boldsymbol{\theta})^2, \tag{90}$$

*where we have defined*

$$\gamma_q : \boldsymbol{\omega} \mapsto \sqrt{\frac{3}{2} + \frac{1}{4\pi} \sum_{i=0}^{n_q-1} \left( \sin \delta H_i^{(q)}(\boldsymbol{\omega}) - 8 \sin \frac{\delta H_i^{(q)}(\boldsymbol{\omega})}{2} \right)}, \tag{91}$$

*with $\delta H_i^{(q)}(\boldsymbol{\omega}) \in [0, 2\pi]$ (79) being the length of arc $\mathfrak{A}_i^{(q)}(\boldsymbol{\omega})$.*

*Proof.* For the sake of readability, in this proof we omit the argument of functions $Z_{\boldsymbol{p}}$ (75), $Z_i^{(q)}$, $H_i^{(q)}$ (77), $\mathfrak{A}_i^{(q)}$ (78), and $\delta H_i^{(q)}$ (79); we assume they are evaluated at $\boldsymbol{\omega} \leftarrow m\boldsymbol{\theta}$. We consider the "Lebesgue" Borel $\sigma$-algebra on $\mathbb{S}^1$ generated by $\{[z, z']_{\mathbb{S}^1} \mid z, z' \in \mathbb{S}^1\} \cup \{\mathbb{S}^1\}$, on which we have defined the angular measure $\vartheta$ such that $\vartheta(\mathbb{S}^1) := 2\pi$, and

$$\forall z, z' \in \mathbb{S}^1, \vartheta\left([z, z']_{\mathbb{S}^1}\right) := \angle(z^* z'). \tag{92}$$

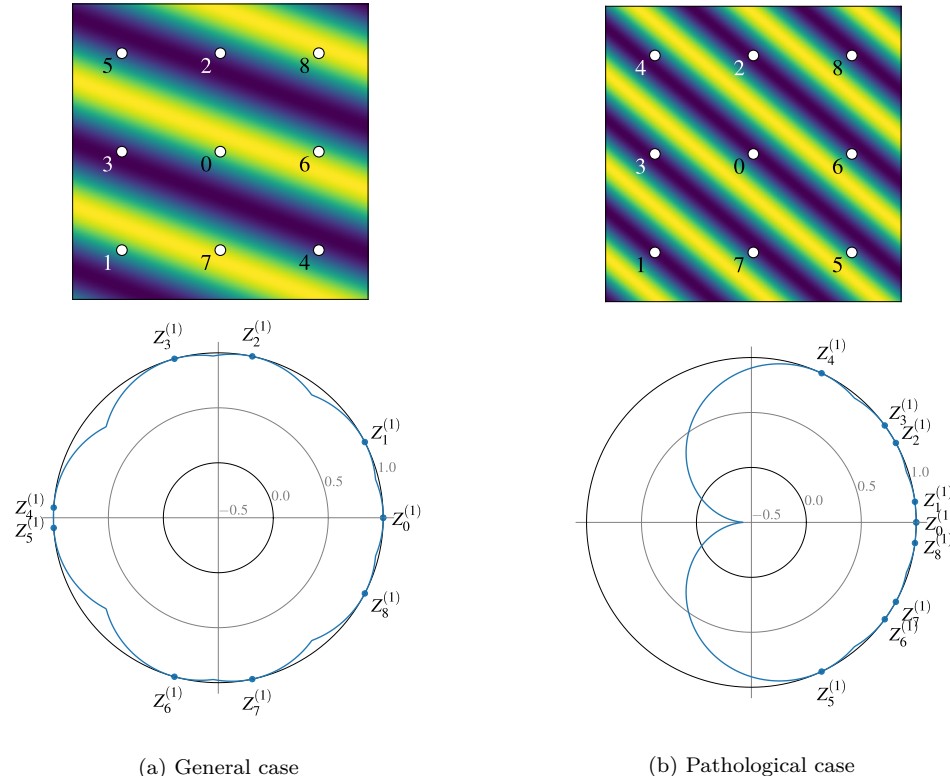

(a) General case                    (b) Pathological case

Figure 4. Top: 2D representation of $\boldsymbol{h} \mapsto G_{\mathrm{X}}(\boldsymbol{x_n}, \boldsymbol{h})$ (57), for two different values of $\boldsymbol{\theta} \in \mathbb{R}^2$, $q = 1$ and arbitrary values of $m \in \mathbb{N} \setminus \{0\}$ and $s \in \mathbb{R} \setminus \{0\}$. Assuming the plots are centered around $\boldsymbol{h} = \boldsymbol{0}$, each point materializes a location $\boldsymbol{h_p}$ in the max pooling grid, for $\boldsymbol{p} \in \{-q, \ldots, q\}^2$. The desirable situation occurs when one of these locations falls near a ridge (bright areas), in which case the outputs produced by $\mathbb{R}$Max and $\mathbb{C}$Mod are similar—see (56). Each number $i \in \{0, \ldots, 8\}$ represents the rank of $Z_{\boldsymbol{p}} \in \mathbb{S}^1$ (75), when these values are sorted by ascending order of their arguments (77). If location $\boldsymbol{h_p}$ gets ranked $i$, then we have $Z_{\boldsymbol{p}} = Z_i^{(q)}$. Bottom: polar representations of $g_{\max} : \mathbb{S}^1 \to [-1, 1]$ (93), corresponding to the same settings. The closer the curve is from the outer ring, the more likely some points $\boldsymbol{h_p}$ will fall near a ridge of $G_{\mathrm{X}}$. (a) Case where the values $Z_{\boldsymbol{p}}$ are roughly evenly distributed on $\mathbb{S}^1$. (b) Case where these values are concentrated in a small portion of the unit circle. The most extreme cases occurs when $Z_{\boldsymbol{p}} = 1$ for any $\boldsymbol{p}$. Figure 3 (middle and right) depicts two such situations.

For any $p \in \mathbb{N} \setminus \{0\}$, we compute the $p$-th moment of $\mathsf{G}_{\mathrm{X}}^{\max}(\boldsymbol{x})$ defined in (82). By considering

$$
\begin{aligned}
g_{\max} : \mathbb{S}^1 &\to [-1, 1] \\
z &\mapsto \max_{\|\boldsymbol{p}\|_\infty \leq q} \mathrm{Re}\big(z^* Z_{\boldsymbol{p}}\big),
\end{aligned}
\tag{93}
$$

we get $\mathsf{G}_{\mathrm{X}}^{\max}(\boldsymbol{x}) = g_{\max}(\mathsf{Z}_{\mathrm{X}}(\boldsymbol{x}))$. A visual representation of $g_{\max}$ is provided in Figure 4, for two different values of $\boldsymbol{\theta}$. According to Hypothesis 2, $\mathsf{Z}_{\mathrm{X}}(\boldsymbol{x})$ follows a uniform distribution on $\mathbb{S}^1$. Therefore,

$$
\mathbb{E}\big[\mathsf{G}_{\mathrm{X}}^{\max}(\boldsymbol{x})^p\big] = \frac{1}{2\pi} \int_{\mathbb{S}^1} g_{\max}(z)^p \, \mathrm{d}\vartheta(z),
\tag{94}
$$

which proves that $\mathbb{E}\big[\mathsf{G}_{\mathrm{X}}^{\max}(\boldsymbol{x})^p\big]$ does not depend on $\boldsymbol{x}$. Let us split the unit circle $\mathbb{S}^1$ into the arcs $\mathfrak{A}_0^{(q)}, \ldots, \mathfrak{A}_{n_q-1}^{(q)}$ such as introduced in (78):

$$
\mathbb{E}\big[\mathsf{G}_{\mathrm{X}}^{\max}(\boldsymbol{x})^p\big] = \frac{1}{2\pi} \sum_{i=0}^{n_q-1} \int_{\mathfrak{A}_i^{(q)}} g_{\max}(z)^p \, \mathrm{d}\vartheta(z).
\tag{95}
$$

Let $i \in \{0, \ldots, n_q - 1\}$. We show that

$$\forall z \in \mathfrak{A}_i^{(q)}, \; g_{\max}(z) = \max\left(\mathrm{Re}(z^* Z_i^{(q)}), \, \mathrm{Re}(z^* Z_{i+1}^{(q)})\right). \tag{96}$$

Let $z \in \mathfrak{A}_i^{(q)}$ and $i' \notin \{i, i+1\}$. We prove that

$$\mathrm{Re}(z^* Z_{i'}^{(q)}) \leq \mathrm{Re}(z^* Z_i^{(q)}) \qquad \text{or} \qquad \mathrm{Re}(z^* Z_{i'}^{(q)}) \leq \mathrm{Re}(z^* Z_{i+1}^{(q)}). \tag{97}$$

On the one hand, we assume that $\angle(z^* Z_{i'}^{(q)}) \leq \pi$. By design of $(Z_i^{(q)})_{i \in \{0, \ldots, n_q-1\}}$, we have

$$Z_{i+1}^{(q)} \in [z, \, Z_{i'}^{(q)}]_{\mathbb{S}^1}. \tag{98}$$

Therefore, by definition of arcs on the unit circle (73), we get

$$\angle(z^* Z_{i+1}^{(q)}) \leq \angle(z^* Z_{i'}^{(q)}). \tag{99}$$

Then, since cos is non-increasing on $[0, \pi]$, we get

$$\cos\angle(z^* Z_{i+1}^{(q)}) \geq \cos\angle(z^* Z_{i'}^{(q)}), \tag{100}$$

which yields the right part of (97). On the other hand, if $\angle(z^* Z_{i'}^{(q)}) \geq \pi$, a similar reasoning yields the left part of (97). Then, (96) holds.

Next, we show that, as observed in Figure 4, $g_{\max}$ is piecewise-symmetric with respect to the center value of each arc $\mathfrak{A}_i^{(q)}$, denoted by

$$\overline{Z}_i^{(q)} := \sqrt{Z_i^{(q)} Z_{i+1}^{(q)}}. \tag{101}$$

Let $z_1, z_2 \in \mathfrak{A}_i^{(q)}$ which are symmetric with respect to $\overline{Z}_i^{(q)}$. Therefore, there exists $z' \in \mathbb{S}^1$ such that $z_1 = \overline{Z}_i^{(q)} z'$ and $z_2 = \overline{Z}_i^{(q)} z'^*$. We now prove that

$$g_{\max}(z_1) = g_{\max}(z_2). \tag{102}$$

A simple calculation yields

$$z_1^* Z_{i+1}^{(q)} = z'^* \widetilde{Z}_i^{(q)} \qquad \text{and} \qquad z_2^* Z_i^{(q)} = (z'^* \widetilde{Z}_i^{(q)})^*, \tag{103}$$

with

$$\widetilde{Z}_i^{(q)} := (Z_i^{(q)*} \overline{Z}_i^{(q)}) = (\overline{Z}_i^{(q)*} Z_{i+1}^{(q)}). \tag{104}$$

Therefore,

$$\mathrm{Re}(z_1^* Z_{i+1}^{(q)}) = \mathrm{Re}(z_2^* Z_i^{(q)}). \tag{105}$$

Since $z_1, z_2$ both belong to $\mathfrak{A}_i^{(q)}$, $g_{\max}(z_1)$ and $g_{\max}(z_2)$ satisfy (96). Then, by symmetry, (105) implies (102). One can observe from Figure 4 that $g_{\max}$ reaches its local minimum at the center of arc $\mathfrak{A}_i^{(q)}$, i.e., $\overline{Z}_i^{(q)}$. This corresponds to a point where $g_{\max}$ is non-differentiable.

We denote by $\overline{\mathfrak{A}}_i^{(q)} := [Z_i^{(q)}, \overline{Z}_i^{(q)}]_{\mathbb{S}^1}$ the first half of arc $\mathfrak{A}_i^{(q)}$. Then,

$$\forall z \in \overline{\mathfrak{A}}_i^{(q)}, \; g_{\max}(z) = \mathrm{Re}(z^* Z_i^{(q)}). \tag{106}$$

As a consequence, using symmetry, we get

$$\int_{\mathfrak{A}_i^{(q)}} g_{\max}(z)^p \, \mathrm{d}\vartheta(z) = 2 \int_{\overline{\mathfrak{A}}_i^{(q)}} g_{\max}(z)^p \, \mathrm{d}\vartheta(z)$$

$$= 2 \int_{\overline{\mathfrak{A}}_i^{(q)}} \mathrm{Re}(z^* Z_i^{(q)})^p \, \mathrm{d}\vartheta(z).$$

By using the change of variable formula (Athreya & Lahiri, 2006, p. 81) with $z \leftarrow e^{i\eta}$, we get

$$\int_{\mathfrak{A}_i^{(q)}} g_{\max}(z)^p \, \mathrm{d}\vartheta(z) = 2 \int_{H_i^{(q)}}^{\overline{H}_i^{(q)}} \cos^p\big(\eta - H_i^{(q)}\big) \, \mathrm{d}\eta, \tag{107}$$

where $\overline{H}_i^{(q)} := \big(H_i^{(q)} + H_{i+1}^{(q)}\big)/2$ denotes the argument of $\overline{Z}_i^{(q)}$. Then, the change of variable $\eta' \leftarrow \eta - H_i^{(q)}$ yields

$$\int_{\mathfrak{A}_i^{(q)}} g_{\max}(z)^p \, \mathrm{d}\vartheta(z) = 2 \int_0^{\delta H_i^{(q)}/2} \cos^p \eta' \, \mathrm{d}\eta'. \tag{108}$$

Next, we insert (108) into (95), and compute $\mathbb{E}\left[\mathsf{G}_{\mathrm{X}}^{\max}(\boldsymbol{x})^p\right]$ for $p \leftarrow 1$ and $p \leftarrow 2$:

$$\mathbb{E}\left[\mathsf{G}_{\mathrm{X}}^{\max}(\boldsymbol{x})\right] = \frac{1}{\pi} \sum_{i=0}^{n_q-1} \sin \frac{\delta H_i^{(q)}}{2};$$

$$\mathbb{E}\left[\mathsf{G}_{\mathrm{X}}^{\max}(\boldsymbol{x})^2\right] = \frac{1}{2} + \frac{1}{4\pi} \sum_{i=0}^{n_q-1} \sin \delta H_i^{(q)}.$$

We recall that $\mathsf{Q}_{\mathrm{X}} := 1 - \mathsf{G}_{\mathrm{X}}^{\max}$. By linearity of $\mathbb{E}$, we get

$$\mathbb{E}\left[\mathsf{Q}_{\mathrm{X}}(\boldsymbol{x})^2\right] := \frac{3}{2} + \frac{1}{4\pi} \sum_{i=0}^{n_q-1} \left(\sin \delta H_i^{(q)} - 8 \sin \frac{\delta H_i^{(q)}}{2}\right), \tag{109}$$

which concludes the proof. $\qquad\square$

We consider an ideal scenario where $\big(Z_i^{(q)}(m\boldsymbol{\theta})\big)_{i \in \{0,\,\ldots,\,n_q-1\}}$ are evenly spaced on $\mathbb{S}^1$. Then, an order 2 Taylor expansion yields

$$\gamma_q(m\boldsymbol{\theta}) = o(1/q^2), \tag{110}$$

providing an order-two-polynomial decay rate for $\mathsf{Q}_{\mathrm{X}}(\boldsymbol{x})$, when the grid half-size $q$ increases. Figure 5 displays $\boldsymbol{\theta} \mapsto \gamma_q(m\boldsymbol{\theta})^2$ for $\boldsymbol{\theta} \in [-\pi,\,\pi]^2$, with $m = 4$ and $q = 1$ as in AlexNet. We notice that, for the major part of the Fourier domain, $\gamma_q$ remains close to 0. However, we observe a regular pattern of dark regions, which correspond to pathological frequencies where the repartition of $\big(Z_i^{(q)}(m\boldsymbol{\theta})\big)_{i \in \{0,\,\ldots,\,n_q-1\}}$ is unbalanced.

So far, we established a result at the pixel level. Before stating Theorem 2, which extends the result to the image level, we need the following intermediate statement.

**Proposition 7.** *We consider the random variable*

$$\widetilde{\mathsf{S}}_{\mathrm{X}} := \big\|U_m^{\mathrm{mod}}\mathrm{X}\big\|_2. \tag{111}$$

*Under Hypothesis 3, for any $\boldsymbol{x} \in \mathbb{R}^2$,*

- $\mathsf{Z}_{\mathrm{X}}(\boldsymbol{x})$ *is independent of* $\widetilde{\mathsf{S}}_{\mathrm{X}}$;

- $\mathsf{Z}_{\mathrm{X}}(\boldsymbol{x})$, $\mathsf{M}_{\mathrm{X}}(\boldsymbol{x})$ *are conditionally independent given* $\widetilde{\mathsf{S}}_{\mathrm{X}}$.

*Proof.* See Appendix A.12. $\qquad\square$

Finally, Propositions 6 and 7 yield the following theorem. It provides an upper bound on the expected value of the normalized mean squared error $\widetilde{\mathsf{P}}_{\mathrm{X}}^2$, such as defined in (86).

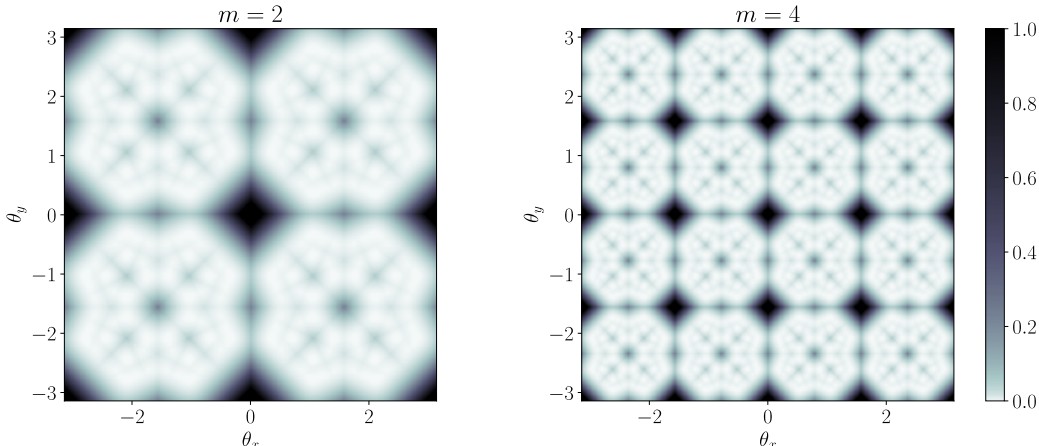

Figure 5. $\gamma(m\boldsymbol{\theta})^2$ as a function of the kernel characteristic frequency $\boldsymbol{\theta} \in [-\pi, \pi]^2$. According to Theorem 2, this quantity provides an approximate bound for the expected quadratic error between $\mathbb{R}$Max and $\mathbb{C}$Mod outputs. The subsampling factor $m$ has been set to 2 as in ResNet (left), and 4 as in AlexNet (right). The bright regions correspond to frequencies for which the two outputs are expected to be similar. However, in the dark regions, pathological cases such as illustrated in Figure 3 are more likely to occur.

**Theorem 2** (MSE between $\mathbb{C}$Mod and $\mathbb{R}$Max). *Let* $W \in \mathcal{J}(\boldsymbol{\theta}, \kappa)$ *denote a discrete Gabor-like filter,* $m \in \mathbb{N} \setminus \{0\}$ *a subsampling factor and* $q \in \mathbb{N} \setminus \{0\}$ *a grid half-size. We consider a stochastic process* $X$ *whose realizations are elements of* $l^2_{\mathbb{R}}(\mathbb{Z}^2)$. *We assume that condition* (38) *is satisfied:* $\kappa \leq \pi/m$. *Then, under Hypotheses 1 to 3,[2]*

$$\mathbb{E}\left[\widetilde{\mathsf{P}}^2_X\right] \leq \left(\beta_q(m\kappa) + \gamma_q(m\boldsymbol{\theta})\right)^2, \tag{112}$$

*where* $\widetilde{\mathsf{P}}^2_X$ (86) *denotes the stochastic quadratic error between* $\mathbb{C}$Mod *and* $\mathbb{R}$Max *outputs. We remind that* $\beta_q$ *and* $\gamma_q$ *have been introduced in* (72) *and* (91), *respectively.*

*Proof.* See Appendix A.13. □

Let us analyze the bound obtained in (112). The first term, $\beta_q(m\kappa)$, accounts for the localized property of the convolution filter $W$. This term decreases linearly with the product $m\kappa$. In the limit case where $\kappa = 0$ (infinite, nonlocal filter), we get $\beta_q(m\kappa) = 0$. Note that a smaller subsampling factor $m$ allows for a larger bandwidth $\kappa$. Moreover, $\beta_q(m\kappa)$ increases linearly with the size of the max pooling grid, which is characterized by $q$. The second term, $\gamma_q(m\boldsymbol{\theta})$, accounts for the discrete nature of the max pooling grid. It strongly depends on the characteristic frequency $\boldsymbol{\theta}$, as illustrated in Figure 5. According to (110), this term has a polynomial decay when $q$ increases. However, increasing the size of the max pooling grid also results in increasing the term $\beta_q(m\kappa)$, as explained above. Therefore, a tradeoff must be found to get an optimal bound.

## 4 Shift Invariance of $\mathbb{R}$Max Outputs

In this section, we present the main theoretical claim of this paper. Based on the previous results, we provide a probabilistic measure of shift invariance for $\mathbb{R}$Max operators. First, we state the following lemma.

**Lemma 6.** *If Hypotheses 2 and 3 are satisfied, then they are also true with* $X \leftarrow \mathcal{T}_{\boldsymbol{u}}X$, *for any* $\boldsymbol{u} \in \mathbb{R}^2$.

*Proof.* See Appendix A.14. □

We are now ready to state the main result about shift invariance of $\mathbb{R}$Max outputs.

---

[2]We can easily prove that these properties are independent from the choice of sampling interval $s > 0$.

**Theorem 3** (Shift invariance of $\mathbb{R}$Max)**.** *We assume that the requirements stated in Theorem 2 are satisfied. Additionally, given a translation vector $\boldsymbol{u} \in \mathbb{R}^2$, we consider the following random variable:*

$$\widetilde{\mathsf{R}}_{\mathrm{X}, \boldsymbol{u}} := \left\| U_{m, q}^{\max}(\mathcal{T}_{\boldsymbol{u}} \mathrm{X}) - U_{m, q}^{\max} \mathrm{X} \right\|_2 / \left\| U_m^{\mathrm{mod}} \mathrm{X} \right\|_2. \tag{113}$$

*Then, under condition* (38)*, we have*

$$\mathbb{E}\left[\widetilde{\mathsf{R}}_{\mathrm{X}, \boldsymbol{u}}\right] \leq 2 \left( \beta_q(m\kappa) + \gamma_q(m\boldsymbol{\theta}) \right) + \alpha(\kappa\boldsymbol{u}), \tag{114}$$

*where $\alpha$, $\beta_q$ and $\gamma_q$ are defined in* (18)*,* (72) *and* (91)*, respectively.*

*Proof.* See Appendix A.15. $\qquad\square$

In the bound established in (114), the sum $\beta_q(m\kappa) + \gamma_q(m\boldsymbol{\theta})$ accounts for the discrepancy between $\mathbb{R}$Max and $\mathbb{C}$Mod outputs, as stated in Theorem 2, whereas the term $\alpha(\kappa\boldsymbol{u})$ characterizes the stability of $\mathbb{C}$Mod outputs, as stated in Theorem 1. If $\kappa$ is sufficiently small, then $\alpha(\kappa\boldsymbol{u})$ and $\beta_q(m\kappa)$ become negligible with respect to $\gamma_q(m\boldsymbol{\theta})$, and the bound can be approximated by $2\,\gamma_q(m\boldsymbol{\theta})$. Theorem 3 therefore provides a validity domain for shift invariance of $\mathbb{R}$Max operators, as illustrated in Figure 5 with $q = 1$.

**Remark 5.** The stochastic discrepancy introduced in (113) is estimated relatively to the $\mathbb{C}$Mod output. This choice is motivated by the perfect shift invariance of its norm, as shown in Proposition 3.

**Remark 6.** In practice, most of the time max pooling is performed on a grid of size $3 \times 3$; therefore $q = 1$. For the sake of conciseness, we shall sometimes drop $q$ in the notations, which implicitly means $q = 1$.

## 5 Adaptation to Multichannel Convolution Operators

In this section, we adapt Theorems 1 to 3 to multichannel inputs (*e.g.*, RGB images), employed in conventional CNNs such as AlexNet or ResNet.

First, we define multichannel $\mathbb{R}$Max and $\mathbb{C}$Mod operators relatively to (1) and (5). We denote by $K$ and $L \in \mathbb{N} \setminus \{0\}$ the number of input and output channels, respectively. Additionally, we consider a *multichannel convolution tensor*

$$\mathbf{W} := (\mathrm{W}_{lk})_{l \in \{0, \ldots, L-1\}, \, k \in \{0, \ldots, K-1\}} \in \left( l_{\mathbb{C}}^2(\mathbb{Z}^2) \right)^{L \times K}. \tag{115}$$

Multichannel $\mathbb{R}$Max and $\mathbb{C}$Mod operators take as input images, denoted by

$$\mathbf{X} := (\mathrm{X}_k)_{k \in \{0, \ldots, K-1\}} \in \left( l_{\mathbb{R}}^2(\mathbb{Z}^2) \right)^K. \tag{116}$$

They are defined, for any given output channel $l \in \{0, \ldots, L-1\}$, by

$$U_{m, q, l}^{\max}[\mathbf{W}] : \mathbf{X} \mapsto \mathrm{MaxPool}_q \left( \sum_{k=0}^{K-1} \left( \mathrm{X}_k * \mathrm{Re}\,\overline{\mathrm{W}}_{lk} \right) \downarrow m \right); \tag{117}$$

$$U_{m, l}^{\mathrm{mod}}[\mathbf{W}] : \mathbf{X} \mapsto \left| \sum_{k=0}^{K-1} \left( \mathrm{X}_k * \overline{\mathrm{W}}_{lk} \right) \downarrow (2m) \right|, \tag{118}$$

where $m, q \in \mathbb{N} \setminus \{0\}$ respectively denote a subsampling factor and the max pooling grid half-size. Analogously to (49) for single-channel inputs, we now consider

$$\mathrm{Y}_l^{\max} := U_{m, q, l}^{\max}[\mathbf{W}]\,(\mathbf{X}) \qquad \text{and} \qquad \mathrm{Y}_l^{\mathrm{mod}} := U_{m, l}^{\mathrm{mod}}[\mathbf{W}]\,(\mathbf{X}). \tag{119}$$

Again, in what follows we omit the parameter between square brackets. To apply Theorems 1 to 3 to the current setting on the $l$-th output channel, we need the following hypotheses.

**Hypothesis 4** (Monochrome filters). Let

$$\widetilde{W}_l := \frac{1}{K} \sum_{k=0}^{K-1} W_{lk} \tag{120}$$

denote the mean kernel of the $l$-th output channel. Then, there exists $\boldsymbol{\mu}_l \in \mathbb{R}^K$ such that

$$\forall k \in \{0, \ldots, K-1\}, \, W_{lk} = \mu_{lk} \widetilde{W}_l. \tag{121}$$

**Hypothesis 5** (Gabor-like filters). There exists a bandwidth $\kappa > 0$ satisfying $\kappa \leq \pi/m$ and a frequency vector $\boldsymbol{\theta}_l \in [-\pi, \pi]^2$ such that

$$\widetilde{W}_l \in \mathcal{J}(\boldsymbol{\theta}_l, \kappa). \tag{122}$$

Note that the bandwidth $\kappa$ is not indexed by $l$, because it shall later be assumed to be shared across the output channels. Then, under Hypothesis 4, $Y_l^{\max}$ and $Y_l^{\mod}$ are the outputs of single-channel $\mathbb{R}$Max and $\mathbb{C}$Mod operators, as introduced in (1) and (5):

$$Y_l^{\max} = U_{m,q}^{\max}[\widetilde{W}_l](X_l^{\text{lum}}) \qquad \text{and} \qquad Y_l^{\mod} = U_m^{\mod}[\widetilde{W}_l](X_l^{\text{lum}}), \tag{123}$$

where $X_l^{\text{lum}} \in l_\mathbb{R}^2(\mathbb{Z}^2)$ ("luminance" image) is defined as the following linear combination:

$$X_l^{\text{lum}} := \sum_{k=0}^{K-1} \mu_{lk} X_k. \tag{124}$$

The results established for single-channel inputs can therefore be extended to multichannel operators. Specifically, we get the following corollaries to Theorems 1 to 3.

**Corollary 2** (Shift invariance of $\mathbb{C}$Mod). *For a given output channel $l \in \{0, \ldots, L-1\}$, we postulate Hypotheses 4 and 5. Then, for any input image $\mathbf{X} \in (l_\mathbb{R}^2(\mathbb{Z}^2))^K$ with finite support and any translation vector $\boldsymbol{u} \in \mathbb{R}^2$,*

$$\left\| U_{m,l}^{\mod}(\mathcal{T}_{\boldsymbol{u}} \mathbf{X}) - U_{m,l}^{\mod} \mathbf{X} \right\|_2 \leq \alpha(\kappa \boldsymbol{u}) \left\| U_{m,l}^{\mod} \mathbf{X} \right\|_2, \tag{125}$$

*where $\alpha$ has been defined in (18).*

**Corollary 3** (MSE between $\mathbb{C}$Mod and $\mathbb{R}$Max). *As in Corollary 2, we postulate Hypotheses 4 and 5. Again, we assume that condition (38) is satisfied: $\kappa \leq \pi/m$. Additionally, we consider $\mathbf{X}$ as a stack of $K$ discrete stochastic processes, and assume Hypotheses 1 to 3 with $X \leftarrow X_l^{\text{lum}}$ and $W \leftarrow \widetilde{W}_l$. Then,*

$$\mathbb{E}\left[\widetilde{P}_{\mathbf{X},l}^2\right] \leq \left(\beta_q(m\kappa) + \gamma_q(m\boldsymbol{\theta}_l)\right)^2, \tag{126}$$

*where we have defined the following random variable:*

$$\widetilde{P}_{\mathbf{X},l} := \left\| U_{m,l}^{\mod} \mathbf{X} - U_{m,l}^{\max} \mathbf{X} \right\|_2 / \left\| U_{m,l}^{\mod} \mathbf{X} \right\|_2. \tag{127}$$

**Corollary 4** (Shift invariance of $\mathbb{R}$Max). *We assume that the requirements stated in Corollary 3 are satisfied. Then, for any translation vector $\boldsymbol{u} \in \mathbb{R}^2$,*

$$\mathbb{E}\left[\widetilde{R}_{\mathbf{X},\boldsymbol{u},l}\right] \leq 2\left(\beta_q(m\kappa) + \gamma_q(m\boldsymbol{\theta}_l)\right) + \alpha(\kappa \boldsymbol{u}), \tag{128}$$

*where we have defined the following random variable:*

$$\widetilde{R}_{\mathbf{X},\boldsymbol{u},l} := \left\| U_{m,l}^{\max}(\mathcal{T}_{\boldsymbol{u}} \mathbf{X}) - U_{m,l}^{\max} \mathbf{X} \right\|_2 / \left\| U_{m,l}^{\mod} \mathbf{X} \right\|_2. \tag{129}$$

**Remark 7.** In the above results, we used a translation operator on multichannel tensors, obtained by applying $\mathcal{T}_{\boldsymbol{u}}$, as defined in (28), to each channel $X_k$.

# 6 A Case Study Implementing the Dual-Tree Complex Wavelet Packet Transform

In this section, we experimentally validate the results stated in Theorems 1 to 3. To this end, we consider a fully-deterministic scenario implementing the dual-tree complex wavelet packet transform (DT-$\mathbb{C}$WPT), which exhibit characteristics akin to those observed in the initial convolution layer of freely-trained CNNs such as AlexNet or ResNet. In particular, as stated in Section 6.1, DT-$\mathbb{C}$WPT achieves subsampled convolutions with oriented band-pass filters tiling the Fourier domain into overlapping square windows. As such, it provides a convenient framework to experimentally validate our theoretical findings in a controlled environment. Then, in Section 6.2, we build $\mathbb{C}$Mod and $\mathbb{R}$Max operators based on DT-$\mathbb{C}$WPT convolution kernels.

## 6.1 Main Properties

In what follows, we outline the principal characteristics of DT-$\mathbb{C}$WPT. A detailed description of the transform itself is provided in Appendix C.1, whereas the results presented hereafter are formally established in Appendices C.2 and C.3.

For a given decomposition depth $J \in \mathbb{N} \setminus \{0\}$, DT-$\mathbb{C}$WPT achieves subsampled convolutions with $4 \times 4^J$ oriented band-pass filters that tile the Fourier domain into overlapping square windows of size

$$\kappa_J := \pi/m_J, \qquad \text{with} \qquad m_J := 2^{J-1}. \tag{130}$$

More specifically, considering an input image $X \in l^2_{\mathbb{R}}(\mathbb{Z}^2)$, it produces a set of $4 \times 4^J$ output feature maps

$$\mathbf{D}^{(J)} := \left( D_l^{\nearrow(J)}, D_l^{\searrow(J)}, D_l^{\swarrow(J)}, D_l^{\nwarrow(J)} \right)_{l \in \{0, \dots, 4^J - 1\}}, \tag{131}$$

where each arrow points to the Fourier quadrant where the feature map's energy is concentrated. Moreover, as stated in Proposition 10, for any $l \in \left\{0, \dots, 4^J - 1\right\}$, there exists $W_l^{\nearrow(J)} \in l^2_{\mathbb{C}}(\mathbb{Z}^2)$ such that

$$D_l^{\nearrow(J)} = \left( X * \overline{W}_l^{\nearrow(J)^*} \right) \downarrow 2^J. \tag{132}$$

An interesting property is that each kernel $W_l^{\nearrow(J)}$ approximately satisfies

$$W_l^{\nearrow(J)} \in \mathcal{J}\left( \boldsymbol{\theta}_l^{(J)}, \kappa_J \right) \tag{133}$$

for a certain characteristic frequency $\boldsymbol{\theta}_l^{(J)} \in [0, \pi]^2$. In other words, it approximately behaves as a Gabor-like filter in the discrete framework (11). Moreover, each kernel corresponds to a different frequency, thereby covering the top-right quadrant of the Fourier domain. Similar results can be established for the other three Fourier quadrants. Graphical representations of $\mathbf{W}^{\nearrow(J)} := \left( W_l^{\nearrow(J)} \right)_{l \in \{0, \dots, 4^J - 1\}}$ and $\mathbf{W}^{\searrow(J)} := \left( W_l^{\searrow(J)} \right)_{l \in \{0, \dots, 4^J - 1\}}$ are provided in Figure 6 with $J = 2$ (Figure 6a, 32 filters) and $J = 3$ (Figure 6b, 128 filters).

The $\mathbb{R}$Max and $\mathbb{C}$Mod operators implemented in our experiments respectively satisfy (1) and (5) with with $W \leftarrow W_l^{\nearrow(J)}$ or $W_l^{\searrow(J)}$, and $m \leftarrow m_J$. Note that increasing the decomposition depth $J$, and therefore the subsampling factor $m_J$, results in a decreased Fourier support size $\kappa_J$, therefore matching the condition stated in (38) $\kappa \leftarrow \kappa_J$ and $m \leftarrow m_J$.

**Remark 8.** Because X is real-valued, the feature maps $D_l^{\swarrow(J)}$ and $D_l^{\nwarrow(J)}$ are the respective complex conjugates of $D_l^{\nearrow(J)}$ and $D_l^{\searrow(J)}$, and thus do not need to be explicitly computed. Then, we can easily show that $W_l^{\swarrow(J)}$ and $W_l^{\nwarrow(J)}$ are also the complex conjugates of $W_l^{\nearrow(J)}$ and $W_l^{\searrow(J)}$, respectively.

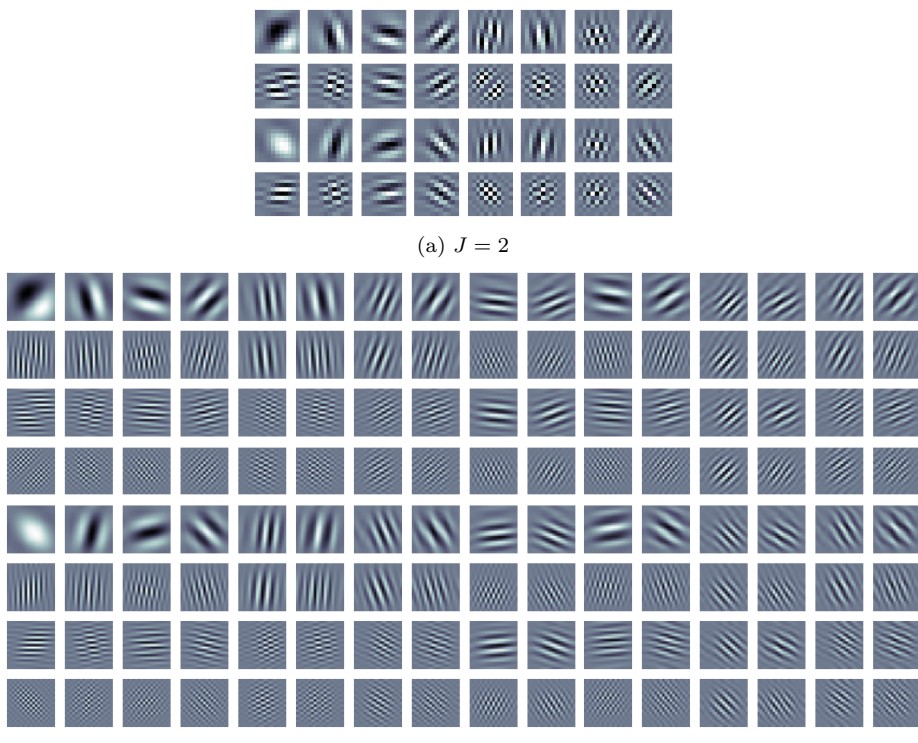

(a) $J = 2$

(b) $J = 3$

Figure 6. Real part of the convolution kernels $\mathbf{W}^{\nearrow(J)}$, $\mathbf{W}^{\searrow(J)}$, with $J = 2$ (32 filters, $m_J = 2$) and $J = 3$ (128 filters, $m_J = 4$), respectively. The kernels have been computed using Q-shift orthogonal QMFs of length 10 (Kingsbury, 2003). The kernels have been respectively cropped to size $11 \times 11$ and $19 \times 19$, for the sake of legibility. Note that the filters displayed in (a) and (b) share similarities with those found in, respectively, ResNet ($m = 2$) and AlexNet ($m = 4$), after training with ImageNet.

## 6.2 DT-$\mathbb{C}$WPT-Based $\mathbb{R}$Max and $\mathbb{C}$Mod Operators

According to (130), (132), and (133), we can apply Theorems 1 to 3 to the dual-tree framework. More precisely, for any output channel $l \in \{0, \ldots, 4^J - 1\}$, we consider the following $\mathbb{R}$Max and $\mathbb{C}$Mod operators:

$$U_l^{\max\nearrow} : \mathrm{X} \mapsto \mathrm{MaxPool}\left(\left(\mathrm{X} * \mathrm{Re}\,\overline{\mathrm{W}}_l^{\nearrow(J)}\right) \downarrow 2^{J-1}\right); \tag{134}$$

$$U_l^{\mathrm{mod}\nearrow} : \mathrm{X} \mapsto \left|\left(\mathrm{X} * \overline{\mathrm{W}}_l^{\nearrow(J)}\right) \downarrow 2^J\right|. \tag{135}$$

Using the notations introduced in (5) and (1), we have

$$U_l^{\max\nearrow} = U_{m_J}^{\max}\left[\mathrm{W}_l^{\nearrow(J)}\right] \qquad \text{and} \qquad U_l^{\mathrm{mod}\nearrow} := U_{m_J}^{\mathrm{mod}}\left[\mathrm{W}_l^{\nearrow(J)}\right], \tag{136}$$

where we have defined $m_J := 2^{J-1}$. Note that, following Remark 6, we have omitted the grid half-size $q$, which is equal to 1 (max pooling operates on a grid of size $3 \times 3$). Furthermore, for the sake of brevity, we have omitted the depth $J$ in the above notations.

**Remark 9.** Both $U_l^{\max\nearrow}$ and $U_l^{\mathrm{mod}\nearrow}$ are implemented using DT-$\mathbb{C}$WPT with $J$ decomposition stages. However, in (134), the subsampling factor is equal to $2^{J-1}$, instead of $2^J$, as stated in (132). In order to accommodate this property of $\mathbb{R}$Max operators, the last stage of DT-$\mathbb{C}$WPT decomposition is carried out without subsampling, resulting in higher redundancy. This is similar to the concept of stationary wavelet transform as described by Nason & Silverman (1995). Furthermore, only the real component of the wavelet feature maps is preserved. On the other hand, $U_l^{\mathrm{mod}\nearrow}$ implements a fully-decimated wavelet packet transform, and keeps both real and imaginary parts. Figure 7 illustrates these technical details.

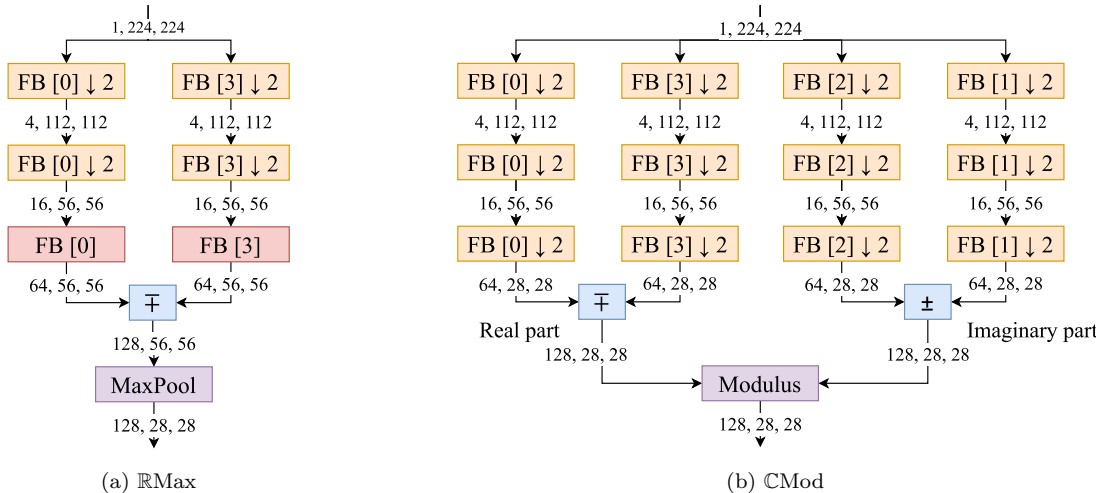

(a) $\mathbb{R}$Max

(b) $\mathbb{C}$Mod

Figure 7. Detailed illustration of the $\mathbb{R}$Max (a) and $\mathbb{C}$Mod (b) operators based on DT-$\mathbb{C}$WPT, with $J = 3$ decomposition stages. The numbers between modules correspond to the number of feature maps, height and width. The orange modules represent subsampled convolutions using one of the four 2D filter banks $\mathbf{G}^{[0-3]}$, such as introduced in (243). The FB index is indicated between square brackets. The $\mathbb{R}$Max model (a) only computes the real part of the dual-tree coefficients, and the last stage of decomposition is performed without subsampling (red modules). Additionally, the blue modules represent linear combinations of feature maps such as described in (248).

## 6.3 Experiments and Results

We implemented the $\mathbb{R}$Max and $\mathbb{C}$Mod operators $U_l^{\max\nearrow}$ and $U_l^{\mod\nearrow}$, as introduced in (134) and (135), with both $J = 2$ and 3 stages of wavelet packet decomposition. To cover the whole frequency plane, we also implemented similar operators, denoted by $U_l^{\max\searrow}$ and $U_l^{\mod\searrow}$. They are associated with the convolution filters $\mathrm{W}_l^{\searrow(J)}$, introduced in Proposition 10, with energy being located in the bottom-right quadrant. However, as explained in Remark 8, we did not need to deal with the two other quadrants (negative $x$-values). Using the validation set of ImageNet-1K (Russakovsky et al., 2015), ($N := 50\,000$ images), we measured the mean discrepancy between $\mathbb{R}$Max and $\mathbb{C}$Mod outputs, and evaluated the shift invariance of both models. Dual-tree decompositions have been performed with Q-shift orthogonal filters of length 10 (Kingsbury, 2003).

### 6.3.1 MSE between $\mathbb{R}$Max and $\mathbb{C}$Mod

Each image $n \in \{0, \ldots, N - 1\}$ in the dataset was converted to grayscale, from which a center crop of size $224 \times 224$ was extracted. We denote by $\mathrm{X}_n \in l_{\mathbb{R}}^2(\mathbb{Z}^2)$ the resulting input feature map. For any $l \in \{0, \ldots, 4^J - 1\}$, corresponding to a specific area in the Fourier plane, we denote by

$$\mathrm{Y}_{nl}^{\max\nearrow} := U_l^{\max\nearrow}(\mathrm{X}_n) \qquad \text{and} \qquad \mathrm{Y}_{nl}^{\mod\nearrow} := U_l^{\mod\nearrow}(\mathrm{X}_n) \tag{137}$$

the outputs of the $l$-th $\mathbb{R}$Max and $\mathbb{C}$Mod operators as defined in (134) and (135), respectively. We adopt similar notations for the bottom-right Fourier quadrant. Then, the normalized mean squared error between $\mathrm{Y}_{nl}^{\mod\nearrow}$ and $\mathrm{Y}_{nl}^{\max\nearrow}$ was computed. It is defined by the square of

$$\rho_{nl}^{\nearrow} := \left\| \mathrm{Y}_{nl}^{\mod\nearrow} - \mathrm{Y}_{nl}^{\max\nearrow} \right\|_2 / \left\| \mathrm{Y}_{nl}^{\mod\nearrow} \right\|_2. \tag{138}$$

Finally, the for each output channel $l$, an empirical estimate for $\mathbb{E}\big[\widetilde{\mathsf{P}}_{\mathrm{X}}^2\big]$, introduced in (86), was obtained by averaging $\rho_{nl}^{\nearrow 2}$ over the whole dataset. We denote by $\widetilde{\rho}_l^{\nearrow 2}$ the corresponding quantity.

Since $U_l^{\max\nearrow}$ and $U_l^{\mod\nearrow}$ are parameterized by $\mathrm{W}_l^{\nearrow(J)}$, it follows that $\widetilde{\rho}_l^{\nearrow 2}$ depends on the filter's characteristic frequency $\boldsymbol{\theta}_l^{(J)}$ (133). According to Proposition 11, these frequencies form a regular grid in the top-right

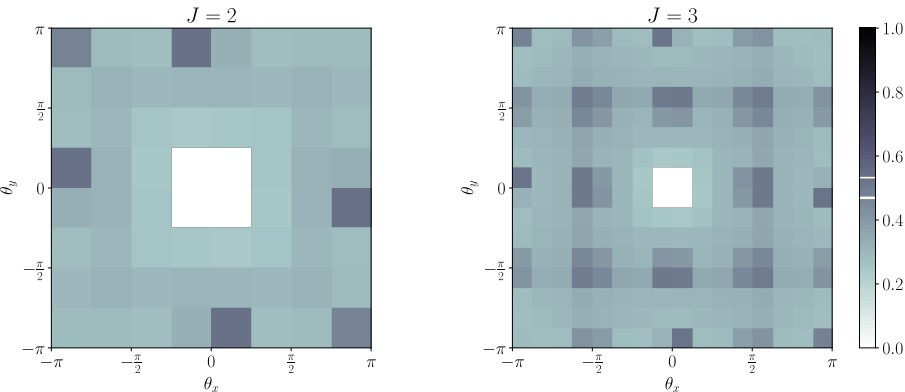

Figure 8. Empirical estimates of the normalized mean squared error between $\mathbb{R}\mathrm{Max}$ and $\mathbb{C}\mathrm{Mod}$ outputs, computed on ImageNet-1K (validation set), for each filter frequency. For each channel $l \in \left\{0, \dots, 4^J - 1\right\}$, $\widetilde{\rho_l^{\nearrow}}^2$ is plotted as a grayscale pixel centered in $\boldsymbol{\theta}_l^{(J)}$ such as introduced in eq. (133) (top-right quadrant). Similarly, $\widetilde{\rho_l^{\searrow}}^2$ is plotted in the bottom-right quadrant. Finally, the bottom- and top-left quadrants ($\widetilde{\rho_l^{\swarrow}}^2$ and $\widetilde{\rho_l^{\nwarrow}}^2$) are simply obtained by symmetrizing the figures. Since the subsampling factor $m_J$ is equal to $2^{J-1}$, these experimental results can be compared with the left and right parts of Figure 5. Note that the low-pass filters have been discarded because they are outside the scope of this study.

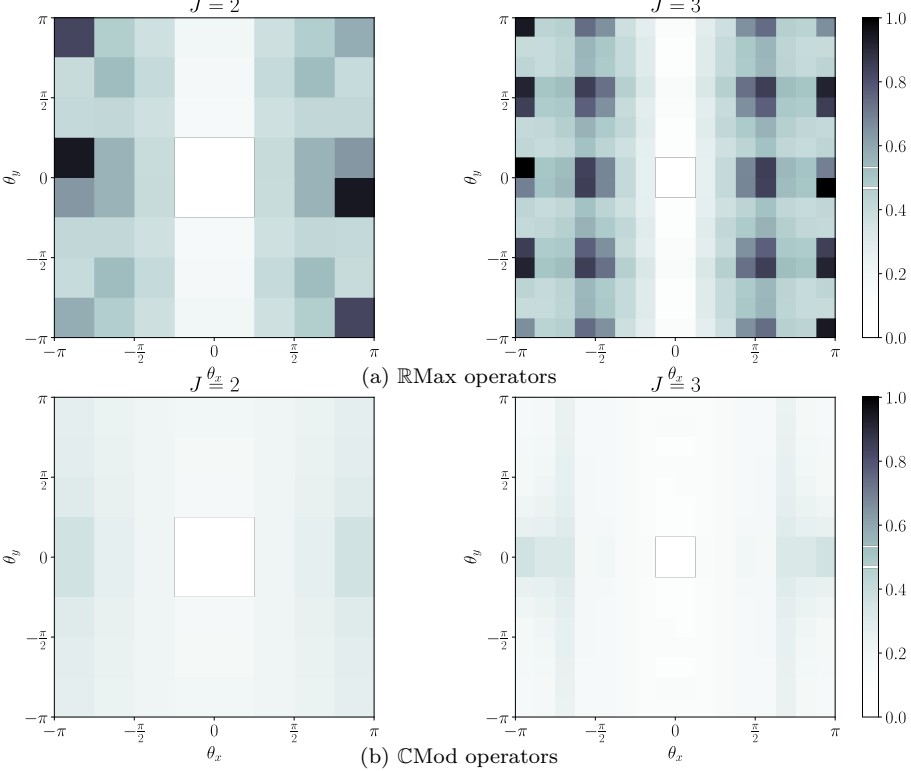

Figure 9. Shift invariance of $\mathbb{R}\mathrm{Max}$ and $\mathbb{C}\mathrm{Mod}$ outputs, computed on ImageNet 2012 (validation set), for each filter frequency. For each $l \in \left\{0, \dots, 4^J - 1\right\}$, $\widetilde{\rho_l^{\mathrm{max}\nearrow}}$ (Figure 9a) and $\widetilde{\rho_l^{\mathrm{mod}\nearrow}}$ (Figure 9b) are plotted by applying the same procedure as in Figure 8.

quadrant of Fourier domain. This provides a visual representation of $\widetilde{\rho}_l^{\nearrow 2}$, as shown in Figure 8. This figure also displays $\widetilde{\rho}_l^{\searrow 2}$, corresponding to the bottom-right quadrant. The half-plane of negative $x$-values has simply been symmetrized, following Remark 8. We can observe a regular pattern of dark spots. More precisely, high discrepancies between max pooling and modulus seem to occur when the energy of $\mathrm{W}_l^{\nearrow(J)}$ or $\mathrm{W}_l^{\searrow(J)}$ overlaps a dark region of Figure 5. This result corroborates Theorem 3, which states that high discrepancies are expected for certain pathological frequencies, due to the search for a maximum value over a discrete grid.

### 6.3.2 Shift invariance

For each input image previously converted to grayscale, two crops of size $224 \times 224$ were extracted, such that the corresponding sequences $\mathrm{X}_n$ and $\mathrm{X}'_n$ are shifted by one pixel along the $x$-axis. From these inputs, the following quantity was then computed:

$$\rho_{nl}^{\max\nearrow} := \left\| \mathrm{Y}_{nl}^{\max'\nearrow} - \mathrm{Y}_{nl}^{\max\nearrow} \right\|_2 / \left\| \mathrm{Y}_{nl}^{\mathrm{mod}\nearrow} \right\|_2, \tag{139}$$

where $\mathrm{Y}_{nl}^{\max'\nearrow}$ satisfies (136) with $\mathrm{X}_n \leftarrow \mathrm{X}'_n$. Finally, for each output channel $l \in \{0, \ldots, 4^J - 1\}$, an empirical estimate for $\mathbb{E}\big[\widetilde{\mathrm{R}}_{\mathrm{X}, \boldsymbol{u}}\big]$, satisfying (113) with $\boldsymbol{u} = (1, 0)^\top$, was obtained by averaging $\rho_{nl}^{\max\nearrow}$ over the whole dataset. We denote by $\widetilde{\rho}_l^{\max\nearrow}$ the corresponding quantity. We point out that shift invariance is measured relatively to the norm of the $\mathbb{C}$Mod output, as explained in Remark 5.

On the other hand, the same procedure was applied to the $\mathbb{C}$Mod operators:

$$\rho_{nl}^{\mathrm{mod}\nearrow} := \left\| \mathrm{Y}_{nl}^{\mathrm{mod}'\nearrow} - \mathrm{Y}_{nl}^{\mathrm{mod}\nearrow} \right\|_2 / \left\| \mathrm{Y}_{nl}^{\mathrm{mod}\nearrow} \right\|_2, \tag{140}$$

and $\widetilde{\rho}_l^{\mathrm{mod}\nearrow}$ was obtained as before by averaging $\rho_{nl}^{\mathrm{mod}\nearrow}$ over the whole dataset.

A visual representation of $\widetilde{\rho}_l^{\max\nearrow}$ and $\widetilde{\rho}_l^{\mathrm{mod}\nearrow}$ are provided in Figure 9 (as well as the other Fourier quadrants). Two observations can be drawn here.

(1) When the filter is horizontally oriented, the corresponding output is highly stable with respect to horizontal shifts. This can be explained by noticing that such kernels perform low-pass filtering along the $x$-axis. The exact transposed phenomenon occurs for vertical shifts.

(2) Elsewhere, we observe that high discrepancies between $\mathbb{R}$Max and $\mathbb{C}$Mod outputs (Figure 8) are correlated with shift instability of $\mathbb{R}$Max (Figure 9, top). This is in line with (112) and (114) in Theorems 2 and 3. Note that $\mathbb{C}$Mod outputs are nearly shift invariant regardless the characteristic frequency $\boldsymbol{\theta}_l^{(J)}$ (Figure 9, bottom), as predicted by Theorem 1 (39).

To conclude this experimental study, we note that the plots from Figures 8 and 9 remarkably align with Figure 5, where the term $\gamma(m\boldsymbol{\theta})$ is represented as a function of the filter's frequency. This term, appearing in Theorems 2 and 3, plays a crucial role in determining the discrepancies between $\mathbb{R}$Max and $\mathbb{C}$Mod outputs, which in turn dictate shift invariance for $\mathbb{R}$Max. The dark regions in Figure 5 correspond to the frequencies where this operator is expected to be less stable, which is consistent with our experimental findings.

## 7 Discussion and Conclusion

In this paper, we conducted a theoretical and empirical study on the shift invariance properties captured by the max pooling operator, when applied to a convolution layer with Gabor-like kernels. We established a validity domain for near-shift invariance and confirmed our predictions through an experimental setting based on the dual-tree complex wavelet packet transform. Our results indicate that the $\mathbb{R}$Max operator inherits the shift-invariance properties of $\mathbb{C}$Mod—except at certain filter frequencies regularly scattered in the Fourier plane, where potential degeneracies may arise after max pooling.

In this context, a follow-up study applying these principles to real-world architectures was published as a conference paper, and is included as supplementary material for anonymization purposes (Authors, 2024).

In this companion paper, we empirically demonstrated that replacing $\mathbb{R}$Max by $\mathbb{C}$Mod in the first layer of AlexNet and ResNet architectures improves both shift invariance and prediction accuracy, while reducing computational costs and memory footprint. We thus evidenced that instability at early stages can propagate through deeper layers and have visible impact on the network's overall behavior. We emphasize that this companion paper is not a prior version of the present submission but rather an application of its principles.

Before wrapping up, we would like to discuss three potential limitations to this study.

- Our results primarily apply to early layers in CNNs, where the learned filters are known to resemble Gabor-like waveforms. While this may seem overly restrictive, we point out that the behavior of these layers can have a significant influence on the overall network stability, as discussed in the previous paragraph. Aliasing and shift sensitivity in deeper layers are important issues as well, but they require a different analytical approach that falls beyond the scope of this work. Nevertheless, we believe that our study provides a partial but significant understanding of shift invariance properties in CNNs, which is grounded in wavelet and image processing theory.

- This study is limited to Gabor-like filters; yet in practice, not all convolution kernels exhibit this property, as observed in Figure 1. However, many of those filters have their spectrum localized in the lower frequencies, and therefore do not suffer from aliasing effects. A few remaining kernels may be more challenging to study from a shift invariance perspective, but we purposely left them outside the scope of our study. Generally speaking, while simplifying assumptions were required for mathematical tractability, they offered a solid framework to understand some properties that are truly observed in CNNs and, to our knowledge, had not been previously documented.

- In recent years, the use of large convolution kernels and max pooling layers has declined in favor of smaller kernels and alternative downsampling strategies. However, relatively large $7 \times 7$ convolution kernels—similar to those used in ResNet—have been reintroduced in modern CNN architectures (Liu et al., 2022; Woo et al., 2023), partly motivated by the localized self-attention windows used in the Swin Transformer architecture. Even setting these recent examples aside, we argue that a theoretical study primarily focused on legacy architectures that have been proven successful in the past can offer new insights into their underlying mechanisms, guide architectural choices, and inspire future research—especially in a field where empirical innovations often precede theoretical understanding. In particular, our finding that $\mathbb{R}$Max exhibits instability at certain filter frequencies may help explain recent shifts in architectural trends.

A link was missing between real- and complex-valued convolutions in CNNs. By comparing the outputs of $\mathbb{C}$Mod and $\mathbb{R}$Max operators, we established a connection between these two worlds, creating opportunities for extensions of the results obtained for complex wavelet transforms. To paraphrase Tygert et al. (2016), the correspondence between standard real-valued CNNs (using max pooling) and complex wavelets is no longer "just a vague analogy."

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

# A  Appendix − Proofs

## A.1  Proof of Lemma 1

*Proof.* Applying the Fourier transform on (15) yields, for any $\boldsymbol{\xi} \in \mathbb{R}^2$,

$$\widehat{F_0}(\boldsymbol{\xi}) = \widehat{(F * \overline{\Psi})}(\boldsymbol{\xi} - \boldsymbol{\nu}) = \mathcal{T}_{\boldsymbol{\nu}}\big(\widehat{F}\,\widehat{\overline{\Psi}}\big)(\boldsymbol{\xi}). \tag{141}$$

By hypothesis on $\Psi$, we have

$$\mathrm{supp}(\widehat{F}\,\widehat{\overline{\Psi}}) \subset \mathrm{supp}\,\widehat{\overline{\Psi}} \subset B_\infty(-\boldsymbol{\nu},\,\varepsilon/2). \tag{142}$$

The translation operator $\mathcal{T}_{\boldsymbol{\nu}}$ shifts the support with respect to $\boldsymbol{\nu}$, which yields (16). □

### A.2 Proof of Proposition 1

*Proof.* Using the 2D Plancherel formula, we compute

$$
\begin{aligned}
\|\mathcal{T}_{\boldsymbol{h}}F_0 - F_0\|_{L^2}^2 &= \frac{1}{4\pi^2}\left\|\widehat{\mathcal{T}_{\boldsymbol{h}}F_0} - \widehat{F_0}\right\|_{L^2}^2 \\
&= \frac{1}{4\pi^2}\iint_{B_\infty(\varepsilon/2)}\left|\widehat{F_0}(\boldsymbol{\xi})\right|^2\left|\mathrm{e}^{-i\langle\boldsymbol{h},\,\boldsymbol{\xi}\rangle} - 1\right|^2\mathrm{d}^2\boldsymbol{\xi} \\
&= \frac{1}{4\pi^2}\iint_{B_\infty(\varepsilon/2)}\left|\widehat{F_0}(\boldsymbol{\xi})\right|^2\left(2 - 2\cos\langle\boldsymbol{h},\,\boldsymbol{\xi}\rangle\right)\mathrm{d}^2\boldsymbol{\xi} \\
&\leq \frac{1}{4\pi^2}\iint_{B_\infty(\varepsilon/2)}\left|\widehat{F_0}(\boldsymbol{\xi})\right|^2\left|\langle\boldsymbol{h},\,\boldsymbol{\xi}\rangle\right|^2\mathrm{d}^2\boldsymbol{\xi},
\end{aligned}
$$

because $\cos t \geq 1 - \frac{t^2}{2}$. Note that the integral is computed on a compact domain because, according to Lemma 1, $\operatorname{supp}\widehat{F_0} \subset B_\infty(\varepsilon/2)$. Next, we use the Cauchy-Schwarz inequality to compute:

$$
\begin{aligned}
\forall\boldsymbol{\xi}\in B_\infty(\varepsilon/2),\ |\langle\boldsymbol{h},\,\boldsymbol{\xi}\rangle| &\leq \|\boldsymbol{h}\|_1\cdot\|\boldsymbol{\xi}\|_\infty \\
&\leq \frac{\varepsilon}{2}\|\boldsymbol{h}\|_1.
\end{aligned}
$$

Therefore,

$$
\|\mathcal{T}_{\boldsymbol{h}}F_0 - F_0\|_{L^2}^2 \leq \frac{\varepsilon}{4}\|\boldsymbol{h}\|_1^2\|F_0\|_{L^2}^2, \tag{143}
$$

which yields the result. □

### A.3 Proof of Lemma 2

*Proof.* Since $F \in \mathcal{V}^{(s)}$, the two-dimensional version of Shannon's sampling theorem (Mallat, 2009, Theorem 3.11, p. 81) yields

$$
F = \sum_{\boldsymbol{n}\in\mathbb{Z}^2}\mathrm{X}[\boldsymbol{n}]\,\Phi_{\boldsymbol{n}}^{(s)}, \qquad\text{and}\qquad \widehat{F} = \sum_{\boldsymbol{n}\in\mathbb{Z}^2}\mathrm{X}[\boldsymbol{n}]\,\widehat{\Phi_{\boldsymbol{n}}^{(s)}}. \tag{144}
$$

Moreover, using (19), we can show that, for any $\boldsymbol{\xi}\in B_\infty(\pi/s)$,

$$
\widehat{\Phi_{\boldsymbol{n}}^{(s)}}(\boldsymbol{\xi}) = \widehat{\Phi^{(s)}}(\boldsymbol{\xi})\,\mathrm{e}^{-i\langle s\boldsymbol{\xi},\,\boldsymbol{n}\rangle} = s\,\mathrm{e}^{-i\langle s\boldsymbol{\xi},\,\boldsymbol{n}\rangle}. \tag{145}
$$

Therefore, plugging (145) into (144) proves (22).

Then, by combining (22) with the Plancherel formula, we get

$$
\begin{aligned}
\|F\|_{L^2}^2 &= \frac{1}{4\pi^2}\|\widehat{F}\|_{L^2}^2 \\
&= \frac{1}{4\pi^2}\iint_{B_\infty(\pi/s)}|\widehat{F}(\boldsymbol{\xi})|^2\,\mathrm{d}^2\boldsymbol{\xi} \\
&= \frac{1}{4\pi^2}\iint_{B_\infty(\pi/s)}\left|s\,\widehat{\mathrm{X}}(s\boldsymbol{\xi})\right|^2\,\mathrm{d}^2\boldsymbol{\xi}.
\end{aligned}
$$

The integral is performed on $B_\infty(\pi/s)$ because $F \in \mathcal{V}^{(s)}$. Then, by applying the change of variable $\boldsymbol{\xi}' \leftarrow s\boldsymbol{\xi}$, we get

$$
\begin{aligned}
\|F\|_{L^2}^2 &= \frac{1}{4\pi^2}\iint_{B_\infty(\pi)}\left|\widehat{\mathrm{X}}(\boldsymbol{\xi}')\right|^2\,\mathrm{d}^2\boldsymbol{\xi}' \\
&= \frac{1}{4\pi^2}\|\widehat{\mathrm{X}}\|_{L^2}^2 = \|\mathrm{X}\|_2^2,
\end{aligned}
$$

hence (23), which concludes the proof. □

### A.4 Proof of Proposition 2

*Proof.* First, $F_X$ and $\Psi_W$ are well defined because $X \in l^2_{\mathbb{R}}(\mathbb{Z}^2)$ and $W \in l^2_{\mathbb{C}}(\mathbb{Z}^2)$. By construction, $F_X$ and $\Psi_W \in \mathcal{V}^{(s)}$. Therefore, according to Shannon's sampling theorem (Mallat, 2009, Theorem 3.11, p. 81),

$$F_X := s \sum_{\boldsymbol{n} \in \mathbb{Z}^2} F_X(s\boldsymbol{n}) \, \Phi_{\boldsymbol{n}}^{(s)} \qquad \text{and} \qquad \Psi_W := s \sum_{\boldsymbol{n} \in \mathbb{Z}^2} \Psi_W(s\boldsymbol{n}) \, \Phi_{\boldsymbol{n}}^{(s)}. \tag{146}$$

By uniqueness of decompositions in an orthonormal basis, we get (26). Moreover, using (22) in Lemma 2, we get, for any $\boldsymbol{\xi} \in B_\infty(\pi/s)$,

$$\widehat{\Psi}_W(\boldsymbol{\xi}) = s \, \widehat{W}(s\boldsymbol{\xi}). \tag{147}$$

Since $\widehat{\Psi}_W(\boldsymbol{\xi}) = 0$ outside $B_\infty(\pi/s)$, (147) is true for any $\boldsymbol{\xi} \in \mathbb{R}^2$. Therefore, by hypothesis on W,

$$\operatorname{supp} \widehat{\Psi}_W \subset B_\infty\big(\boldsymbol{\theta}/s, \, \kappa/(2s)\big), \tag{148}$$

which yields (25).

We now prove (27). For $\boldsymbol{n} \in \mathbb{Z}^2$, we compute:

$$\big(F_X * \overline{\Psi}_W\big)(ms\boldsymbol{n}) = \iint_{\mathbb{R}^2} F_X(ms\boldsymbol{n} - \boldsymbol{x}) \overline{\Psi}_W(\boldsymbol{x}) \, \mathrm{d}^2\boldsymbol{x}$$

$$= \iint_{\mathbb{R}^2} \sum_{\boldsymbol{p} \in \mathbb{Z}^2} X[\boldsymbol{p}] \Phi_{\boldsymbol{p}}^{(s)}(ms\boldsymbol{n} - \boldsymbol{x}) \overline{\Psi}_W(\boldsymbol{x}) \, \mathrm{d}^2\boldsymbol{x}$$

$$= \sum_{\boldsymbol{p} \in \mathbb{Z}^2} X[\boldsymbol{p}] \iint_{\mathbb{R}^2} \Phi_{\boldsymbol{p}}^{(s)}(ms\boldsymbol{n} - \boldsymbol{x}) \overline{\Psi}_W(\boldsymbol{x}) \, \mathrm{d}^2\boldsymbol{x}.$$

The sum-integral interchange is possible because X has a finite support. Then:

$$\big(F_X * \overline{\Psi}_W\big)(ms\boldsymbol{n}) = \sum_{\boldsymbol{p} \in \mathbb{Z}^2} X[\boldsymbol{p}] \iint_{\mathbb{R}^2} \overline{\Psi}_W(\boldsymbol{x}) \, \Phi^{(s)}\big(s(m\boldsymbol{n} - \boldsymbol{p}) - \boldsymbol{x}\big) \, \mathrm{d}^2\boldsymbol{x} \tag{149}$$

$$= \sum_{\boldsymbol{p} \in \mathbb{Z}^2} X[\boldsymbol{p}] \big(\overline{\Psi}_W * \Phi^{(s)}\big)\big(s(m\boldsymbol{n} - \boldsymbol{p})\big) \tag{150}$$

Since $\{\Phi_{\boldsymbol{n}}^{(s)}\}_{\boldsymbol{n} \in \mathbb{Z}^2}$ is an orthonormal basis of $\mathcal{V}^{(s)}$, the definition of $\Psi_W$ in (24) implies, for any $\boldsymbol{p}' \in \mathbb{Z}^2$,

$$\overline{W}[\boldsymbol{p}'] = \big\langle \Psi_W, \, \Phi_{-\boldsymbol{p}'}^{(s)} \big\rangle = \big(\overline{\Psi}_W * \Phi^{(s)}\big)(s\boldsymbol{p}'), \tag{151}$$

because $\Phi^{(s)}$ is even. Therefore, plugging (151) with $\boldsymbol{p}' \leftarrow (m\boldsymbol{n} - \boldsymbol{p})$ into (150) yields

$$\big(F_X * \overline{\Psi}_W\big)(ms\boldsymbol{n}) = \sum_{\boldsymbol{p} \in \mathbb{Z}^2} X[\boldsymbol{p}] \overline{W}[m\boldsymbol{n} - \boldsymbol{p}] = \big(X * \overline{W}\big)[m\boldsymbol{n}], \tag{152}$$

hence the result. $\qquad \square$

### A.5 Proof of Lemma 3

*Proof.* Let $\boldsymbol{u} \in \mathbb{R}^2$. By definition of $F_{\mathcal{T}_{\boldsymbol{u}}X}$ and $\mathcal{T}_{\boldsymbol{u}}X$,

$$F_{\mathcal{T}_{\boldsymbol{u}}X} = s \sum_{\boldsymbol{n} \in \mathbb{Z}^2} \mathcal{T}_{s\boldsymbol{u}} F_X(s\boldsymbol{n}) \, \Phi_{\boldsymbol{n}}^{(s)}. \tag{153}$$

On the other hand, $F_X \in \mathcal{V}^{(s)}$ by construction. Therefore, $\mathcal{T}_{s\boldsymbol{u}} F_X \in \mathcal{V}^{(s)}$. Then, according to Shannon's sampling theorem (Mallat, 2009, Theorem 3.11, p. 81), we get

$$\mathcal{T}_{s\boldsymbol{u}} F_X = s \sum_{\boldsymbol{n} \in \mathbb{Z}^2} \mathcal{T}_{s\boldsymbol{u}} F_X(s\boldsymbol{n}) \, \Phi_{\boldsymbol{n}}^{(s)}, \tag{154}$$

which concludes the proof. $\qquad \square$

### A.6 Proof of Lemma 4

*Proof.* Let us write:

$$\sum_{\boldsymbol{n} \in \mathbb{Z}^2} \left| \mathcal{T}_{\boldsymbol{h}} F_0(s'\boldsymbol{n}) - F_0(s'\boldsymbol{n}) \right|^2 = \sum_{\boldsymbol{n} \in \mathbb{Z}^2} \left| F_1(s'\boldsymbol{n}) \right|^2 = \frac{1}{s'^2} \left\| X_1 \right\|_2^2, \tag{155}$$

where we have denoted, for any $\boldsymbol{n} \in \mathbb{Z}^2$,

$$F_1 := \mathcal{T}_{\boldsymbol{h}} F_0 - F_0 \qquad \text{and} \qquad X_1[\boldsymbol{n}] := s' F_1(s'\boldsymbol{n}). \tag{156}$$

According to Proposition 2 (25), $\Psi_{\mathrm{W}} \in \mathcal{V}(\boldsymbol{\theta}/s, \, \kappa/s)$. Therefore, according to Lemma 1,

$$\operatorname{supp} \widehat{F_0} \subset B_\infty \left( \frac{\kappa}{2s} \right). \tag{157}$$

Moreover, by hypothesis, $\kappa \leq \pi/m$; thus,

$$B_\infty \left( \frac{\kappa}{2s} \right) \subset B_\infty \left( \frac{\pi}{s'} \right), \tag{158}$$

which yields (35), and $F_1 \in \mathcal{V}^{(s')}$. Then, according to Lemma 2 (23) with $X \leftarrow X_1$, $F \leftarrow F_1$ and $s \leftarrow s'$,

$$\left\| X_1 \right\|_2 = \left\| F_1 \right\|_{L^2} = \left\| \mathcal{T}_{\boldsymbol{h}} F_0 - F_0 \right\|_{L^2}. \tag{159}$$

Therefore, plugging (159) into (155) yields (36).

Furthermore, according again to Lemma 2,

$$\left\| F_0 \right\|_{L^2}^2 = \left\| X_0 \right\|_2^2, \tag{160}$$

where we have defined, for any $\boldsymbol{n} \in \mathbb{Z}^2$,

$$X_0[\boldsymbol{n}] := s' F_0(s'\boldsymbol{n}). \tag{161}$$

Then,

$$
\begin{aligned}
\left\| X_0 \right\|_2^2 &= s'^2 \sum_{\boldsymbol{n} \in \mathbb{Z}^2} \left| \left( F_{\mathrm{X}} * \overline{\Psi}_{\mathrm{W}} \right)(s'\boldsymbol{n}) \right|^2 && \text{(acc. to (34))} \\
&= s'^2 \sum_{\boldsymbol{n} \in \mathbb{Z}^2} \left| (X * \overline{W}) \downarrow (2m)[\boldsymbol{n}] \right|^2 && \text{(acc. to Proposition 2 with } m \leftarrow 2m) \\
&= s'^2 \left\| U_m^{\mathrm{mod}} X \right\|_2^2. && \text{(acc. to (5))}
\end{aligned}
$$

Finally, plugging this result into (160) yields (37) and concludes the proof. □

### A.7 Proof of Theorem 1

*Proof.* As in Lemma 4, we consider the low-frequency function $F_0$ satisfying (34), and denote $s' := 2ms$. We can write

$$\left| F_{\mathrm{X}} * \overline{\Psi}_{\mathrm{W}} \right| = \left| F_0 \right| \qquad \text{and} \qquad \left| \mathcal{T}_{s\boldsymbol{u}} F_{\mathrm{X}} * \overline{\Psi}_{\mathrm{W}} \right| = \left| \mathcal{T}_{s\boldsymbol{u}} F_0 \right|. \tag{162}$$

Recall that $U_m^{\mathrm{mod}} X = \left| (X * \overline{W}) \downarrow (2m) \right|$, such as defined in (5). According to Proposition 2 (27) and Corollary 1 (32) with $m \leftarrow 2m$, we therefore get

$$U_m^{\mathrm{mod}} X[\boldsymbol{n}] = \left| F_0(s'\boldsymbol{n}) \right|; \tag{163}$$

$$U_m^{\mathrm{mod}} (\mathcal{T}_{\boldsymbol{u}} X)[\boldsymbol{n}] = \left| (\mathcal{T}_{s\boldsymbol{u}} F_0)(s'\boldsymbol{n}) \right|. \tag{164}$$

Then, using (163), (164) and the reverse triangle inequality,

$$\left\|U_m^{\mathrm{mod}}(\mathcal{T}_{\boldsymbol{u}}\mathrm{X}) - U_m^{\mathrm{mod}}\mathrm{X}\right\|_2^2 = \sum_{\boldsymbol{n}\in\mathbb{Z}^2} \left|\left|(\mathcal{T}_{s\boldsymbol{u}}F_0)(s'\boldsymbol{n})\right| - \left|F_0(s'\boldsymbol{n})\right|\right|^2$$

$$\leq \sum_{\boldsymbol{n}\in\mathbb{Z}^2} \left|(\mathcal{T}_{s\boldsymbol{u}}F_0)(s'\boldsymbol{n}) - F_0(s'\boldsymbol{n})\right|^2.$$

Since condition (38) is satisfied, we can use Lemma 4 (36) with $\boldsymbol{h} \leftarrow s\boldsymbol{u}$:

$$\left\|U_m^{\mathrm{mod}}(\mathcal{T}_{\boldsymbol{u}}\mathrm{X}) - U_m^{\mathrm{mod}}\mathrm{X}\right\|_2^2 \leq \frac{1}{s'^2}\left\|\mathcal{T}_{s\boldsymbol{u}}F_0 - F_0\right\|_{L^2}^2 \tag{165}$$

Next, according to Proposition 1 with $\varepsilon \leftarrow \kappa/s$ and $\boldsymbol{h} \leftarrow s\boldsymbol{u}$, we then get the following bound:

$$\left\|U_m^{\mathrm{mod}}(\mathcal{T}_{\boldsymbol{u}}\mathrm{X}) - U_m^{\mathrm{mod}}\mathrm{X}\right\|_2^2 \leq \frac{\alpha(\kappa\boldsymbol{u})^2}{s'^2}\left\|F_0\right\|_{L^2}^2. \tag{166}$$

Finally, using Lemma 4 (37) yields (39), which completes the proof. □

## A.8 Proof of Proposition 3

*Proof.* Let $\mathrm{X} \in l_{\mathbb{R}}^2(\mathbb{Z}^2)$ and $s > 0$. We consider $F_0 \in L_{\mathbb{C}}^2(\mathbb{R}^2)$ as the "low-frequency" function satisfying (34). Again, we introduce $s' := 2ms$ and $\mathrm{X}_0 \in l_{\mathbb{C}}^2(\mathbb{Z}^2)$ satisfying (161). Moreover, for any $\mathrm{Y} \in l_{\mathbb{R}}^2(\mathbb{Z}^2)$, we denote by $F_{\mathrm{Y}}^{(s')}$ the Shannon interpolation of $\mathrm{Y}$ parameterized by $s'$, analogously to (24):

$$F_{\mathrm{Y}}^{(s')} := \sum_{\boldsymbol{n}\in\mathbb{Z}^2} \mathrm{Y}[\boldsymbol{n}]\,\Phi_{\boldsymbol{n}}^{(s')}. \tag{167}$$

On the one hand, Lemma 4 provides (37). On the other hand, we seek a similar result with $\mathrm{X} \leftarrow \mathcal{T}_{\boldsymbol{u}}\mathrm{X}$. For this purpose, (164) can be rewritten

$$U_m^{\mathrm{mod}}(\mathcal{T}_{\boldsymbol{u}}\mathrm{X})[\boldsymbol{n}] = \left|\mathcal{T}_{s'\boldsymbol{u}'}F_0(s'\boldsymbol{n})\right|, \tag{168}$$

with $\boldsymbol{u}' := \boldsymbol{u}/(2m)$. Furthermore, according to Lemma 4 (35), $F_0 \in \mathcal{V}^{(s')}$. Therefore, Shannon's sampling theorem (Mallat, 2009, Theorem 3.11, p. 81) with $s \leftarrow s'$ yields

$$F_0 = s' \sum_{\boldsymbol{n}\in\mathbb{Z}^2} F_0(s'\boldsymbol{n})\,\Phi_{\boldsymbol{n}}^{(s')}$$

$$= \sum_{\boldsymbol{n}\in\mathbb{Z}^2} \mathrm{X}_0[\boldsymbol{n}]\,\Phi_{\boldsymbol{n}}^{(s')} = F_{\mathrm{X}_0}^{(s')},$$

where we have used the notations introduced in (161) and (167). Then, using Lemma 3 with $\mathrm{X} \leftarrow \mathrm{X}_0$, $\boldsymbol{u} \leftarrow \boldsymbol{u}'$ and $s \leftarrow s'$, we get

$$F_{\mathcal{T}_{\boldsymbol{u}'}\mathrm{X}_0}^{(s')} = \mathcal{T}_{s'\boldsymbol{u}'}F_{\mathrm{X}_0}^{(s')} = \mathcal{T}_{s'\boldsymbol{u}'}F_0. \tag{169}$$

Moreover, (26) (from Proposition 2) with $\mathrm{X} \leftarrow \mathcal{T}_{\boldsymbol{u}'}\mathrm{X}_0$ and $s \leftarrow s'$ becomes

$$\mathcal{T}_{\boldsymbol{u}'}\mathrm{X}_0[\boldsymbol{n}] = s'\,F_{\mathcal{T}_{\boldsymbol{u}'}\mathrm{X}_0}^{(s')}(s'\boldsymbol{n}), \tag{170}$$

and inserting (169) into (170) yields

$$\mathcal{T}_{\boldsymbol{u}'}\mathrm{X}_0[\boldsymbol{n}] = s'\,\mathcal{T}_{s'\boldsymbol{u}'}F_0(s'\boldsymbol{n}). \tag{171}$$

Therefore, (168) and (171) imply

$$\left\|U_m^{\mathrm{mod}}(\mathcal{T}_{\boldsymbol{u}}\mathrm{X})\right\|_2 = \frac{1}{s'}\left\|\mathcal{T}_{\boldsymbol{u}'}\mathrm{X}_0\right\|_2. \tag{172}$$

Moreover, since $F_0 \in \mathcal{V}^{(s')}$, and according to (171), we can use Lemma 2 with $s \leftarrow s'$, $F \leftarrow \mathcal{T}_{s'u'}F_0$ and $X \leftarrow \mathcal{T}_{u'}X_0$. We get

$$\|\mathcal{T}_{u'}X_0\|_2 = \|\mathcal{T}_{s'u'}F_0\|_{L^2} = \|F_0\|_{L^2}, \tag{173}$$

and plugging (173) into (172) yields

$$\left\|U_m^{\mathrm{mod}}(\mathcal{T}_u X)\right\|_2 = \frac{1}{s'}\|F_0\|_{L^2}. \tag{174}$$

Finally, considering Lemma 4 (37) concludes the proof. □

## A.9 Proof of Proposition 4

*Proof.* Let us write:

$$(F * \mathrm{Re}\,\overline{\Psi})(\boldsymbol{x} + \boldsymbol{h}) - \left|(F * \overline{\Psi})(\boldsymbol{x})\right| G(\boldsymbol{x}, \boldsymbol{h})$$
$$= \mathrm{Re}\left((F * \overline{\Psi})(\boldsymbol{x} + \boldsymbol{h})\right) - \left|(F * \overline{\Psi})(\boldsymbol{x})\right| \mathrm{Re}\left(\mathrm{e}^{-i\langle \boldsymbol{\nu}, \boldsymbol{h}\rangle}\,\mathrm{e}^{iH(\boldsymbol{x})}\right)$$
$$= \mathrm{Re}\left((F * \overline{\Psi})(\boldsymbol{x} + \boldsymbol{h})\right) - \mathrm{Re}\left(\left|(F * \overline{\Psi})(\boldsymbol{x})\right|\mathrm{e}^{iH(\boldsymbol{x})}\,\mathrm{e}^{-i\langle \boldsymbol{\nu}, \boldsymbol{h}\rangle}\right)$$
$$= \mathrm{Re}\left((F * \overline{\Psi})(\boldsymbol{x} + \boldsymbol{h})\right) - \mathrm{Re}\left((F * \overline{\Psi})(\boldsymbol{x})\,\mathrm{e}^{-i\langle \boldsymbol{\nu}, \boldsymbol{h}\rangle}\right)$$
$$= \mathrm{Re}\left((F * \overline{\Psi})(\boldsymbol{x} + \boldsymbol{h}) - (F * \overline{\Psi})(\boldsymbol{x})\,\mathrm{e}^{-i\langle \boldsymbol{\nu}, \boldsymbol{h}\rangle}\right).$$

Therefore,

$$\left|(F * \mathrm{Re}\,\overline{\Psi})(\boldsymbol{x} + \boldsymbol{h}) - \left|(F * \overline{\Psi})(\boldsymbol{x})\right| G(\boldsymbol{x}, \boldsymbol{h})\right| \leq \left|(F * \overline{\Psi})(\boldsymbol{x} + \boldsymbol{h}) - (F * \overline{\Psi})(\boldsymbol{x})\,\mathrm{e}^{-i\langle \boldsymbol{\nu}, \boldsymbol{h}\rangle}\right|$$
$$= \left|F_0(\boldsymbol{x} + \boldsymbol{h})\,\mathrm{e}^{-i\langle \boldsymbol{\nu}, \boldsymbol{x}+\boldsymbol{h}\rangle} - F_0(\boldsymbol{x})\,\mathrm{e}^{-i\langle \boldsymbol{\nu}, \boldsymbol{x}+\boldsymbol{h}\rangle}\right|,$$

which yields (44) and concludes the proof. □

## A.10 Proof of Lemma 5

*Proof.* We apply Proposition 4 with $\boldsymbol{h} \leftarrow \boldsymbol{h}_{\boldsymbol{n}}^{\mathrm{max}}$ and $\boldsymbol{h} \leftarrow \boldsymbol{h}'^{\mathrm{max}}_{\boldsymbol{n}}$, respectively:

$$A_{\mathrm{X}}^{\mathrm{max}}(\boldsymbol{x_n}) \leq \widetilde{A}_{\mathrm{X}}(\boldsymbol{x_n}, \boldsymbol{h}_{\boldsymbol{n}}^{\mathrm{max}}) + \left|F_0(\boldsymbol{x_n} + \boldsymbol{h}_{\boldsymbol{n}}^{\mathrm{max}}) - F_0(\boldsymbol{x_n})\right|; \tag{175}$$
$$\widetilde{A}_{\mathrm{X}}^{\mathrm{max}}(\boldsymbol{x_n}) \leq A_{\mathrm{X}}(\boldsymbol{x_n}, \boldsymbol{h}'^{\mathrm{max}}_{\boldsymbol{n}}) + \left|F_0(\boldsymbol{x_n} + \boldsymbol{h}'^{\mathrm{max}}_{\boldsymbol{n}}) - F_0(\boldsymbol{x_n})\right|. \tag{176}$$

By construction, we have, for any $\boldsymbol{p} \in \{-q, \ldots, q\}^2$,

$$\widetilde{A}_{\mathrm{X}}(\boldsymbol{x_n}, \boldsymbol{h_p}) \leq \widetilde{A}_{\mathrm{X}}^{\mathrm{max}}(\boldsymbol{x_n}) \qquad \text{and} \qquad A_{\mathrm{X}}(\boldsymbol{x_n}, \boldsymbol{h_p}) \leq A_{\mathrm{X}}^{\mathrm{max}}(\boldsymbol{x_n}). \tag{177}$$

Moreover, by definition, there exists $\boldsymbol{p}$ and $\boldsymbol{p}' \in \{-q, \ldots, q\}^2$ such that $\boldsymbol{h}_{\boldsymbol{n}}^{\mathrm{max}} = \boldsymbol{h_p}$ and $\boldsymbol{h}'^{\mathrm{max}}_{\boldsymbol{n}} = \boldsymbol{h_{p'}}$. Therefore, (175) and (176) yield, respectively,

$$A_{\mathrm{X}}^{\mathrm{max}}(\boldsymbol{x_n}) \leq \widetilde{A}_{\mathrm{X}}^{\mathrm{max}}(\boldsymbol{x_n}) + \left|F_0(\boldsymbol{x_n} + \boldsymbol{h}_{\boldsymbol{n}}^{\mathrm{max}}) - F_0(\boldsymbol{x_n})\right|; \tag{178}$$
$$\widetilde{A}_{\mathrm{X}}^{\mathrm{max}}(\boldsymbol{x_n}) \leq A_{\mathrm{X}}^{\mathrm{max}}(\boldsymbol{x_n}) + \left|F_0(\boldsymbol{x_n} + \boldsymbol{h}'^{\mathrm{max}}_{\boldsymbol{n}}) - F_0(\boldsymbol{x_n})\right|, \tag{179}$$

which yields (68) and concludes the proof. □

### A.11 Proof of Proposition 5

*Proof.* Let us write:

$$\left\| U_m^{\mathrm{mod}} \mathrm{X} - U_{m,q}^{\max} \mathrm{X} \right\|_2^2 = \sum_{\boldsymbol{n} \in \mathbb{Z}^2} \left( U_m^{\mathrm{mod}} \mathrm{X}[\boldsymbol{n}] - U_{m,q}^{\max} \mathrm{X}[\boldsymbol{n}] \right)^2$$

$$= \sum_{\boldsymbol{n} \in \mathbb{Z}^2} \left( U_m^{\mathrm{mod}} \mathrm{X}[\boldsymbol{n}] - U_m^{\mathrm{mod}} \mathrm{X}[\boldsymbol{n}] \max_{\|\boldsymbol{p}\|_\infty \leq q} G_{\mathrm{X}}(\boldsymbol{x_n}, \boldsymbol{h_p}) \right.$$

$$\left. + U_m^{\mathrm{mod}} \mathrm{X}[\boldsymbol{n}] \max_{\|\boldsymbol{p}\|_\infty \leq q} G_{\mathrm{X}}(\boldsymbol{x_n}, \boldsymbol{h_p}) - U_{m,q}^{\max} \mathrm{X}[\boldsymbol{n}] \right)^2$$

$$= \sum_{\boldsymbol{n} \in \mathbb{Z}^2} \left( \delta_{m,q} \mathrm{X}[\boldsymbol{n}] + \widetilde{A}_{\mathrm{X}}^{\max}(\boldsymbol{x_n}) - A_{\mathrm{X}}^{\max}(\boldsymbol{x_n}) \right)^2,$$

according to (60), (65) and (66). Then, using the triangle inequality, we get

$$\left\| U_m^{\mathrm{mod}} \mathrm{X} - U_{m,q}^{\max} \mathrm{X} \right\|_2 \leq \left\| \delta_{m,q} \mathrm{X} \right\|_2 + \left( \sum_{\boldsymbol{n} \in \mathbb{Z}^2} \left( \widetilde{A}_{\mathrm{X}}^{\max}(\boldsymbol{x_n}) - A_{\mathrm{X}}^{\max}(\boldsymbol{x_n}) \right)^2 \right)^{1/2}. \tag{180}$$

Furthermore, Lemma 5 yields

$$\sum_{\boldsymbol{n} \in \mathbb{Z}^2} \left( \widetilde{A}_{\mathrm{X}}^{\max}(\boldsymbol{x_n}) - A_{\mathrm{X}}^{\max}(\boldsymbol{x_n}) \right)^2 \leq \sum_{\boldsymbol{n} \in \mathbb{Z}^2} \max_{\boldsymbol{h} \in \{\boldsymbol{h_n^{\max}}, \boldsymbol{h'_n^{\max}}\}} \left| F_0(\boldsymbol{x_n} + \boldsymbol{h}) - F_0(\boldsymbol{x_n}) \right|^2 \tag{181}$$

$$\leq \sum_{\boldsymbol{n} \in \mathbb{Z}^2} \left| F_0(\boldsymbol{x_n} + \boldsymbol{h_0}) - F_0(\boldsymbol{x_n}) \right|^2, \tag{182}$$

according to Hypothesis 1. Next, since (38) is satisfied, we can use Lemma 4 (36) with $\boldsymbol{h} \leftarrow \boldsymbol{h_0}$. We get

$$\sum_{\boldsymbol{n} \in \mathbb{Z}^2} \left( \widetilde{A}_{\mathrm{X}}^{\max}(\boldsymbol{x_n}) - A_{\mathrm{X}}^{\max}(\boldsymbol{x_n}) \right)^2 \leq \frac{1}{4m^2 s^2} \left\| \mathcal{T}_{\boldsymbol{h_0}} F_0 - F_0 \right\|_{L^2}^2$$

$$\leq \alpha(\kappa \boldsymbol{h_0}/s)^2 \frac{1}{4m^2 s^2} \left\| F_0 \right\|_{L^2}^2 \qquad \text{(acc. to Proposition 1)}$$

$$= \alpha(\kappa \boldsymbol{h_0}/s)^2 \left\| U_m^{\mathrm{mod}} \mathrm{X} \right\|_2^2. \qquad \text{(acc. to Lemma 4 (37))}$$

Since, according to Hypothesis 1, $\|\boldsymbol{h_0}\|_2 = \sqrt{2}qms$, it comes that $\|\boldsymbol{h_0}\|_1 = 2qms$. Therefore,

$$\alpha(\kappa \boldsymbol{h_0}/s)^2 = \frac{\kappa^2 \|\boldsymbol{h_0}\|_1^2}{4s^2} = (qm\kappa)^2, \tag{183}$$

which yields

$$\sum_{\boldsymbol{n} \in \mathbb{Z}^2} \left( \widetilde{A}_{\mathrm{X}}^{\max}(\boldsymbol{x_n}) - A_{\mathrm{X}}^{\max}(\boldsymbol{x_n}) \right)^2 \leq \beta_q (m\kappa)^2 \left\| U_m^{\mathrm{mod}} \mathrm{X} \right\|_2^2. \tag{184}$$

Finally, plugging (184) into (180) concludes the proof. $\qquad\square$

### A.12 Proof of Proposition 7

*Proof.* We suppose that Hypothesis 3 is satisfied and we consider $\boldsymbol{x} \in \mathbb{R}^2$. For a given $n \in \mathbb{N} \setminus \{0\}$, we introduce the random variable

$$\widetilde{\mathsf{S}}_{\mathrm{X},n} := \sqrt{\sum_{\|\boldsymbol{p}\|_\infty \leq n} \mathsf{M}_{\mathrm{X}}(\boldsymbol{x_p})^2}. \tag{185}$$

According to Hypothesis 3, $\mathsf{Z}_{\mathrm{X}}(\boldsymbol{x})$ is jointly independent of $\mathsf{M}_{\mathrm{X}}(\boldsymbol{x_p})$ for $\boldsymbol{p} \in \{-n, \ldots, n\}^2$. Therefore, by composition, $\mathsf{Z}_{\mathrm{X}}(\boldsymbol{x})$ is also independent of $\widetilde{\mathsf{S}}_{\mathrm{X},n}$. Moreover, according to (83) and (111), $\widetilde{\mathsf{S}}_{\mathrm{X},n}$ converges almost surely towards $\widetilde{\mathsf{S}}_{\mathrm{X}}$, which proves independence between $\mathsf{Z}_{\mathrm{X}}(\boldsymbol{x})$ and $\widetilde{\mathsf{S}}_{\mathrm{X}}$.

Next, we prove conditional independence between $Z_X(\boldsymbol{x})$ and $M_X(\boldsymbol{x})$ given $\widetilde{S}_X$. According to Hypothesis 3,

$$\left( M_X(\boldsymbol{x}), \widetilde{S}_{X,n} \right) \perp\!\!\!\perp Z_X(\boldsymbol{x}), \tag{186}$$

where $\perp\!\!\!\perp$ stands for independence. This is because $\widetilde{S}_{X,n}$ only depends on a finite number of $M_X(\boldsymbol{x_p})$. Therefore,

$$Z_X(\boldsymbol{x}) \perp\!\!\!\perp M_X(\boldsymbol{x}) \mid \widetilde{S}_{X,n}. \tag{187}$$

Finally, since $\widetilde{S}_{X,n}$ converges almost surely towards $\widetilde{S}_X$, it comes that $Z_X(\boldsymbol{x})$ and $M_X(\boldsymbol{x})$ are conditionally independent given $\widetilde{S}_X$. $\qquad\square$

### A.13 Proof of Theorem 2

*Proof.* We consider $\boldsymbol{n} \in \mathbb{Z}^2$. By construction, $Q_X(\boldsymbol{x_n}) := 1 - G_X^{\max}(\boldsymbol{x_n})$ only depends on $Z_X(\boldsymbol{x_n})$. Therefore, under Hypothesis 3, Proposition 7 implies

$$Q_X(\boldsymbol{x_n}) \perp\!\!\!\perp M_X(\boldsymbol{x_n}) \mid \widetilde{S}_X^2 \qquad \text{and} \qquad Q_X(\boldsymbol{x_n}) \perp\!\!\!\perp \widetilde{S}_X^2. \tag{188}$$

Additionally, we introduce

$$\widetilde{\Delta}_X := \|\delta_{m,q}X\|_2, \tag{189}$$

where $\delta_{m,q}X$ is defined in (87). Then, using the linearity of $\mathbb{E}$, we get

$$\begin{aligned}
\mathbb{E}\left[ \widetilde{\Delta}_X^2 \mid \widetilde{S}_X^2 = \sigma \right] &= \sum_{\boldsymbol{n} \in \mathbb{Z}^2} \mathbb{E}\left[ \delta_{m,q}[\boldsymbol{n}]^2 \mid \widetilde{S}_X^2 = \sigma \right] \\
&= \sum_{\boldsymbol{n} \in \mathbb{Z}^2} \mathbb{E}\left[ U_{m,l}^{\mathrm{mod}}X[\boldsymbol{n}]^2 \left( 1 - G_X^{\max}(\boldsymbol{x_n}) \right)^2 \mid \widetilde{S}_X^2 = \sigma \right] \\
&= \sum_{\boldsymbol{n} \in \mathbb{Z}^2} \mathbb{E}\left[ M_X(\boldsymbol{x_n})^2 Q_X(\boldsymbol{x_n})^2 \mid \widetilde{S}_X^2 = \sigma \right] \quad \text{(acc. to (83) and (88))} \\
&= \sum_{\boldsymbol{n} \in \mathbb{Z}^2} \mathbb{E}\left[ M_X(\boldsymbol{x_n})^2 \mid \widetilde{S}_X^2 = \sigma \right] \mathbb{E}\left[ Q_X(\boldsymbol{x_n})^2 \right] \quad \text{(acc. to (188))}.
\end{aligned}$$

Using the monotone convergence theorem, we get

$$\mathbb{E}\left[ \widetilde{\Delta}_X^2 \mid \widetilde{S}_X^2 = \sigma \right] = \mathbb{E}\left[ \sum_{\boldsymbol{n} \in \mathbb{Z}^2} M_X(\boldsymbol{x_n})^2 \mid \widetilde{S}_X^2 = \sigma \right] \mathbb{E}\left[ Q_X(\boldsymbol{x_n})^2 \right]. \tag{190}$$

According to (83) and (111), we have

$$\sum_{\boldsymbol{n} \in \mathbb{Z}^2} M_X(\boldsymbol{x_n})^2 = \left\| U_m^{\mathrm{mod}}X \right\|_2^2 = \widetilde{S}_X^2. \tag{191}$$

Therefore, we get

$$\begin{aligned}
\mathbb{E}\left[ \widetilde{\Delta}_X^2 \mid \widetilde{S}_X^2 = \sigma \right] &= \mathbb{E}\left[ \widetilde{S}_X^2 \mid \widetilde{S}_X^2 = \sigma \right] \mathbb{E}\left[ Q_X(\boldsymbol{x_n})^2 \right] \\
&= \sigma \cdot \mathbb{E}\left[ Q_X(\boldsymbol{x_n})^2 \right].
\end{aligned}$$

Under Hypothesis 2, Proposition 6 yields

$$\mathbb{E}\left[ \widetilde{\Delta}_X^2 \mid \widetilde{S}_X^2 = \sigma \right] = \sigma \cdot \gamma_q(m\boldsymbol{\theta})^2. \tag{192}$$

Moreover, we can reformulate $\widetilde{Q}_X$ such as defined in (89): $\widetilde{Q}_X = \widetilde{\Delta}_X / \widetilde{S}_X$. Therefore,

$$\mathbb{E}\left[ \widetilde{Q}_X^2 \mid \widetilde{S}_X^2 = \sigma \right] = \frac{1}{\sigma} \mathbb{E}\left[ \widetilde{\Delta}_X^2 \mid \widetilde{S}_X^2 = \sigma \right] = \gamma_q(m\boldsymbol{\theta})^2. \tag{193}$$

According to (193), the conditional expected value of $\widetilde{\mathsf{Q}}_X^2$ remains the same whatever the outcome of $\widetilde{\mathsf{S}}_X^2$. Thus, the law of total expectation states that

$$\mathbb{E}\big[\widetilde{\mathsf{Q}}_X^2\big] = \mathbb{E}\left[\mathbb{E}\big[\widetilde{\mathsf{Q}}_X^2 \mid \widetilde{\mathsf{S}}_X^2\big]\right] = \gamma_q(m\boldsymbol{\theta})^2. \tag{194}$$

Since we have assumed Hypothesis 1, we can apply Proposition 5. Using the definition of $\widetilde{\mathsf{P}}_X$ (86) and $\widetilde{\mathsf{Q}}_X$ (89), we get

$$\widetilde{\mathsf{P}}_X \le \widetilde{\mathsf{Q}}_X + \beta_q(m\kappa). \tag{195}$$

Then,

$$\mathbb{E}\big[\widetilde{\mathsf{P}}_X^2\big] \le \mathbb{E}\big[\widetilde{\mathsf{Q}}_X^2\big] + 2\beta_q(m\kappa)\,\mathbb{E}\big[\widetilde{\mathsf{Q}}_X\big] + \beta_q(m\kappa)^2. \tag{196}$$

According to Jensen's inequality,

$$\mathbb{E}\big[\widetilde{\mathsf{Q}}_X\big] \le \sqrt{\mathbb{E}\big[\widetilde{\mathsf{Q}}_X^2\big]} = \gamma_q(m\boldsymbol{\theta}). \tag{197}$$

Thus,

$$\mathbb{E}\big[\widetilde{\mathsf{P}}_X^2\big] \le \gamma_q(m\boldsymbol{\theta})^2 + 2\beta_q(m\kappa)\gamma_q(m\boldsymbol{\theta}) + \beta_q(m\kappa)^2, \tag{198}$$

which yields (112). $\qquad\square$

### A.14 Proof of Lemma 6

*Proof.* First, we show that, for any $\boldsymbol{x} \in \mathbb{R}^2$,

$$\mathsf{M}_{\mathcal{T}_{\boldsymbol{u}}X}(\boldsymbol{x}) = \mathcal{T}_{s\boldsymbol{u}}\mathsf{M}_X(\boldsymbol{x}); \tag{199}$$

$$\mathsf{Z}_{\mathcal{T}_{\boldsymbol{u}}X}(\boldsymbol{x}) = \mathcal{T}_{s\boldsymbol{u}}\mathsf{Z}_X(\boldsymbol{x}). \tag{200}$$

According to Lemma 3, and since the convolution product commutes with translations, we have

$$\big(F_{\mathcal{T}_{\boldsymbol{u}}X} * \overline{\varPsi}_W\big)(\boldsymbol{x}) = \mathcal{T}_{s\boldsymbol{u}}\big(F_X * \overline{\varPsi}_W\big)(\boldsymbol{x}). \tag{201}$$

Then, using (81), the above expression becomes

$$\mathsf{M}_{\mathcal{T}_{\boldsymbol{u}}X}(\boldsymbol{x}) \times \mathsf{Z}_{\mathcal{T}_{\boldsymbol{u}}X}(\boldsymbol{x}) = (\mathcal{T}_{s\boldsymbol{u}}\mathsf{M}_X)(\boldsymbol{x}) \times (\mathcal{T}_{s\boldsymbol{u}}\mathsf{Z}_X)(\boldsymbol{x}), \tag{202}$$

which yields (199) and (200), by uniqueness of the magnitude-phase decomposition. Finally, we remind that

$$\mathcal{T}_{s\boldsymbol{u}}\mathsf{M}_X(\boldsymbol{x}) = \mathsf{M}_X(\boldsymbol{x} - s\boldsymbol{u}) \qquad \text{and} \qquad \mathcal{T}_{s\boldsymbol{u}}\mathsf{Z}_X(\boldsymbol{x}) = \mathsf{Z}_X(\boldsymbol{x} - s\boldsymbol{u}). \tag{203}$$

Then, considering hypotheses Hypotheses 2 and 3 with $\boldsymbol{x} \leftarrow \boldsymbol{x} - s\boldsymbol{u}$ yields the result. $\qquad\square$

### A.15 Proof of Theorem 3

*Proof.* Using the triangle inequality, we compute

$$\big\|U_{m,q}^{\max}(\mathcal{T}_{\boldsymbol{u}}X) - U_{m,q}^{\max}X\big\|_2$$
$$\le \big\|U_m^{\mathrm{mod}}(\mathcal{T}_{\boldsymbol{u}}X)\big\|_2 \widetilde{\mathsf{P}}_{\mathcal{T}_{\boldsymbol{u}}X} + \big\|U_m^{\mathrm{mod}}X\big\|_2 \widetilde{\mathsf{P}}_X + \big\|U_m^{\mathrm{mod}}(\mathcal{T}_{\boldsymbol{u}}X) - U_m^{\mathrm{mod}}X\big\|_2, \tag{204}$$

where $\widetilde{\mathsf{P}}_X$ and $\widetilde{\mathsf{P}}_{\mathcal{T}_{\boldsymbol{u}}X}$ are defined in (86). According to (38), we can apply Proposition 3 on the first term of (204):

$$\big\|U_m^{\mathrm{mod}}(\mathcal{T}_{\boldsymbol{u}}X)\big\|_2 = \big\|U_m^{\mathrm{mod}}X\big\|_2. \tag{205}$$

Moreover, we can apply Theorem 1 to the third term of (204):

$$\big\|U_m^{\mathrm{mod}}(\mathcal{T}_{\boldsymbol{u}}X) - U_m^{\mathrm{mod}}X\big\|_2 \le \alpha(\kappa\boldsymbol{u})\big\|U_m^{\mathrm{mod}}X\big\|_2. \tag{206}$$

We therefore get

$$\big\|U_{m,q}^{\max}(\mathcal{T}_{\boldsymbol{u}}X) - U_{m,q}^{\max}X\big\|_2 \le \Big[\widetilde{\mathsf{P}}_{\mathcal{T}_{\boldsymbol{u}}X} + \widetilde{\mathsf{P}}_X + \alpha(\kappa\boldsymbol{u})\Big]\big\|U_m^{\mathrm{mod}}X\big\|_2. \tag{207}$$

Then, by linearity of $\mathbb{E}$, we get

$$\mathbb{E}\big[\widetilde{\mathsf{R}}_{\mathrm{X},\,\boldsymbol{u}}\big] \leq \mathbb{E}\big[\widetilde{\mathsf{P}}_{\mathcal{T}_{\boldsymbol{u}}\mathrm{X}}\big] + \mathbb{E}\big[\widetilde{\mathsf{P}}_{\mathrm{X}}\big] + \alpha(\kappa\boldsymbol{u}), \tag{208}$$

where $\widetilde{\mathsf{R}}_{\mathrm{X},\,\boldsymbol{u}}$ has been introduced in (113).

For any stochastic process $\mathrm{X}'$ satisfying Hypotheses 2 and 3, Theorem 2 and Jensen's inequality yield:

$$\mathbb{E}\big[\widetilde{\mathsf{P}}_{\mathrm{X}'}\big] \leq \beta_q(m\kappa) + \gamma_q(m\boldsymbol{\theta}). \tag{209}$$

According to Lemma 6, Hypotheses 2 and 3 are also satisfied for $\mathrm{X} \leftarrow \mathcal{T}_{\boldsymbol{u}}\mathrm{X}$. Therefore, (209) is valid for both $\mathrm{X}' \leftarrow \mathrm{X}$ and $\mathrm{X}' \leftarrow \mathcal{T}_{\boldsymbol{u}}\mathrm{X}$, and plugging it into (208) concludes the proof. □

## B  Appendix – Theoretical Foundations for our Hypotheses

In this section, we provide theoretical arguments for justifying Hypotheses 2 and 3. As will be discussed, these hypotheses rely on some degree of shift invariance for input images, which implies the notions of stationarity and phase-shift-equivariance for stochastic processes.

### B.1  Stationary and Local Stationarity

Given $n \in \mathbb{N} \setminus \{0\}$, we define *n-th order stationarity* of a given stochastic process $\mathsf{F}$ as stated by Park & Park (2018, p. 152): For any $n' \leq n$, $(\boldsymbol{x}_0, \ldots, \boldsymbol{x}_{n'-1}) \in (\mathbb{R}^2)^{n'}$ and $\boldsymbol{h} \in \mathbb{R}^2$, the joint distribution of $\big(\mathsf{F}(\boldsymbol{x}_0), \ldots, \mathsf{F}(\boldsymbol{x}_{n'-1})\big)$ is identical to the one of $\big(\mathcal{T}_{\boldsymbol{h}}\mathsf{F}(\boldsymbol{x}_0), \ldots, \mathcal{T}_{\boldsymbol{h}}\mathsf{F}(\boldsymbol{x}_{n'-1})\big)$. Additionally, *strict-sense stationarity* is defined as $n$-th order stationarity for any $n \in \mathbb{N} \setminus \{0\}$.

Strict-sense stationarity suggests that any translated version of a given image is equally likely. However, this property is seldom fully achieved for images (Tygert et al., 2016). First, X has fixed boundaries. Consequently, any realization of $\mathsf{F}_{\mathrm{X}}(\boldsymbol{x})$ quickly vanishes as $\|\boldsymbol{x}\|$ tends to $\infty$. Furthermore, depending on which category the image belongs to, the pixel distribution is likely to vary across various regions. For instance, we can expect the main subject to be located at the center of the image. We refer readers to Torralba & Oliva (2003) for more details on statistical properties of images from natural versus man-made objects. For this reason, we introduce a notion of local stationarity as follows.

**Definition 1.** Given $\tau \geq 0$, a stochastic process $\mathsf{F}$ is *$\tau$-locally stationary* to the $n$-th order if, for any $0 < n'' \leq n' \leq n$, any $(\boldsymbol{x}_0, \ldots, \boldsymbol{x}_{n'-1}) \in (\mathbb{R}^2)^{n'}$, any displacement vector $\boldsymbol{h} \in \mathbb{R}^2$, and any pair of measurable sets $\mathfrak{X}, \mathfrak{Y} \subset \mathbb{R}^{n''} \times \mathbb{R}^{n'-n''}$,

$$\Bigg| \mathbb{P}\Big\{ \big(\mathcal{T}_{\boldsymbol{h}}\mathsf{F}(\boldsymbol{x}_0), \ldots, \mathcal{T}_{\boldsymbol{h}}\mathsf{F}(\boldsymbol{x}_{n''-1})\big)^{\top} \in \mathfrak{X} \,\Big|\, \big(\mathcal{T}_{\boldsymbol{h}}\mathsf{F}(\boldsymbol{x}_{n''}), \ldots, \mathcal{T}_{\boldsymbol{h}}\mathsf{F}(\boldsymbol{x}_{n'-1})\big)^{\top} \in \mathfrak{Y} \Big\}$$

$$- \mathbb{P}\Big\{ \big(\mathsf{F}(\boldsymbol{x}_0), \ldots, \mathsf{F}(\boldsymbol{x}_{n''-1})\big)^{\top} \in \mathfrak{X} \,\Big|\, \big(\mathsf{F}(\boldsymbol{x}_{n''}), \ldots, \mathsf{F}(\boldsymbol{x}_{n'-1})\big)^{\top} \in \mathfrak{Y} \Big\} \Bigg| \leq \tau \, \|\boldsymbol{h}\|_2. \tag{210}$$

Moreover, $\mathsf{F}$ is *strict-sense $\tau$-locally stationary* if this property is satisfied for any $n \in \mathbb{N}$.

**Remark 10.** The special case where $\tau = 0$ corresponds to the standard definition of $n$-th order stationarity. Additionally, the use of conditional probabilities is essential for defining local stationarity, as it helps prevent instabilities that may arise when conditioning on low-probability events.

In what follows, we consider the following narrowband stochastic process:

$$\mathsf{F}_{1,\,\mathrm{X}} : \boldsymbol{x} \mapsto (\mathsf{F}_{\mathrm{X}} * \overline{\varPsi}_{\mathrm{W}})(\boldsymbol{x}). \tag{211}$$

We assume that $\mathsf{F}_{\mathrm{X}}$, and therefore $\mathsf{F}_{1,\,\mathrm{X}}$, is nearly shift-invariant for displacement vectors that are much smaller than the image "characteristic" size in the continuous domain, which is equal to $sN$, where, as a reminder, $N \in \mathbb{N}$ denotes the support size of input images and $s > 0$ denotes the sampling interval. More formally, $\mathsf{F}_{1,\,\mathrm{X}}$ is assumed to be strict-sense $\tau$-locally stationary with

$$\tau := \frac{1}{sN}. \tag{212}$$

## B.2 Translations and Phase Shifts

We consider the following stochastic processes:

$$\mathsf{F}_{0,\mathrm{X}} : \boldsymbol{x} \mapsto (\mathsf{F}_{\mathrm{X}} * \overline{\Psi}_{\mathrm{W}})(\boldsymbol{x})\, \mathrm{e}^{i\langle \boldsymbol{\nu}, \boldsymbol{x}\rangle} \qquad \text{and} \qquad \mathsf{F}_{1,\mathrm{X}} : \boldsymbol{x} \mapsto (\mathsf{F}_{\mathrm{X}} * \overline{\Psi}_{\mathrm{W}})(\boldsymbol{x}). \tag{213}$$

To justify our hypotheses, we also need to characterize $\mathcal{T}_{\boldsymbol{h}}\mathsf{F}_{1,\mathrm{X}}$ as a function of $\mathsf{F}_{1,\mathrm{X}}$. Having:

$$\mathcal{T}_{\boldsymbol{h}}\mathsf{F}_{1,\mathrm{X}}(\boldsymbol{x}) = \mathcal{T}_{\boldsymbol{h}}\mathsf{F}_{0,\mathrm{X}}(\boldsymbol{x})\, \mathrm{e}^{-i\langle \boldsymbol{\nu}, \boldsymbol{x}\rangle}\, \mathrm{e}^{i\langle \boldsymbol{\nu}, \boldsymbol{h}\rangle}, \tag{214}$$

we get

$$\left|\mathcal{T}_{\boldsymbol{h}}\mathsf{F}_{1,\mathrm{X}}(\boldsymbol{x}) - \mathsf{F}_{1,\mathrm{X}}(\boldsymbol{x})\, \mathrm{e}^{i\langle \boldsymbol{\nu}, \boldsymbol{h}\rangle}\right| = \left|\mathcal{T}_{\boldsymbol{h}}\mathsf{F}_{0,\mathrm{X}}(\boldsymbol{x}) - \mathsf{F}_{0,\mathrm{X}}(\boldsymbol{x})\right|. \tag{215}$$

According to Lemma 1, the support of $\widehat{\mathsf{F}}_{0,\mathrm{X}}$ is contained within the ball $B_\infty\left(\frac{\kappa}{2s}\right)$. Intuitively, we can define a "minimal wavelength" $\lambda := 2\pi s/\kappa$ such that, if $\|\boldsymbol{h}\|_2 \ll \lambda$, we can approximate $\mathcal{T}_{\boldsymbol{h}}\mathsf{F}_{0,\mathrm{X}}(\boldsymbol{x}) \approx \mathsf{F}_{0,\mathrm{X}}(\boldsymbol{x})$, and therefore

$$\mathcal{T}_{\boldsymbol{h}}\mathsf{F}_{1,\mathrm{X}}(\boldsymbol{x}) \approx \mathsf{F}_{1,\mathrm{X}}(\boldsymbol{x})\, \mathrm{e}^{i\langle \boldsymbol{\nu}, \boldsymbol{h}\rangle}. \tag{216}$$

If the two terms were strictly identical for any $\boldsymbol{x} \in \mathbb{R}^2$, we would get, for any $n \in \mathbb{N}$, $(\boldsymbol{x}_0, \ldots, \boldsymbol{x}_n) \in (\mathbb{R}^2)^n$, and any measurable set $\mathfrak{X} \subset \mathbb{R}^n$,

$$\left(\mathsf{F}_{1,\mathrm{X}}(\boldsymbol{x}_0), \ldots, \mathsf{F}_{1,\mathrm{X}}(\boldsymbol{x}_{n-1})\right)^\top \in \mathfrak{A} \iff \left(\mathcal{T}_{\boldsymbol{h}}\mathsf{F}_{1,\mathrm{X}}(\boldsymbol{x}_0), \ldots, \mathcal{T}_{\boldsymbol{h}}\mathsf{F}_{1,\mathrm{X}}(\boldsymbol{x}_{n-1})\right)^\top \in \mathrm{e}^{i\langle \boldsymbol{\nu}, \boldsymbol{h}\rangle}\mathfrak{A}, \tag{217}$$

and therefore,

$$\mathbb{P}\left\{\left(\mathsf{F}_{1,\mathrm{X}}(\boldsymbol{x}_0), \ldots, \mathsf{F}_{1,\mathrm{X}}(\boldsymbol{x}_{n-1})\right)^\top \in \mathfrak{A}\right\} = \mathbb{P}\left\{\left(\mathcal{T}_{\boldsymbol{h}}\mathsf{F}_{1,\mathrm{X}}(\boldsymbol{x}_0), \ldots, \mathcal{T}_{\boldsymbol{h}}\mathsf{F}_{1,\mathrm{X}}(\boldsymbol{x}_{n-1})\right)^\top \in \mathrm{e}^{i\langle \boldsymbol{\nu}, \boldsymbol{h}\rangle}\mathfrak{A}\right\}. \tag{218}$$

Instead, we relax the above equality (218) and introduce the following definition.

**Definition 2.** Given $\tau \geq 0$, a stochastic process $\mathsf{F}$ is $\tau$-*locally phase-shift-equivariant* to the $n$-th order with respect to the frequency vector $\boldsymbol{\nu} \in \mathbb{R}^2$ if, for any $0 < n'' \leq n' \leq n$, any $(\boldsymbol{x}_0, \ldots, \boldsymbol{x}_{n'-1}) \in (\mathbb{R}^2)^{n'}$, any displacement vector $\boldsymbol{h} \in \mathbb{R}^2$, and any pair of measurable sets $\mathfrak{X}, \mathfrak{Y} \subset \mathbb{R}^{n''} \times \mathbb{R}^{n'-n''}$,

$$\left|\mathbb{P}\left\{\left(\mathcal{T}_{\boldsymbol{h}}\mathsf{F}(\boldsymbol{x}_0), \ldots, \mathcal{T}_{\boldsymbol{h}}\mathsf{F}(\boldsymbol{x}_{n''-1})\right)^\top \in \mathrm{e}^{i\langle \boldsymbol{\nu}, \boldsymbol{h}\rangle}\mathfrak{X} \,\middle|\, \left(\mathcal{T}_{\boldsymbol{h}}\mathsf{F}(\boldsymbol{x}_{n''}), \ldots, \mathcal{T}_{\boldsymbol{h}}\mathsf{F}(\boldsymbol{x}_{n'-1})\right)^\top \in \mathrm{e}^{i\langle \boldsymbol{\nu}, \boldsymbol{h}\rangle}\mathfrak{Y}\right\}\right.$$

$$\left.-\mathbb{P}\left\{\left(\mathsf{F}(\boldsymbol{x}_0), \ldots, \mathsf{F}(\boldsymbol{x}_{n''-1})\right)^\top \in \mathfrak{X} \,\middle|\, \left(\mathsf{F}(\boldsymbol{x}_{n''}), \ldots, \mathsf{F}(\boldsymbol{x}_{n'-1})\right)^\top \in \mathfrak{Y}\right\}\right| \leq \tau\|\boldsymbol{h}\|_2. \tag{219}$$

Moreover, $\mathsf{F}$ is *strict-sense $\tau$-locally phase-shift-equivariant* if this property is satisfied for any $n \in \mathbb{N}$.

In what follows, we will assume that the stochastic process $\mathsf{F}_{1,\mathrm{X}}$ is nearly phase-shift-equivariant for displacement vectors that are much smaller than the minimal wavelength $\lambda$. More formally, $\mathsf{F}_{1,\mathrm{X}}$ is assumed to be strict-sense $\tau'$-locally phase-shift-equivariant with

$$\tau' := 1/\lambda = \frac{\kappa}{2\pi s}. \tag{220}$$

## B.3 Justification for Hypothesis 2

We then consider the following proposition, which states that the probability measure of $\mathsf{Z}_{\mathrm{X}}(\boldsymbol{x})$ is nearly invariant with respect to phase shifts.

**Proposition 8.** *We assume that $\mathsf{F}_{1,\mathrm{X}}$ is $\tau$-locally stationary (Definition 1) and $\tau'$-locally phase-shift-equivariant with respect to $\boldsymbol{\nu}$ (Definition 2), both to the first order. Then, for any measurable set $\mathfrak{A} \subset \mathbb{S}^1$,*

$$\forall \omega \in [0, 2\pi], \left|\mu(\mathfrak{A}) - \mu(\mathrm{e}^{i\omega}\mathfrak{A})\right| \leq 2\pi(\tau + \tau')/\|\boldsymbol{\nu}\|_2. \tag{221}$$

*where $\mu : \mathfrak{A} \mapsto \mathbb{P}\{\mathsf{Z}_{\mathrm{X}}(\boldsymbol{x}) \in \mathfrak{A}\}$ denotes the probability measure of $\mathsf{Z}_{\mathrm{X}}(\boldsymbol{x})$.*

*Proof.* Let $\omega \in [0, 2\pi]$. We compute:

$$\left|\mu(\mathfrak{A}) - \mu(\mathrm{e}^{i\omega}\mathfrak{A})\right| = \left|\mathbb{P}\left\{\mathsf{Z}_\mathrm{X}(\boldsymbol{x}) \in \mathfrak{A}\right\} - \mathbb{P}\left\{\mathsf{Z}_\mathrm{X}(\boldsymbol{x}) \in \mathrm{e}^{i\omega}\mathfrak{A}\right\}\right|$$

$$= \left|\mathbb{P}\left\{\mathsf{Z}_\mathrm{X}(\boldsymbol{x}) \in \mathfrak{A}\right\} - \mathbb{P}\left\{\mathsf{Z}_\mathrm{X}(\boldsymbol{x}) \in \mathrm{e}^{i\langle\boldsymbol{\nu},\,\boldsymbol{h}\rangle}\mathfrak{A}\right\}\right|,$$

where we have denoted $\boldsymbol{h} := \omega\,\boldsymbol{\nu}/\|\boldsymbol{\nu}\|_2^2$. Then, using the triangle inequality,

$$\left|\mu(\mathfrak{A}) - \mu(\mathrm{e}^{i\omega}\mathfrak{A})\right| \le \left|\mathbb{P}\left\{\mathsf{Z}_\mathrm{X}(\boldsymbol{x}) \in \mathfrak{A}\right\} - \mathbb{P}\left\{\mathcal{T}_{\boldsymbol{h}}\mathsf{Z}_\mathrm{X}(\boldsymbol{x}) \in \mathrm{e}^{i\langle\boldsymbol{\nu},\,\boldsymbol{h}\rangle}\mathfrak{A}\right\}\right|$$

$$+ \left|\mathbb{P}\left\{\mathsf{Z}_\mathrm{X}(\boldsymbol{x}) \in \mathrm{e}^{i\langle\boldsymbol{\nu},\,\boldsymbol{h}\rangle}\mathfrak{A}\right\} - \mathbb{P}\left\{\mathcal{T}_{\boldsymbol{h}}\mathsf{Z}_\mathrm{X}(\boldsymbol{x}) \in \mathrm{e}^{i\langle\boldsymbol{\nu},\,\boldsymbol{h}\rangle}\mathfrak{A}\right\}\right|$$

$$= \left|\mathbb{P}\left\{\mathsf{F}_{1,\,\mathrm{X}}(\boldsymbol{x}) \in u^{-1}(\mathfrak{A})\right\} - \mathbb{P}\left\{\mathcal{T}_{\boldsymbol{h}}\mathsf{F}_{1,\,\mathrm{X}}(\boldsymbol{x}) \in u^{-1}\!\left(\mathrm{e}^{i\langle\boldsymbol{\nu},\,\boldsymbol{h}\rangle}\mathfrak{A}\right)\right\}\right|$$

$$+ \left|\mathbb{P}\left\{\mathsf{F}_{1,\,\mathrm{X}}(\boldsymbol{x}) \in u^{-1}\!\left(\mathrm{e}^{i\langle\boldsymbol{\nu},\,\boldsymbol{h}\rangle}\mathfrak{A}\right)\right\} - \mathbb{P}\left\{\mathcal{T}_{\boldsymbol{h}}\mathsf{F}_{1,\,\mathrm{X}}(\boldsymbol{x}) \in u^{-1}\!\left(\mathrm{e}^{i\langle\boldsymbol{\nu},\,\boldsymbol{h}\rangle}\mathfrak{A}\right)\right\}\right|,$$

where we have denoted $u : z \mapsto \mathrm{e}^{i\angle(z)}$. We can easily show that $u^{-1}\!\left(\mathrm{e}^{i\langle\boldsymbol{\nu},\,\boldsymbol{h}\rangle}\mathfrak{A}\right) = \mathrm{e}^{i\langle\boldsymbol{\nu},\,\boldsymbol{h}\rangle}u^{-1}(\mathfrak{A})$. Therefore, the above expression can be re-written:

$$\left|\mu(\mathfrak{A}) - \mu(\mathrm{e}^{i\omega}\mathfrak{A})\right| \le \left|\mathbb{P}\left\{\mathsf{F}_{1,\,\mathrm{X}}(\boldsymbol{x}) \in \mathfrak{X}\right\} - \mathbb{P}\left\{\mathcal{T}_{\boldsymbol{h}}\mathsf{F}_{1,\,\mathrm{X}}(\boldsymbol{x}) \in \mathrm{e}^{i\langle\boldsymbol{\nu},\,\boldsymbol{h}\rangle}\mathfrak{X}\right\}\right|$$

$$+ \left|\mathbb{P}\left\{\mathsf{F}_{1,\,\mathrm{X}}(\boldsymbol{x}) \in \mathfrak{X}'\right\} - \mathbb{P}\left\{\mathcal{T}_{\boldsymbol{h}}\mathsf{F}_{1,\,\mathrm{X}}(\boldsymbol{x}) \in \mathfrak{X}'\right\}\right|, \quad (222)$$

where we have denoted $\mathfrak{X} := u^{-1}(\mathfrak{A})$ and $\mathfrak{X}' := u^{-1}\!\left(\mathrm{e}^{i\langle\boldsymbol{\nu},\,\boldsymbol{h}\rangle}\mathfrak{A}\right)$. Then, by applying (210) and (219) with $n' = 1$ to (222), we get:

$$\left|\mu(\mathfrak{A}) - \mu(\mathrm{e}^{i\omega}\mathfrak{A})\right| \le (\tau + \tau')\,\|\boldsymbol{h}\|_2. \quad (223)$$

Finally, using the definition of $\boldsymbol{h}$ and bounding $\omega$ by $2\pi$ yields the result. $\qquad\square$

If we replace $\tau$ and $\tau'$ in (221) by their respective values in (212) and (220), we get,

$$\forall \omega \in [0, 2\pi],\ \left|\mu(\mathfrak{A}) - \mu(\mathrm{e}^{i\omega}\mathfrak{A})\right| \le \frac{2\pi}{sN} \times \|\boldsymbol{\nu}\|_2^{-1} + \frac{\kappa}{s} \times \|\boldsymbol{\nu}\|_2^{-1}. \quad (224)$$

We recall that $\boldsymbol{\nu} := \boldsymbol{\theta}/s$. Then, by applying the constraints stated in (84) and (85), we get

$$\forall \omega \in [0, 2\pi],\ \left|\mu(\mathfrak{A}) - \mu(\mathrm{e}^{i\omega}\mathfrak{A})\right| \ll 1. \quad (225)$$

Therefore, $\mu$ is almost invariant to phase shifts. Since the only probability measure satisfying the phase-shift invariance property is the uniform probability measure,[3] we deduce that $\mathsf{Z}_\mathrm{X}(\boldsymbol{x})$ follows a near-uniform distribution on $\mathbb{S}^1$. For the sake of simplicity, in Hypothesis 2 we have assumed a strictly-uniform distribution.

## B.4 Justification for Hypothesis 3

Let $n \in \mathbb{N} \setminus \{0\}$ and $\boldsymbol{x}, \boldsymbol{y}_0, \ldots, \boldsymbol{y}_{n-1} \in \mathbb{R}^2$. To simplify notations, we consider the random vector

$$\mathbf{M} := \left(\mathsf{M}_\mathrm{X}(\boldsymbol{y}_0), \ldots, \mathsf{M}_\mathrm{X}(\boldsymbol{y}_{n-1})\right)^\top \quad (226)$$

with outcomes in $\mathbb{R}_+^n$. This section is organized as follows. Using reasoning similar to that in Proposition 8, we show that, for any measurable subset $\mathfrak{S} \subset \mathbb{R}_+^n$, $\mathsf{Z}_\mathrm{X}$ follows a near-uniform probability distribution conditionally to $\mathbf{M} \in \mathfrak{S}$. Since we already assumed that $\mathsf{Z}_\mathrm{X}$ follows an unconditional uniform distribution, we deduce that $\mathsf{Z}_\mathrm{X}$ and $\mathbf{M}$ are nearly independent.

---

[3]Any probability measure defined on $\mathbb{S}^1$ is a Radon measure. Therefore, according to Haar's theorem (Halmos, 2013), there exists a unique probability measure on $\mathbb{S}^1$ satisfying the phase-shift invariance property, and it turns out that the uniform probability measure is one such candidate.

**Proposition 9.** *We assume that* $\mathsf{F}_{1,\mathrm{X}}$ *is* $\tau$*-locally stationary (Definition 1) and* $\tau'$*-locally phase-shift-equivariant with respect to* $\boldsymbol{\nu}$ *(Definition 2), both in the strict sense. Then, for any measurable sets* $\mathfrak{A} \subset \mathbb{S}^1$ *and* $\mathfrak{S} \subset \mathbb{R}^n_+$,

$$\forall \omega \in [0, 2\pi], \; \left|\mu_{\mathfrak{S}}(\mathfrak{A}) - \mu_{\mathfrak{S}}(\mathrm{e}^{i\omega}\mathfrak{A})\right| \leq 2\pi(\tau + \tau')/\left\|\boldsymbol{\nu}\right\|_2. \tag{227}$$

*where* $\mu_{\mathfrak{S}} : \mathfrak{A} \mapsto \mathbb{P}\left\{\mathsf{Z}_{\mathrm{X}}(\boldsymbol{x}) \in \mathfrak{A} \mid \mathbf{M} \in \mathfrak{S}\right\}$ *denotes the conditional probability measure of* $\mathsf{Z}_{\mathrm{X}}(\boldsymbol{x})$.

*Proof.* Let $\omega \in [0, 2\pi]$. We compute:

$$\left|\mu_{\mathfrak{S}}(\mathrm{e}^{i\omega}\mathfrak{A}) - \mu_{\mathfrak{S}}(\mathfrak{A})\right| = \left|\mathbb{P}\{\mathsf{Z}_{\mathrm{X}}(\boldsymbol{x}) \in \mathrm{e}^{i\omega}\mathfrak{A} \mid \mathbf{M} \in \mathfrak{S}\} - \mathbb{P}\{\mathsf{Z}_{\mathrm{X}}(\boldsymbol{x}) \in \mathfrak{A} \mid \mathbf{M} \in \mathfrak{S}\}\right|$$

$$= \left|\mathbb{P}\{\mathsf{Z}_{\mathrm{X}}(\boldsymbol{x}) \in \mathrm{e}^{i\langle\boldsymbol{\nu},\boldsymbol{h}\rangle}\mathfrak{A} \mid \mathbf{M} \in \mathfrak{S}\} - \mathbb{P}\{\mathsf{Z}_{\mathrm{X}}(\boldsymbol{x}) \in \mathfrak{A} \mid \mathbf{M} \in \mathfrak{S}\}\right|,$$

where we have denoted

$$\boldsymbol{h} := \omega\,\boldsymbol{\nu}/\left\|\boldsymbol{\nu}\right\|_2^2 \tag{228}$$

Then, using the triangle inequality,

$$\left|\mu_{\mathfrak{S}}(\mathrm{e}^{i\omega}\mathfrak{A}) - \mu_{\mathfrak{S}}(\mathfrak{A})\right|$$
$$\leq \left|\mathbb{P}\{\mathcal{T}_{\boldsymbol{h}}\mathsf{Z}_{\mathrm{X}}(\boldsymbol{x}) \in \mathrm{e}^{i\langle\boldsymbol{\nu},\boldsymbol{h}\rangle}\mathfrak{A} \mid \mathcal{T}_{\boldsymbol{h}}\mathbf{M} \in \mathfrak{S}\} - \mathbb{P}\{\mathsf{Z}_{\mathrm{X}}(\boldsymbol{x}) \in \mathrm{e}^{i\langle\boldsymbol{\nu},\boldsymbol{h}\rangle}\mathfrak{A} \mid \mathbf{M} \in \mathfrak{S}\}\right|$$
$$+ \left|\mathbb{P}\{\mathcal{T}_{\boldsymbol{h}}\mathsf{Z}_{\mathrm{X}}(\boldsymbol{x}) \in \mathrm{e}^{i\langle\boldsymbol{\nu},\boldsymbol{h}\rangle}\mathfrak{A} \mid \mathcal{T}_{\boldsymbol{h}}\mathbf{M} \in \mathfrak{S}\} - \mathbb{P}\{\mathsf{Z}_{\mathrm{X}}(\boldsymbol{x}) \in \mathfrak{A} \mid \mathbf{M} \in \mathfrak{S}\}\right|, \tag{229}$$

where we have denoted

$$\mathcal{T}_{\boldsymbol{h}}\mathbf{M} := \left(\mathcal{T}_{\boldsymbol{h}}\mathsf{M}_{\mathrm{X}}(\boldsymbol{y}_0), \, \ldots, \, \mathcal{T}_{\boldsymbol{h}}\mathsf{M}_{\mathrm{X}}(\boldsymbol{y}_{n-1})\right)^\top. \tag{230}$$

Let us split this expression for the sake of readability. The first term after the $\leq$ sign can be equivalently written as follows:

$$\left|\mathbb{P}\{\mathcal{T}_{\boldsymbol{h}}\mathsf{Z}_{\mathrm{X}}(\boldsymbol{x}) \in \mathrm{e}^{i\langle\boldsymbol{\nu},\boldsymbol{h}\rangle}\mathfrak{A} \mid \mathcal{T}_{\boldsymbol{h}}\mathbf{M} \in \mathfrak{S}\} - \mathbb{P}\{\mathsf{Z}_{\mathrm{X}}(\boldsymbol{x}) \in \mathrm{e}^{i\langle\boldsymbol{\nu},\boldsymbol{h}\rangle}\mathfrak{A} \mid \mathbf{M} \in \mathfrak{S}\}\right|$$
$$= \left|\mathbb{P}\{\mathcal{T}_{\boldsymbol{h}}\mathsf{F}_{1,\mathrm{X}}(\boldsymbol{x}) \in \mathfrak{X}' \mid \left(\mathcal{T}_{\boldsymbol{h}}\mathsf{F}_{1,\mathrm{X}}(\boldsymbol{y}_0), \, \ldots, \, \mathcal{T}_{\boldsymbol{h}}\mathsf{F}_{1,\mathrm{X}}(\boldsymbol{y}_{n-1})\right)^\top \in \mathfrak{Y}\}\right.$$
$$\left.- \mathbb{P}\{\mathsf{F}_{1,\mathrm{X}}(\boldsymbol{x}) \in \mathfrak{X}' \mid \left(\mathsf{F}_{1,\mathrm{X}}(\boldsymbol{y}_0), \, \ldots, \, \mathsf{F}_{1,\mathrm{X}}(\boldsymbol{y}_{n-1})\right)^\top \in \mathfrak{Y}\}\right|, \tag{231}$$

where we have denoted

$$\mathfrak{X}' := u^{-1}\left(\mathrm{e}^{i\langle\boldsymbol{\nu},\boldsymbol{h}\rangle}\mathfrak{A}\right) \qquad \text{and} \qquad \mathfrak{Y} := \boldsymbol{v}^{-1}(\mathfrak{S}), \tag{232}$$

with

$$u : z \mapsto \mathrm{e}^{i\angle(z)} \qquad \text{and} \qquad \boldsymbol{v} : (z_0, \, \ldots, \, z_{n-1})^\top \mapsto (|z_0|, \, \ldots, \, |z_{n-1}|)^\top. \tag{233}$$

Therefore, according the local stationarity hypothesis, we apply (210) to (231), which yields

$$\left|\mathbb{P}\{\mathcal{T}_{\boldsymbol{h}}\mathsf{Z}_{\mathrm{X}}(\boldsymbol{x}) \in \mathrm{e}^{i\langle\boldsymbol{\nu},\boldsymbol{h}\rangle}\mathfrak{A} \mid \mathcal{T}_{\boldsymbol{h}}\mathbf{M} \in \mathfrak{S}\} - \mathbb{P}\{\mathsf{Z}_{\mathrm{X}}(\boldsymbol{x}) \in \mathrm{e}^{i\langle\boldsymbol{\nu},\boldsymbol{h}\rangle}\mathfrak{A} \mid \mathbf{M} \in \mathfrak{S}\}\right| \leq \tau\left\|\boldsymbol{h}\right\|_2. \tag{234}$$

Next, the second term after the $\leq$ sign in (229) can be equivalently written as follows:

$$\left|\mathbb{P}\{\mathcal{T}_{\boldsymbol{h}}\mathsf{Z}_{\mathrm{X}}(\boldsymbol{x}) \in \mathrm{e}^{i\langle\boldsymbol{\nu},\boldsymbol{h}\rangle}\mathfrak{A} \mid \mathcal{T}_{\boldsymbol{h}}\mathbf{M} \in \mathfrak{S}\} - \mathbb{P}\{\mathsf{Z}_{\mathrm{X}}(\boldsymbol{x}) \in \mathfrak{A} \mid \mathbf{M} \in \mathfrak{S}\}\right|$$
$$= \left|\mathbb{P}\{\mathcal{T}_{\boldsymbol{h}}\mathsf{F}_{1,\mathrm{X}}(\boldsymbol{x}) \in u^{-1}\left(\mathrm{e}^{i\langle\boldsymbol{\nu},\boldsymbol{h}\rangle}\mathfrak{A}\right) \mid \left(\mathcal{T}_{\boldsymbol{h}}\mathsf{F}_{1,\mathrm{X}}(\boldsymbol{y}_0), \, \ldots, \, \mathcal{T}_{\boldsymbol{h}}\mathsf{F}_{1,\mathrm{X}}(\boldsymbol{y}_{n-1})\right)^\top \in \boldsymbol{v}^{-1}(\mathfrak{S})\}\right.$$
$$\left.- \mathbb{P}\{\mathsf{F}_{1,\mathrm{X}}(\boldsymbol{x}) \in u^{-1}(\mathfrak{A}) \mid \left(\mathsf{F}_{1,\mathrm{X}}(\boldsymbol{y}_0), \, \ldots, \, \mathsf{F}_{1,\mathrm{X}}(\boldsymbol{y}_{n-1})\right)^\top \in \boldsymbol{v}^{-1}(\mathfrak{S})\}\right|. \tag{235}$$

As explained in the proof of Proposition 8, we have

$$u^{-1}\left(\mathrm{e}^{i\langle\boldsymbol{\nu},\boldsymbol{h}\rangle}\mathfrak{A}\right) = \mathrm{e}^{i\langle\boldsymbol{\nu},\boldsymbol{h}\rangle}u^{-1}(\mathfrak{A}). \tag{236}$$

Moreover, we can easily show that

$$\boldsymbol{v}^{-1}(\boldsymbol{\mathfrak{S}}) = \mathrm{e}^{i\langle \boldsymbol{\nu}, \boldsymbol{h}\rangle} \boldsymbol{v}^{-1}(\boldsymbol{\mathfrak{S}}). \tag{237}$$

Therefore, (235) can be re-written:

$$\left| \mathbb{P}\left\{ \mathcal{T}_{\boldsymbol{h}} \mathsf{Z}_{\mathrm{X}}(\boldsymbol{x}) \in \mathrm{e}^{i\langle \boldsymbol{\nu}, \boldsymbol{h}\rangle}\mathfrak{A} \mid \mathcal{T}_{\boldsymbol{h}}\mathsf{M} \in \boldsymbol{\mathfrak{S}} \right\} - \mathbb{P}\left\{ \mathsf{Z}_{\mathrm{X}}(\boldsymbol{x}) \in \mathfrak{A} \mid \mathsf{M} \in \boldsymbol{\mathfrak{S}} \right\} \right|$$
$$= \left| \mathbb{P}\left\{ \mathcal{T}_{\boldsymbol{h}}\mathsf{F}_{1,\,\mathrm{X}}(\boldsymbol{x}) \in \mathrm{e}^{i\langle \boldsymbol{\nu}, \boldsymbol{h}\rangle}\mathfrak{X} \mid \left(\mathcal{T}_{\boldsymbol{h}}\mathsf{F}_{1,\,\mathrm{X}}(\boldsymbol{y}_0), \,\dots,\, \mathcal{T}_{\boldsymbol{h}}\mathsf{F}_{1,\,\mathrm{X}}(\boldsymbol{y}_{n-1})\right)^{\top} \in \mathrm{e}^{i\langle \boldsymbol{\nu}, \boldsymbol{h}\rangle}\boldsymbol{\mathfrak{Y}} \right\} \right.$$
$$\left. - \mathbb{P}\left\{ \mathsf{F}_{1,\,\mathrm{X}}(\boldsymbol{x}) \in \mathfrak{X} \mid \left(\mathsf{F}_{1,\,\mathrm{X}}(\boldsymbol{y}_0), \,\dots,\, \mathsf{F}_{1,\,\mathrm{X}}(\boldsymbol{y}_{n-1})\right)^{\top} \in \boldsymbol{\mathfrak{Y}} \right\} \right|, \quad (238)$$

where we have denoted

$$\mathfrak{X} := u^{-1}(\mathfrak{A}) \qquad \text{and} \qquad \boldsymbol{\mathfrak{Y}} := \boldsymbol{v}^{-1}(\boldsymbol{\mathfrak{S}}). \tag{239}$$

Therefore, according the local phase-shift equivariance hypothesis, we apply (219) to (238), which yields

$$\left| \mathbb{P}\left\{ \mathcal{T}_{\boldsymbol{h}} \mathsf{Z}_{\mathrm{X}}(\boldsymbol{x}) \in \mathrm{e}^{i\langle \boldsymbol{\nu}, \boldsymbol{h}\rangle}\mathfrak{A} \mid \mathcal{T}_{\boldsymbol{h}}\mathsf{M} \in \boldsymbol{\mathfrak{S}} \right\} - \mathbb{P}\left\{ \mathsf{Z}_{\mathrm{X}}(\boldsymbol{x}) \in \mathfrak{A} \mid \mathsf{M} \in \boldsymbol{\mathfrak{S}} \right\} \right| \le \tau' \left\| \boldsymbol{h} \right\|_2. \tag{240}$$

Finally, plugging (234) and (240) into (229), using the definition of $\boldsymbol{h}$ in (228), and bounding $\omega$ by $2\pi$, yields the result. $\qquad\square$

By applying the constraints sated in (84) and (85), we get (see Appendix B.3),

$$\forall \omega \in [0,\, 2\pi], \; \left| \mu_{\boldsymbol{\mathfrak{S}}}(\mathfrak{A}) - \mu_{\boldsymbol{\mathfrak{S}}}(\mathrm{e}^{i\omega}\mathfrak{A}) \right| \ll 1. \tag{241}$$

Therefore, $\mathsf{Z}_{\mathrm{X}}(\boldsymbol{x})$ follows a near-uniform conditional distribution on $\mathbb{S}^1$ given $\mathsf{M} \in \boldsymbol{\mathfrak{S}}$.

Furthermore, according to Appendix B.3, $\mathsf{Z}_{\mathrm{X}}(\boldsymbol{x})$ also follows a near-uniform unconditional distribution. Therefore, we get, for any measurable sets $\mathfrak{A} \subset \mathbb{S}^1$ and $\boldsymbol{\mathfrak{S}} \subset \mathbb{R}_+^n$,

$$\mathbb{P}\left\{ \mathsf{Z}_{\mathrm{X}}(\boldsymbol{x}) \in \mathfrak{A} \mid (\mathsf{M} \in \boldsymbol{\mathfrak{S}}) \right\} \approx \mathbb{P}\left\{ \mathsf{Z}_{\mathrm{X}}(\boldsymbol{x}) \in \mathfrak{A} \right\}. \tag{242}$$

We deduce that $\mathsf{Z}_{\mathrm{X}}(\boldsymbol{x})$ and $\mathsf{M}$ are nearly independent. For the sake of simplicity, in Hypothesis 3 we have assumed strict independence.

# C  Appendix – Details on DT-$\mathbb{C}$WPT

A description of the transform itself is provided in Appendix C.1. Then, Appendix C.2 shows that DT-$\mathbb{C}$WPT performs convolutions with a subsampling factor $m_J$ which depends on the decomposition depth $J$. Finally, the Gabor-like nature of the convolution kernels is established in Appendix C.3.

## C.1  Background

We provide a brief overview of the classical, real-valued 2D wavelet packet transform (WPT) algorithm (Mallat, 2009, p. 377), before introducing the redundant, complex-valued and oriented DT-$\mathbb{C}$WPT (Bayram & Selesnick, 2008).

### C.1.1  Discrete Wavelet Packet Transform

Given a pair of low- and high-pass 1D orthogonal filters h, g $\in l_{\mathbb{R}}^2(\mathbb{Z})$ satisfying a *quadrature mirror filter* (QMF) relationship, we consider a separable 2D filter bank (FB), denoted by $\mathbf{G} := (\mathsf{G}_l)_{l \in \{0,\,\dots,\,3\}}$, defined by

$$\mathsf{G}_0 = \mathrm{h} \otimes \mathrm{h}; \qquad\quad \mathsf{G}_1 = \mathrm{h} \otimes \mathrm{g}; \qquad\quad \mathsf{G}_2 = \mathrm{g} \otimes \mathrm{h}; \qquad\quad \mathsf{G}_3 = \mathrm{g} \otimes \mathrm{g}. \tag{243}$$

Let $X \in l^2_{\mathbb{R}}(\mathbb{Z})$. The decomposition starts with $D_0^{(0)} = X$. Given $j \in \mathbb{N}$, suppose that we have computed $4^j$ sequences of wavelet packet coefficients at stage $j$, denoted by $D_l^{(j)} \in l^2_{\mathbb{R}}(\mathbb{Z})$ for each $l \in \{0, \dots, 4^j - 1\}$. They are referred to as *feature maps*.

At stage $j + 1$, we compute a new representation of $X$ with increased frequency resolution—and decreased spatial resolution. It is obtained by further decomposing each feature map $D_l^{(j)}$ into four sub-sequences, using subsampled (or strided) convolutions with kernels $G_k$, for each $k \in \{0, \dots, 3\}$:

$$\forall k \in \{0, \dots, 3\}, \; D_{4l+k}^{(j+1)} = \left(D_l^{(j)} * \overline{G_k}\right) \downarrow 2. \tag{244}$$

The algorithm stops after reaching the desired number of stages $J > 0$—referred to as *decomposition depth*. Then,

$$\mathbf{D}^{(J)} := \left(D_l^{(J)}\right)_{l \in \{0, \dots, 4^J - 1\}} \tag{245}$$

constitutes a multichannel representation of $X$ in an orthonormal basis, from which the original image can be retrieved.

### C.1.2 Dual-Tree Complex Wavelet Packet Transform

Despite having interesting properties such as sparse signal representation, WPT is unstable with respect to small shifts and suffers from a poor directional selectivity. To overcome this, Kingsbury (2001) designed a new type of discrete wavelet transform, where images are decomposed in a redundant frame of nearly-analytic, complex-valued waveforms. It was later extended to the wavelet packet framework by Bayram & Selesnick (2008). The latter operation, referred to as *dual-tree complex wavelet packet transform* (DT-$\mathbb{C}$WPT), is performed as follows.

Let $(h^{[0]}, g^{[0]})$ and $(h^{[1]}, g^{[1]})$ denote two pairs of QMFs as defined in Appendix C.1.1, satisfying the *half-sample delay* condition:

$$\forall \omega \in [-\pi, \pi], \; \widehat{h}^{[1]}(\omega) = e^{-i\omega/2}\, \widehat{h}^{[0]}(\omega). \tag{246}$$

Then, for any $k \in \{0, \dots, 3\}$, we build a 2D FB $\mathbf{G}_k := (G_{k,l})_{l \in \{0, \dots, 3\}}$ similarly to (243):

$$G_{k,0} = h_i \otimes h_j; \qquad G_{k,1} = h_i \otimes g_j; \qquad G_{k,2} = g_i \otimes h_j; \qquad G_{k,3} = g_i \otimes g_j, \tag{247}$$

where $i, j \in \{0, 1\}$ are defined such that $k = 2 \times i + j$.[4]

Let $J > 0$ denote a decomposition depth. Using each of the four FBs $\mathbf{G}_{0-3}$ as defined above, we assume that we have decomposed an input image $X$ into four multichannel WPT representations $\mathbf{D}_{0-3}^{(J)}$, each of which satisfies (244) and (245). Then, for any $l \in \{0, \dots, 4^J - 1\}$, the following complex feature maps are computed:

$$\begin{pmatrix} D_l^{\nearrow(J)} \\ D_l^{\searrow(J)} \end{pmatrix} = \begin{pmatrix} 1 & -1 \\ 1 & 1 \end{pmatrix} \begin{pmatrix} D_l^{[0](J)} \\ D_l^{[3](J)} \end{pmatrix} - i \begin{pmatrix} 1 & 1 \\ 1 & -1 \end{pmatrix} \begin{pmatrix} D_l^{[2](J)} \\ D_l^{[1](J)} \end{pmatrix}. \tag{248}$$

As explained in Appendix C.3, the feature maps of dual-tree coefficients have their Fourier transform restricted to a compact region of the frequency plane, and as such can be considered as Gabor-like coefficients. In the above expression, the arrow points to the Fourier quadrant where energy is concentrated. Furthermore, in the specific case where input images are real-valued, $D_l^{\swarrow(J)}$ and $D_l^{\nwarrow(J)}$ are defined as the complex conjugates of the above feature maps, and therefore do not need to be explicitly computed. Then,

$$\mathbf{D}^{(J)} := \left(D_l^{\nearrow(J)}, D_l^{\searrow(J)}, D_l^{\swarrow(J)}, D_l^{\nwarrow(J)}\right)_{l \in \{0, \dots, 4^J - 1\}} \tag{249}$$

constitutes a complex-valued, four-time redundant multichannel representation of $X$ from which the original image can be reconstructed.

---

[4]Actually, the FB design requires some technicalities which are not described here.

## C.2 Convolution Operators

We now show that DT-$\mathbb{C}$WPT performs subsampled convolutions with Gabor-like filters, whose characteristics will be specified. First, we state the following lemma concerning the real-valued WPT algorithm, such as introduced in Appendix C.1.1. It is a simple reformulation of the well-known result that two successive convolutions can be written as another convolution with a wider kernel.

**Lemma 7.** *For any $l \in \left\{0, \ldots, 4^J - 1\right\}$, there exists $\mathrm{V}_l^{(J)} \in l_{\mathbb{R}}^2(\mathbb{Z}^2)$ such that*

$$\mathrm{D}_l^{(J)} = \left(\mathrm{X} * \overline{\mathrm{V}}_l^{(J)}\right) \downarrow 2^J. \tag{250}$$

*Proof.* We introduce the upsampling operator: $(\mathrm{X} \uparrow m)[\boldsymbol{n}] := \mathrm{X}[\boldsymbol{n}/m]$ if $\boldsymbol{n}/m \in \mathbb{Z}^2$, and 0 otherwise. We also consider the "identity" filter $\mathrm{I} \in l_{\mathbb{R}}^2(\mathbb{Z}^2)$ such that $\mathrm{I}[\boldsymbol{0}] = 1$ and $\mathrm{I}[\boldsymbol{n}] = 0$ otherwise. First, for any $\mathrm{U}, \mathrm{V} \in l_{\mathbb{R}}^2(\mathbb{Z}^2)$ and any $s, t \in \mathbb{N}^*$, we have

$$((\mathrm{U} \downarrow s) * \mathrm{V}) \downarrow t = (\mathrm{U} * (\mathrm{V} \uparrow s)) \downarrow (st). \tag{251}$$

Then, a simple reasoning by induction yields the result, with

$$\mathrm{V}_0^{(0)} := \mathrm{I}; \qquad \mathrm{V}_{4l+k}^{(j+1)} := \mathrm{V}_l^{(j)} * \left(\mathrm{G}_k \uparrow 2^j\right) \tag{252}$$

for any $l \in \{0, \ldots, j-1\}$ and any $k \in \{0, \ldots, 3\}$. $\qquad\square$

Based on Lemma 7, the following proposition introduces complex kernels characterizing DT-$\mathbb{C}$WPT.

**Proposition 10.** *For any $l \in \left\{0, \ldots, 4^J - 1\right\}$, there exists $\mathrm{W}_l^{\nearrow(J)} \in l_{\mathbb{C}}^2(\mathbb{Z}^2)$ such that (132) is satisfied. Identical results are obtained with the three other Fourier quadrants.*

*Proof.* For each of the four filter banks $m \in \{0, \ldots, 3\}$, and any channel $l \in \left\{0, \ldots, 4^J - 1\right\}$, Lemma 7 provides a convolution kernel $\mathrm{V}_l^{[m](J)} \in l_{\mathbb{R}}^2(\mathbb{Z}^2)$ such that

$$\mathrm{D}_l^{[m](J)} = \left(\mathrm{X} * \overline{\mathrm{V}}_l^{[m](J)}\right) \downarrow 2^J. \tag{253}$$

Then, the result is obtained by plugging (253) into (248) for all $m \in \{0, \ldots, 3\}$, and by denoting

$$\begin{pmatrix} \mathrm{W}^{\nearrow(J)}_l \\ \mathrm{W}^{\searrow(J)}_l \end{pmatrix} = \begin{pmatrix} 1 & -1 \\ 1 & 1 \end{pmatrix} \begin{pmatrix} \mathrm{V}_l^{[0](J)} \\ \mathrm{V}_l^{[3](J)} \end{pmatrix} + i \begin{pmatrix} 1 & 1 \\ 1 & -1 \end{pmatrix} \begin{pmatrix} \mathrm{V}_l^{[2](J)} \\ \mathrm{V}_l^{[1](J)} \end{pmatrix}. \tag{254}$$

$\qquad\square$

**Remark 11.** DT-$\mathbb{C}$WPT, computed on a discrete image X, approximates the decomposition of a continuous 2D signal $F \in L_{\mathbb{R}}^2(\mathbb{R}^2)$ into a tight frame

$$\boldsymbol{\Psi}_{\mathbb{C}}^{(J)} := \biguplus_{l=0}^{4^J-1} \left(\Psi_{l,\boldsymbol{n}}^{\nearrow(J)}, \Psi_{l,\boldsymbol{n}}^{\searrow(J)}, \Psi_{l,\boldsymbol{n}}^{\swarrow(J)}, \Psi_{l,\boldsymbol{n}}^{\nwarrow(J)}\right)_{\boldsymbol{n}\in\mathbb{Z}^2}. \tag{255}$$

In this context, the feature maps of dual-tree wavelet packet coefficients satisfy

$$\mathrm{D}_l^{\nearrow(J)}[\boldsymbol{n}] \approx \left(F * \overline{\Psi}_l^{\nearrow(J)*}\right)(2^J\boldsymbol{n}), \qquad \text{with} \qquad \Psi_l^{\nearrow(J)} := \Psi_{l,\boldsymbol{0}}^{\nearrow(J)}. \tag{256}$$

Expression (256) is only an approximation because of implementation technicalities that occur in practice. A "perfect" dual-tree transform should be initialized with four different inputs $\mathrm{X}^{[0-3]}$. Instead, all four WPT decompositions operate on the same input image X, leading to non-analytic outputs for small values of $J$. In order to counterbalance this shortcoming, the first stage of DT-$\mathbb{C}$WPT decomposition must be performed with a special set of filters that satisfy the *one-sample delay* condition. We refer to Selesnick et al. (2005) for more details on this matter.

### C.3 Gabor-Like Convolution Kernels

In this section, we show that the convolution kernels $W_l^{\nearrow(J)}$ and $W_l^{\searrow(J)}$, introduced in (132), approximately behave as Gabor-like filters, as defined in (11). To begin with, we assume that $h^{[0]}$ is a Shannon filter, which is associated with a sinc scaling function (Shannon, 1949). Let $J \in \mathbb{N} \setminus \{0\}$ denote the number of decomposition stages. The following proposition states that DT-$\mathbb{C}$WPT tiles the frequency plane with square windows.

**Proposition 11.** *There exists a permutation* $\big(\boldsymbol{\sigma}_l^{(J)}\big)_{l \in \{0, \dots, 4^J - 1\}}$ *of* $\big\{0, \dots, 2^J - 1\big\}^2$ *such that, for any* $l \in \big\{0, \dots, 4^J - 1\big\}$,

$$\Psi_l^{\nearrow(J)} \in \mathcal{V}\big(\boldsymbol{\theta}_l^{(J)}, \kappa_J\big), \tag{257}$$

*where* $\Psi_l^{\nearrow(J)}$ *has been introduced in Remark 11, and where we have defined*

$$\boldsymbol{\theta}_l^{(J)} := \left(\boldsymbol{\sigma}_l^{(J)} + \frac{1}{2}\right)\frac{\pi}{2^J} \qquad and \qquad \kappa_J := \frac{\pi}{2^J}. \tag{258}$$

*We remind the reader that* $\mathcal{V}\big(\boldsymbol{\nu}, \varepsilon\big)$, *defined in* (8), *denotes a space of Gabor-like filters in the continuous framework.*

*Proof.* The atoms $\Psi_l^{\nearrow(J)}$ of the wavelet packet tight frame $\boldsymbol{\Psi}_{\mathbb{C}}^{(J)}$ can be written as the tensor product of two 1D wavelet packets:

$$\Psi_l^{\nearrow(J)} = \psi_{l_1}^{(J)} \otimes \psi_{l_2}^{(J)}, \tag{259}$$

for some indices $l_1$ and $l_2 \in \big\{0, \dots, 2^J - 1\big\}$. Moreover, for any $l' \in \big\{0, \dots, 2^J - 1\big\}$, we have

$$\psi_{l'}^{(J)} = \psi_{l'}^{[0](J)} + i\,\psi_{l'}^{[1](J)}, \tag{260}$$

where $\psi_{l'}^{[0](J)} \in L_{\mathbb{R}}^2(\mathbb{R})$ is an atom of the standard Shannon wavelet packet orthonormal basis, and $\psi_{l'}^{[1](J)}$ is the one-dimensional Hilbert transform of $\psi_{l'}^{[0](J)}$. Therefore, since the Hilbert transform suppresses negative frequencies, we get

$$\widehat{\psi}_{l'}^{(J)} = 2\,\widehat{\psi}_{l'}^{[0](J)}\,\mathbb{1}_{\mathbb{R}_+}. \tag{261}$$

Consequently, according to the Coifman-Wickerhauser theorem (Mallat, 2009, pp. 384-385), there exists $k \in \big\{0, \dots, 2^J - 1\big\}$ such that

$$\operatorname{supp}\widehat{\psi}_{l'}^{(J)} \subset \left[\frac{k\pi}{2^J}, \frac{(k+1)\pi}{2^J}\right]. \tag{262}$$

Finally, the tensor product (259) yields the result. $\qquad\square$

According to Proposition 11, each atom $\Psi_l^{\nearrow(J)}$, for $l \in \big\{0, \dots, 4^J - 1\big\}$, is supported in a square window of size $\kappa_J \times \kappa_J$ included in the top-right quadrant of the Fourier domain. Similar results can be obtained for the three remaining quadrants, with $\Psi_l^{\searrow(J)}$, $\Psi_l^{\swarrow(J)}$ and $\Psi_l^{\nwarrow(J)}$. We would like to deduce from Proposition 11 that the discrete filter $W_l^{\nearrow(J)} \in l_{\mathbb{C}}^2(\mathbb{Z}^2)$ satisfies the Gabor property (133). However, as mentioned in Remark 11, (256) is only an approximation. In fact, the Fourier support of $W_l^{\nearrow(J)}$ is contained in four square regions of size $\kappa_J$ (one in each quadrant), its energy becoming negligible outside the top-right quadrant when $J$ increases. Nevertheless, employing, in the first stage, a specific pair of low-pass filters satisfying the one-sample delay condition (Selesnick et al., 2005) yields near-analytic solutions even for small values of $J$. We therefore assume that (133) is a reasonable approximation if $J \geq 2$.

**Remark 12.** Proposition 11 tiles the top-right Fourier quadrant with $4^J$ square cells of size $\kappa_J := \pi/2^J$. However, the Shannon wavelet is poorly suited for sparse image representations, because of its slow decay rate. Moreover, it deviates from what is typically observed in freely-trained CNNs, because $W_l^{\nearrow(J)}$ must be approximated with very large filters to avoid numerical instabilities. Practical implementations of DT-$\mathbb{C}$WPT use fast-decaying filters such as these associated to Meyer wavelets (Meyer, 1985), or finite-length filters that approximate the half-sample delay condition (Selesnick et al., 2005). Therefore, energy is leaking

| Depth $J$ | Bandwidth $\kappa_J$ | Mean | Std |
|---|---|---|---|
| 2 | $\pi/2$ | 0.98 | 0.00 |
| 3 | $\pi/4$ | 0.95 | 0.02 |

Table 1. Energy concentration of the DT-$\mathbb{C}$WPT filters within a Fourier window of size $\kappa_J \times \kappa_J$, with $\kappa_J := \pi/2^{J-1}$.

outside the square cells tiling the Fourier domain. To counterbalance this, we increase the window size up to

$$\kappa_J := \frac{\pi}{2^{J-1}} = \pi/m_J, \tag{263}$$

and suggest that (133) remains a reasonable approximation. Therefore, the conditions to apply Theorems 1 to 3 are approximately satisfied in this context.

In order to numerically assess this assumption, we measured the maximum percentage of energy within a square window of size $\kappa_J \times \kappa_J$ in the Fourier domain:

$$\rho_l^{\nearrow} := \frac{\max_{\boldsymbol{\theta} \in [-\pi,\,\pi]^2} \left\| \mathbb{1}_{B_\infty(\boldsymbol{\theta},\,\kappa_J/2)} \widehat{\mathrm{W}}_l^{\nearrow(J)} \right\|_{L^2}^2}{\left\| \widehat{\mathrm{W}}_l^{\nearrow(J)} \right\|_{L^2}^2}, \tag{264}$$

where the $l^\infty$-ball $B_\infty(\boldsymbol{\theta},\,\kappa_J/2)$ is defined in the quotient space $[-\pi,\,\pi]^2/(2\pi\mathbb{Z}^2)$, as explained in Remark 2. If (133) is perfectly satisfied, then $\rho_l^{\nearrow} = 1$. The statistics computed over the collection $\left(\rho_l^{\nearrow},\,\rho_l^{\searrow}\right)_{l \in \{0,\,\ldots,\,4^J-1\}}$ are reported in table 1.

**Remark 13.** For "boundary filters", *i.e.*, when $\left\| \boldsymbol{\theta}_l^{(J)} \right\|_\infty = \left(1 - 2^{-(J+1)}\right)\pi$, Remark 2 states that a small fraction of the filter's energy remains located at the far end of the Fourier domain—see also Bayram & Selesnick (2008). Therefore, these filters do not strictly comply with the conditions of Theorems 1 to 3. We nevertheless include them in our experiments.

