# OpenReview forum: "On the Shift Invariance of Max Pooling Feature Maps in Convolutional Neural Networks"
_TMLR — Rejected by TMLR_

### Review · Reviewer_YRpJ · 2025-04-06

**Summary Of Contributions:**

The authors clarify in how far a real convolution followed by max pooling (Rmax) gives similar outputs to a complex convolution followed by a modulus operation (Cmod), when the convolutional filters are Gabor-like. This is relevant because Cmod is known to be invariant to small shifts of the input image in this setting. In particular, the results are extended to discrete convolutions with subsampling, and to some degree with multiple channels, which is the practically relevant case in the first layers of CNNs.

**Audience:**

Yes

**Broader Impact Concerns:**

/

**Claims And Evidence:**

Yes

**Requested Changes:**

- The abbreviation DT-CWPT is explicitly introduced on page 4, but it is already used on p.2
- In the outline sect. 1.3, I was missing a direct connection to the terminology of stride in the CNN. Is my understanding correct that the subsampling factor $m$ corresponds to the stride of the convolution, and the stride of the max-pooling operation is fixed to 2 (while the size of the max-pooling window is $q$) ?
- Why is a specific function $\beta_q$ defined in eq.72 if simply $\beta_q(k) = qk$ (and similarly the definition of $\alpha$ in eq.18)? This seems to save neither space nor increase clarity.
- Typo in Hyp.5: "Monochorome"

**Strengths And Weaknesses:**

## Strengths

- Clear presentation
- Rigorous results
- In-depth analysis of the connection between Rmax and Cmod

## Weaknesses

- Applies to filters localized in frequency domain (Gabor-like). However, this is only the case in the very first layers of CNNs.
- The experimental results in sect.6.3, figs. 8+9, are not quantitatively compared against theoretical bounds.
- Limitations of the results are not discussed explicitly.

In sum, the manuscript clearly written and provides practically relevant insights on the shift invariance of discrete convolution filters with max-pooling. I have found no issues which would require significant revisions.
Therefore I believe that the manuscript can be published with minimal changes, if also the other referees do not identify significant technical flaws.

Please note that I did not validate the proofs.

Below a few minor questions and comments are listed.

---

> ### Author Response · Authors · 2025-05-20
> **Response to Reviewer YRpJ**
>
> We sincerely thank the reviewer for their positive and constructive feedback. We appreciate their recognition of the clarity, rigor, and relevance of our study. Below we address the few minor concerns raised.
>
> **Scope and Assumptions**
>
> > “Applies to filters localized in frequency domain (Gabor-like). However, this is only the case in the very first layers of CNNs.”
>
> We fully agree that frequency-localized filters—such as Gabor-like waveforms—primarily occur in the first layer of CNNs. This observation is consistent with empirical findings in earlier work and forms the basis for our focus. We emphasize, however, that instabilities in the first layer can have significant influence on the overall behavior of the network. In our previous work (anonymized version provided as supplementary material), for instance, we showed experimentally that replacing the standard RMax operator by CMod in the first layer alone has a visible stabilizing effect throughout the network (in particular, see Fig. 2). Therefore, our work provides a partial but significant understanding of shift invariance properties in CNNs. We revised our manuscript to highlight this point more explicitly.
>
> **Comparison Between Theory and Experiments**
>
> > “The experimental results in sect.6.3, figs. 8+9, are not quantitatively compared against theoretical bounds.”
>
> We agree that a quantitative comparison to theoretical predictions would further strengthen the work. While a full quantitative analysis would be demanding due to the complexity of the operators involved, we point out that the empirical plots in Figures 8 and 9 remarkably correlate with the theoretical results displayed in Figure 5, especially regarding the frequency-dependent behavior of the operators. We have included this qualitative agreement in the discussion to better emphasize the consistency between theory and experiments.
>
> **Discussion of Limitations**
>
> > “Limitations of the results are not discussed explicitly.”
>
> We thank the reviewer for this suggestion. We have added a dedicated paragraph in the conclusion to discuss limitations, including the focus on Gabor-like filters and the restriction to the first layer. While we believe the theoretical insights provided are nonetheless significant and of interest to the community, we agree that these limitations should be clearly acknowledged.
>
> **Minor Revisions and Clarifications**
>
> > “The abbreviation DT-CWPT is explicitly introduced on page 4, but it is already used on p.2”
>
> We thank the reviewer for catching this. We have moved the definition earlier so that it is introduced on first use.
>
> > “In the outline sect. 1.3, I was missing a direct connection to the terminology of stride in the CNN. Is my understanding correct that the subsampling factor corresponds to the stride of the convolution, and the stride of the max-pooling operation is fixed to 2 (while the size of the max-pooling window is)?”
>
> Yes, that is correct. The subsampling factor corresponds to the stride, and the max-pooling stride is fixed to 2, which is standard practice in many CNN architectures. Our analysis actually provides a theoretical motivation for this particular configuration.
>
> > “Why is a specific function defined in eq.72 if simply (and similarly the definition of in eq.18)? This seems to save neither space nor increase clarity.”
>
> These definitions were introduced to improve traceability across intermediate steps and to clarify the origin of various terms arising in our results.
>
> > “Typo in Hyp. 5: ‘Monochorome’”
>
> Corrected—thank you.
>
> We again thank the reviewer for their thoughtful and constructive feedback. We have implemented all minor corrections and added the clarifications and discussion points as suggested. We hope the revised manuscript meets your expectations.

---

### Review · Reviewer_9oEX · 2025-04-29

**Summary Of Contributions:**

The paper revisits the shift-invariance properties of Complex Modulus (CMod) operators that were proposed initially in ScatterNets and complex-valued convolutional networks (Tygert, 2016). As CMod have been shown to not be applicable in practice (by the lack of conventional, freely-trained parameters), they propose what is called Real-Max-Pooling (RMax) operator. Assuming restrictions on filter localization (κ) and pooling grid size (q) (Hypotheses 1-3), the paper transfer near shift-invariance property established for CMod (in Section 2) to the RMax operator (in Section 4). The proposed operator only computes the real part
of the dual-tree coefficients.

The experiments on section 6 measure the discrepancy (MSE) between RMax and CMod operators implemented with DT-CWPT filters, and applied to imagenet images converted to grayscale.

**Audience:**

No

**Broader Impact Concerns:**

The practical and theoretical significance of the contributions is not straightforward. The paper findings are limited to strong constraints on the filters modeled that do not reflect any of the trends and large number of investigations around CNNs architecture design.

The main motivation for the paper are references pointing that convolution kernels in the first layer resemble band-pass oriented waveforms (Yosinski et al., 2014; Rai & Rivas, 2020). Since the release of such references, CNN architecture models have shifted to smaller filters and reduced use of max-pooling. Naturally the aliasing investigation moved to covering cumulative effects and considering not only its impact on first but also intermediary layers. Models like ConvNext (v1 and v2) reduced/eliminated the use of max-pooling in favor of “patchfied” subsampling and other aggregate operators.

Overall, it is not clear how the paper contributes to theoretical or practical challenges when learning shift-invariant CNN architectures.

**Claims And Evidence:**

No

**Requested Changes:**

The authors argue that its use on real-world architectures has been published as a conference paper, but the supplementary material experiments only cover small and outdated alexnet and resnet (34) models. I suggest mentioning the specific model tested.

It is hard to judge the contribution of the paper in comparison to existing work as the paper explores none of the following:
-  alternative references investigating Gabor-based formulations and comparative evaluations;
- the integration of the operator on realistic CNNs architectures;
- the effect of aliasing and lack of shift invariance on intermediary layers and the role of non-linearities other than max-pooling.

A detailed review and comparison in at least one of the 3 would clarify the paper impact of the contributions.

The experimental settings are quite hard to replicate and be used to verify the theoretical claims. I suggest adding clarifications on the learnable parameters introduced and metrics that reflect spectral difference (and not MSE only).


The paper misses other references in the area, especially when it comes to recent advances in CNNs architectures, and the impact of shift-invariance / aliasing, and modeling Gabor filters. I listed some here, but I suggest updating the reference section.
- Aliasing on CNNs:
Frequency-Adaptive Dilated Convolution for Semantic Segmentation. Chen et al (CVPR 2024)
Alias-Free Generative Adversarial Networks. Karras et al  (NeurIPS 2021)
 Impact of Aliasing on Generalization in Deep Convolutional Networks. Vasconcelos et al. (ICCV2021)
- SOTA CNNs
A ConvNet for the 2020s. Liu at al. (CVPR 2022)
ConvNeXt V2: Co-designing and Scaling ConvNets with Masked Autoencoders. Woo et al. 2023


At the end of section 1.1, the authors mention a connection between findings and hybrid architectures (transformers and convolutions), but do not extend the argument to justify if the strong assumptions of the formulation actually limit the practical extension of the contributions into the cited architectures. I suggest revolving the paragraph completely, or describing the practical limitations of applying the findings to VIT-like architectures. If kept, I also suggest adding Swim-transformers family models as an important reference of hybrid architecture.

Reference for hybrid architecture:

Swin Transformer: Hierarchical Vision Transformer using Shifted Windows. Ze Liu et al (CVPR 2022).

**Strengths And Weaknesses:**

The strength of the paper consists in revisiting Complex Modulus (CMod) operators that are found in ScatterNets and complex-valued convolutional networks (Tygert, 2016)  (analyzed in Section 2 for its near shift-invariance, but not applicable in practice) and proposing the Real-Max-Pooling (RMax) operator. THe RMax is proposed as an alternative, and is implemented in the first layer of CNNs, assumed as defined as a convolution followed by max pooling.
The key idea from there, is to assume the restricted case in which the filter Ψ is well-localized in frequency (small bandwidth ε), then the magnitude |F1(x)| changes slowly around a point x, while the phase H(x) can vary more rapidly. In this setting, the paper proposed the RMax operator that approximately computes the CMod output when restricted to settings in which the filter models a small bandwidth, with appropriate pooling region size relative to frequency ν and bandwidth ε (as per eq. 48).


My major concern is that the paper makes a number of significant strong assumptions that do not necessarily reflect realistic aliasing and shift invariance problems of CNNs used in practice. It also does not investigate the effects of combining the proposed operator in a deep model.
Adding to that it does not explicitly address the effect of aliasing on intermediary (deeper) layers of CNN. Even if perfect stability is incorporated by non-discretized filters on the first layer (for instance by the use of closed-form representation of Gabor filters with learned parameters), that does not solve the aliasing problem on CNNs.
The authors correctly argue that wavelet and multi-resolution analysis are built
upon a well-established mathematical framework. But, in my understanding the paper does not help in closing the gap between the existent solid mathematical framework  and the use of non-linearities and deep and discrete layers on practical settings.

Finally the experimental section is very brief and uses a single metric MSE for comparing CMOD and RMax operators. In practice MSE is highly sensitive to lower parts of the spectrum that concentrates most of the signal energy. For that reason, may hide mid and higher bands difference between the two.

---

> ### Author Response · Authors · 2025-05-20
> **Response to Reviewer 9oEX (1/2)**
>
> We thank the reviewer for their careful reading of our manuscript and for raising important concerns. We address the main points below, with clarifications and proposed revisions to strengthen the manuscript.
>
> **Precisions regarding the intent of the paper**
>
> > “The strength of the paper consists in revisiting Complex Modulus (CMod) operators [...] and proposing the Real-Max-Pooling (RMax) operator. The RMax is proposed as an alternative, and is implemented in the first layer of CNNs [...].”
> “[The paper] does not investigate the effects of combining the proposed operator in a deep model.”
>
> To avoid misunderstanding, we clarify that the goal of this paper is not to propose the Real-Max-Pooling (RMax) operator as an alternative to the Complex Modulus (CMod) operator, but to conduct a theoretical study of RMax—which is already widely used in CNNs—through analogy with CMod. We aim to understand the conditions under which RMax can inherit shift-invariance properties similar to CMod, and more importantly, when it fails to do so. Accordingly, we did not explore the effect of integrating RMax in deep models—it already exists in standard architectures. Instead, our focus is on analyzing its contribution to the overall network (in)stability. In our companion paper however, we do study the effect of replacing RMax by CMod in deep models, and observe that this leads to improved stability and prediction accuracy, in accordance with our theoretical analysis.
>
> **On the use of assumptions**
>
> > “My major concern is that the paper makes a number of significant strong assumptions that do not necessarily reflect realistic aliasing and shift invariance problems of CNNs used in practice.”
>
> While we indeed made a certain number of assumptions for mathematical tractability, these are grounded in empirical observations. For example, the hypothesis of frequency-localized filters aligns with evidence that first-layer filters in CNNs often resemble Gabor-like functions. We acknowledge that some assumptions may oversimplify practical architectures, but they offer a solid framework to understand some behaviors that are truly observed in CNNs. For instance, there is a remarkable correlation between the empirical results presented in Figure 9 and the theoretical results displayed in Figure 5. Our experiments empirically confirm predictions derived from the theoretical framework, notably that stability properties vary with filter frequency, supporting the practical relevance of our assumptions.
>
> **Scope of the paper and role of the first layer**
>
> Our study deliberately focuses on the first layer of CNNs (convolution + max pooling), motivated by both theoretical considerations and empirical evidence. This layer actually plays a crucial role in the network’s overall stability. In our companion paper, we experimentally confirmed that instabilities can propagate through layers: Fig. 2 (supplementary material) shows that replacing RMax by CMod in the first layer alone tends to stabilize network outputs. We agree that aliasing and shift sensitivity in deeper layers are important issues, but they require a different analytical approach that falls beyond the scope of this work. Nevertheless, we believe that our paper provides a partial but significant understanding of shift invariance properties in CNNs, which is grounded in wavelet and image processing theory.
>
> **Use of MSE Metric and Frequency Bands**
>
> > “MSE is highly sensitive to lower parts of the spectrum... may hide mid and higher bands difference.”
>
> We respectfully point out that this is a misunderstanding. As shown in Figures 8 and 9, we compute MSE separately for each frequency band, precisely to avoid this issue. Moreover, we exclude the lowest-frequency bands from the analysis because they fall outside the scope of this work. We have revised the caption and text to make this methodological point more explicit and avoid confusion.

---

> ### Author Response · Authors · 2025-05-20
> **Response to Reviewer 9oEX (2/2)**
>
> **Lack of Comparison to Existing Work**
>
> > “A detailed review and comparison in at least one of the 3 would clarify the paper impact.”
>
> Our work addresses a question that, to the best of our knowledge, has not been explored so far, especially regarding the interaction between real-valued convolution, max pooling, and shift invariance under specific (but practically motivated) assumptions. Existing work about shift invariance can be found in the literature, and has been cited in our related work section. However, most papers focus on the continuous framework, overlooking aliasing effects that appear when discretizing and subsampling. While exploring some of the alternative directions mentioned by the reviewer are interesting, pursuing them would have diluted the theoretical focus of the paper. We hope this sharp focus offers clarity and mathematical insight. Nevertheless, we agree that it is beneficial to cite related efforts about aliasing on CNNs as well as recent architectural advances. We have updated the related work section and reference list to include several of the suggested works.
>
> **Connection to Hybrid Architectures and Transformers**
>
> We agree that this part of the discussion can be clarified. Our intent was not to claim direct applicability to all transformer-based models, but to suggest that stability in early convolutional layers might remain relevant even in hybrid CNN-transformer settings. We have revised the paragraph to clarify this point, and included additional references to recent hybrid models as suggested.
>
> **Practical and Theoretical Relevance**
>
> > “It is not clear how the paper contributes to theoretical or practical challenges when learning shift-invariant CNNs. [...] Since the release of such references, CNN architecture models have shifted to smaller filters and reduced use of max-pooling.”
>
> Our contribution is primarily theoretical, but with practical implications. By characterizing when and how a real-valued convolution + max pooling setup can approximate the shift-invariance of a complex modulus operator, we offer a principled basis for designing CNNs with better stability properties, as shown in our companion paper. Even if certain components like large convolution kernels and max pooling are less fashionable in current SOTA pipelines, we believe that a theoretical study primarily focused on legacy architectures that have been proven successful in the past can offer new understanding of their underlying mechanisms, guide architectural choices and inspire future research directions. As TMLR’s call for papers highlights, even modest contributions grounded in careful theory can benefit the community.
>
> We thank the reviewer again for these valuable comments. We believe that the revised manuscript more clearly articulates the motivation, scope, and implications of our work.

---

### Review · Reviewer_uJt4 · 2025-05-09

**Summary Of Contributions:**

The authors study shift invariance properties of a convolutional layer with max pooling. They specifically study the connection between real valued filters with unimodal spectra ("Gabor-like" filters) under a max pool operator and associated analytic filters under a modulus operator. They establish that the two behave similarly when the downsampling factor of the real valued filter is reduced. They test their claims on the dual-tree complex wavelet packet transform.

**Audience:**

No

**Broader Impact Concerns:**

-

**Claims And Evidence:**

Yes

**Requested Changes:**

I think the main missing ingredient is motivation. I don't see why the readers would be interested in such a lengthy treatment on the invariance properties of the first layer of a CNN. I would strongly suggest demonstrating the impact of the work on a real and practical problem. I don't think the authors have to train something to beat whatever is the state of the art in some problem. But it is a win if the outcomes of the paper can be shown to facilitate training (e.g., maybe make data augmentation unnecessary, which in turn reduces training time), or some other practical issue.

Below are minor comments :
- Page 1, "absence of this property, known as translation invariance" : please reword as it's the property not the absence of it that is "translation invariance".

- Page 1, "consequently, a reliable model must learn this property through trainin on ... large datasets" : if this property can be already learned via training in practive, what is the contribution of this paper?

- Page 2 : please briefly describe what you mean by Gabor-like filters, and Gabor-like coefficients.

- Page 2 : please highlight that your study is restricted to a single CNN layer.

- Page 3 : "continuous framework" : continuous in what? (please either point to Section 2, or briefly state what this is).

- Page 4 : Why do you flip the filter and resort to convolution instead of direct cross correlation?

- Page 4 : What is the "Gabor hypothesis"?

- Page 4 : Please expand the discussion on "Waldspurger's idea". What is the connection between the real and complex wavelets?

- Section 2, "The primary goal of this paper is to ... establish conditions for near shift invariance at the output of the first max pooling layer" : this is a very modest claim. If CNNs are known to already have this property in practice, what does the paper contribute to the field?

**Strengths And Weaknesses:**

The paper tackles a challenging and important problem, which is translation invariance in CNNs. However, what it establishes is very modest, in contrast to the length of the technical development. The paper also does not really show how the results would be useful in practice.

---

> ### Author Response · Authors · 2025-05-20
> **Response to Reviewer utj4 (1/2)**
>
> We thank the reviewer for their comments. We appreciate their recognition that our paper addresses a fundamental and challenging problem—shift invariance in convolutional neural networks (CNNs). We would like to take this opportunity to clarify the motivation and contributions of our work, and to explain how we believe it offers value to the research community, even within its theoretical scope.
>
> **Motivation and Relevance of the Study**
>
> > “I don't see why the readers would be interested in such a lengthy treatment on the invariance properties of the first layer of a CNN.”
>
> We acknowledge that our study is restricted to the first convolutional layer and relies on specific assumptions (e.g., Gabor-like filters). However, we believe that understanding the behavior of this layer is essential for both theoretical and practical reasons. In our own prior work (included as supplementary material), we showed that instability introduced at early stages can propagate through the network and affect deeper representations. Moreover, filters learned in the first layer often resemble Gabor-like waveforms, which are effective for capturing high-frequency structures such as edges or textures, but are known to be prone to aliasing, with a negative impact on shift invariance. This observation motivated our investigation into how pooling operations can help mitigate aliasing effects without discarding critical high-frequency information.
>
> Our aim is therefore not to claim a new state-of-the-art result, but to offer a principled and rigorous characterization of a widely used component in CNNs—real-valued convolution followed by max pooling. We believe such foundational work is especially valuable in a field where empirical innovations often precede theoretical understanding.
> To address concerns about the length of the technical development, we have adjusted the manuscript structure to better guide readers through the core results without necessarily delving into all technical details.
>
> **Practical Impact**
>
> > “The paper also does not really show how the results would be useful in practice.”
>
> We would like to emphasize that the practical relevance of our theoretical findings is demonstrated in our companion paper, where we show that replacing RMax with CMod improves both stability and prediction accuracy. However, we acknowledge that broader applicability remains constrained by technical challenges, such as training complex-valued filters while preserving a Gabor-like structure. In both the present study and the companion work, we addressed this limitation by using deterministic filters derived from the dual-tree complex wavelet packet transform (DT-CWPT), which allowed us to empirically validate the theoretical insights. That said, this work should be viewed as a proof of concept rather than a turnkey architectural proposal. Moreover, while not a primary focus of the paper, we note that using fixed filters in place of fully trained ones has the potential to reduce model complexity and training time.
> Our work therefore contributes to a deeper understanding of how and why certain design choices affect stability—offering insights that could guide future developments in architecture design and training strategies.
>
> **On the Modesty of the Claims**
>
> > “Page 1, ‘consequently, a reliable model must learn this property through training on [...] large datasets’ : if this property can be already learned via training in practice, what is the contribution of this paper?”
> “Section 2, ‘The primary goal of this paper is to [...] establish conditions for near shift invariance at the output of the first max pooling layer’ : this is a very modest claim. If CNNs are known to already have this property in practice, what does the paper contribute to the field?”
>
> Desirable properties such as shift invariance in deep learning models can be achieved either through extensive training and data augmentation, or through careful architectural design. Relying on the former may result in increased computational cost, reduced generalizability, and limited interpretability. In contrast, understanding when and why a model exhibits such properties by design provides a principled path toward more efficient and robust architectures. The contribution of our paper is to analyze to what extent existing CNNs exhibit shift invariance, and under what conditions this behavior can be expected. In particular, we show that models containing real-valued convolutions followed by max pooling possess near shift invariance when first-layer convolution kernels are Gabor-like, except at specific filter frequencies regularly scattered in the Fourier plane. This result, to our knowledge, has not been previously formalized and provides new insight into the spectral dependencies of RMax stability. Our theoretical analysis thus reveals underexplored aspects of commonly used or legacy architectures, with the goal of informing and guiding future research.

---

> ### Author Response · Authors · 2025-05-20
> **Response to Reviewer utj4 (2/2)**
>
> > “What it establishes is very modest, in contrast to the length of the technical development.”
>
> We understand this concern and acknowledge that our study only provides a partial understanding of shift invariance in CNNs, under specific assumptions that are nonetheless grounded in empirical observation—albeit with simplifications for mathematical tractability. However, our analysis revealed phenomena that, to our knowledge, had not been previously documented. Notably,  we showed that the stability of the RMax operator is highly frequency-dependent, as predicted by our theoretical analysis in Figure 5 and strongly supported by the empirical results in Figures 8 and 9. The complexity of the theoretical developments reflects the technical challenge of rigorously linking real and complex-valued CNN operations in a discrete setting that incorporates subsampling.
> As emphasized in TMLR’s acceptance guidelines, papers need not make bold claims to be valuable. Modest but well-supported contributions—particularly those that improve theoretical understanding—can be of significant interest to a subset of the machine learning community. We believe our paper meets this criterion, offering mathematically grounded conclusions that are reinforced by empirical findings.
>
> **Minor Points**
>
> We appreciate the reviewer’s detailed minor comments and have revised the manuscript accordingly. Below, we provide clarifications on a few specific points.
>
> > “Page 4 : Why do you flip the filter and resort to convolution instead of direct cross correlation?”
>
> We used standard convolution instead of cross-correlation to take advantage of its mathematical properties—most notably, that the Fourier transform of a convolution between two functions or vectors equals the product of their Fourier transforms.
>
> > “Page 4 : Please expand the discussion on ‘Waldspurger's idea’. What is the connection between the real and complex wavelets?”
>
> The main idea of our work builds on a conjecture outlined in the conclusion of Irène Waldspurger’s PhD thesis. The formulation, however, was limited to continuous images and filters, and therefore missed the frequency-dependent nature of shift invariance, as displayed in Figure 5. Our study extends this idea to the discrete setting, incorporating discrete convolutions and max pooling grids. We have updated the introduction to clarify this connection.
>
> > “Page 4 : What is the ‘Gabor hypothesis’?” We agree that this term may be ambiguous. We were referring to the property that learned filters in early CNN layers tend to resemble Gabor-like waveforms. The sentence has been rephrased for clarity.
>
> We thank the reviewer once again for their comments. We hope that the clarified scope, revised manuscript, and supporting empirical work in the companion paper now make the motivation and relevance of our study more apparent.

---

### Author Response · Authors · 2025-05-20
**General response**

We thank all three reviewers for their thoughtful evaluations and constructive feedback. We are pleased that the clarity, theoretical rigor, and focus of our contribution were appreciated. We address below several broader concerns that emerged across multiple reviews.

**Scope of the Study and Focus on the First Layer**

Multiple reviewers noted that our study is restricted to the first convolutional layer, with Gabor-like filters and max pooling. We fully acknowledge this limitation and now highlight it more explicitly in the revised manuscript. However, we emphasize that this choice is both empirically motivated and theoretically justified. Filters in the first layer of CNNs consistently resemble Gabor-like waveforms, which are known to be prone to aliasing and sensitive to shifts. As shown in our companion paper (for which an anonymized version is provided in the supplementary material), instabilities originating from this layer can propagate through the network—see Fig. 2. Thus, understanding its behavior is not only relevant but critical for improving architectural robustness.

**Assumptions and Theoretical Tractability**

Several concerns were raised about the strong assumptions made (e.g., frequency localization). These assumptions were carefully chosen to make the problem mathematically tractable while remaining faithful to observed network behavior. They allow us to rigorously characterize the conditions under which commonly used architectures approximate shift-invariance. While simplifying, these assumptions allowed us to reveal a previously undocumented spectral dependency in the stability of max-pooling-based CNNs, with strong agreement between theory (Figure 5) and experiment (Figures 8 and 9, as well as Fig. 2 in the companion paper).

**Practical Relevance**

Two reviewers questioned whether our theoretical findings translate into practical benefits. While our paper is primarily theoretical, we have also provided empirical validation in our companion paper, where replacing RMax with CMod in the first layer of CNNs improves both stability and accuracy. We present this not as a turnkey solution, but as a proof of concept demonstrating that theoretical insight can inform architectural design. Moreover, such insights can lead to models that are both more interpretable and computationally efficient, particularly when replacing trained filters with deterministic ones.

**Outdated Architectures and Modesty of Claims**

Some concerns were raised regarding the recent shift in the deep learning community toward smaller convolution kernels and the decline of max pooling in favor of alternative downsampling strategies—potentially rendering our study obsolete. While we acknowledge these trends, we emphasize that empirical achievements and theoretical understanding often progress on different time scales, particularly in a field as fast-moving as deep learning. As such, we strongly believe that a theoretical study focusing on legacy architectures—that have been proven successful in the past—can still benefit the community by offering new understanding of the underlying mechanisms at play, inform architectural design decisions, and inspire future research directions.
We also recognize concerns that our contributions may appear modest relative to the technical depth of the manuscript. However, as TMLR’s guidelines emphasize, modest yet well-founded contributions that improve theoretical understanding fall within the journal’s scope and are worth being published. We believe our paper meets this criterion.

---

### Decision · Action_Editor_Gmpv · 2025-07-07

**Recommendation:** Reject

**Additional Comments:**

The submission does not quite meet the bar for acceptance at TMLR. To a majority of the reviewers, the claims made in the submission are simply too narrowly scoped and do not provide a tangible enough new understanding beyond that provided by previous works to be of interest to TMLR's readership.

**Audience:**

No

**Audience Explanation:**

Opinions are split among reviewers on the Audience criterion. Reviewer YRpJ finds that "to [their] best judgement, the analysis is solid and of interest to theoricians". Reviewers uJt4 and 9oEX, however, disagree.

Reviewer uJt4 notes that the submission presents a very lengthy treatment for claims that are restricted to a very narrow setting (first convolutional layer, Gabor-like filter), and that it's unclear how the results would be useful in practice. The authors adjusted the manuscript structure to better guide readers through the core results without delving into all the technical details, and they point out that according to TMLR's acceptance guidelines, papers need not make bold claims to be valuable. This is true, although it applies to the significance of the claims being made. Modest claims can still be valuable, but they have to be of interest to at least part of the TMLR readership, which is what Reviewer uJt4 disputes here.

As for the usefulness of the results in practice, the authors point to the companion paper in the supplementary material which demonstrates practical usefulness. This constitutes a challenge from a presentation perspective: rather than introducing its contributions as e.g. revisiting empirical results and providing theory that explains those results, the submission goes over follow-up results in the Discussion section and presents them as empirical demonstration that "replacing RMax by CMod in the first layer of AlexNet and ResNet architectures improves both shift invariance and prediction accuracy, while reducing computational costs and memory footprint". Were it not for the fact that those results have already been published, Reviewer uJt4 could have rightfully pointed out that they would have greatly strengthened the submission's appeal to the TMLR audience, and that without them the submission does not meet the bar for the Audience criterion.

Reviewer 9oEX is also skeptical of the submission's appeal to at least part of the TMLR readership and points out that CNN architecture models have shifted to smaller filters and reduced use of max-pooling, which reduces the interest of the paper's study. The authors counter that "even if certain components like large convolution kernels and max pooling are less fashionable in current SOTA pipelines, [they] believe that a theoretical study primarily focused on legacy architectures that have been proven successful in the past can offer new understanding of their underlying mechanisms, guide architectural choices and inspire future research directions".

Following the authors' response, Reviewers uJt4 and 9oEX remain unconvinced that the findings are of interest to at least part of the TMLR audience. For instance, Reviewer 9oEX writes: "As previous papers have already shown that high-frequency elements are prone to aliasing and sensitive to shifts (Zhang and other papers mentioned), the case of Gabor filters in the first layer are in my view another class of high pass filters, and not a particular new finding."

**Claims And Evidence:**

No

**Claims Explanation:**

The reviewers have not raised major concerns over claims and evidence pertaining to empirical or theoretical results. Reviewer 9oEX however remains worried that the paper does not contribute to a better understanding of a blind spot in studying aliasing on CNNs with a discrete framework. In their review, they listed recent advances in CNN architectures and works on understanding the impact of shift-invariance / aliasing and modeling Gabor filters. The authors did update the submission to acknowledge those works, but in Reviewer 9oEX's opinion they have not satisfyingly explained what new understanding their submission provides beyond those previous works.